# Ser14 phosphorylation of Bcl-xL mediates compensatory cardiac hypertrophy in male mice

Michinari Nakamura [1] ✉, Mariko Aoyagi Keller [1], Nadezhda Fefelova[1], Peiyong Zhai[1], Tong Liu[2], Yimin Tian[1], Shohei Ikeda[1], Dominic P. Del Re[1], Hong Li[2], Lai-Hua Xie[1] & Junichi Sadoshima [1] ✉

The anti-apoptotic function of Bcl-xL in the heart during ischemia/reperfusion is diminished by K-Ras-Mst1-mediated phosphorylation of Ser14, which allows dissociation of Bcl-xL from Bax and promotes cardiomyocyte death. Here we show that Ser14 phosphorylation of Bcl-xL is also promoted by hemodynamic stress in the heart, through the H-Ras-ERK pathway. Bcl-xL Ser14 phosphorylation-resistant knock-in male mice develop less cardiac hypertrophy and exhibit contractile dysfunction and increased mortality during acute pressure overload. Bcl-xL Ser14 phosphorylation enhances the Ca2+ transient by blocking the inhibitory interaction between Bcl-xL and IP3Rs, thereby promoting Ca2+ release and activation of the calcineurin-NFAT pathway, a Ca2+-dependent mechanism that promotes cardiac hypertrophy. These results suggest that phosphorylation of Bcl-xL at Ser14 in response to acute pressure overload plays an essential role in mediating compensatory hypertrophy by inducing the release of Bcl-xL from IP3Rs, alleviating the negative constraint of Bcl-xL upon the IP3R-NFAT pathway.

The Bcl-2 family proteins mediate cell survival and death through apoptosis-dependent and -independent mechanisms[1,2]. The anti-apoptotic property of Bcl-2 and Bcl-xL provides terminally differentiated organs, such as the heart and brain, with crucial protection against oxidative stress. We previously showed that mammalian Hippo kinase Mst1 phosphorylates Bcl-xL at Serine 14 (Ser14) in cardiomyocytes in response to oxidative stress, which disrupts its interaction with the pro-apoptotic protein Bax, thereby increasing the abundance of active Bax, apoptosis, and myocardial ischemia/reperfusion (I/R) injury in mice[3,4]. In a series of intensive analyses of this signaling pathway in cardiac pathology, we found a biphasic increase of Ser14 phosphorylation in the mouse heart in response to acute pressure overload. This led us to investigate whether this post-translational modification

contributes to cell survival and death in the context of hemodynamic stress in a manner similar to that during oxidative stress and, if so, how it affects cardiac structure and function. Mst1 is not activated in the heart[5] or the cardiomyocytes therein[3] during the acute phase of hypertrophic stimulation, raising the possibility that Bcl-xL Ser14 phosphorylation is mediated by an Mst1-independent mechanism during hypertrophy.

The heart adapts to increased blood pressure by increasing individual cardiomyocyte size, namely hypertrophy, to increase contractility and reduce ventricular wall stress and oxygen consumption[6]. The MEK-ERK and Ca2+-NFAT signaling pathways have been intensively studied as crucial mechanisms for the development of cardiac hypertrophy[7–9]. An improper response to increased workload results in

[1]Department of Cell Biology and Molecular Medicine, Cardiovascular Research Institute, Rutgers-New Jersey Medical School, 185 South Orange Ave, Newark, NJ 07103, USA. [2]Center for Advanced Proteomics Research, Department of Biochemistry & Molecular Biology, Rutgers New Jersey Medical School, Newark, NJ 07103, USA. ✉e-mail: nakamumi@njms.rutgers.edu; sadoshju@njms.rutgers.edu

acute decompensated heart failure with high mortality, due in part to incompletely understood pathophysiology and limited available new therapies[10]. It is important to unveil the missing piece(s) in hypertrophic signaling, normalization of which should prevent acute decompensated heart failure. The current study aims to investigate the functional significance and role of Bcl-xL Ser14 phosphorylation in response to pressure overload. Here we report the unexpected observation that phosphorylation of Bcl-xL at Ser14 plays a critical role in the development of compensatory hypertrophy by enhancing $Ca^{2+}$ transients and calcineurin-NFAT signaling. We found that acute pressure overload induces phosphorylation of Bcl-xL at Ser14 through an H-Ras-ERK1/2-dependent mechanism, thereby alleviating the negative constraint of Bcl-xL upon $Ca^{2+}$ release from IP3Rs.

## Results

### Inhibition of endogenous Bcl-xL phosphorylation exacerbates acute decompensated heart failure in response to pressure overload

First, we examined the phosphorylation status of endogenous Bcl-xL at Ser14 in the heart in the presence of acute pressure overload by applying transverse aortic constriction (TAC) to mouse hearts. Ser14 phosphorylation was significantly increased in the heart within one hour of TAC, returning to baseline by Day 3 but increasing again at around 1 week and thereafter (Fig. 1a), while there was little change in the phosphorylation level in the heart after sham surgery (Supplementary Fig. 1a). In order to demonstrate the functional significance of the increased phosphorylation of Bcl-xL at Ser14,

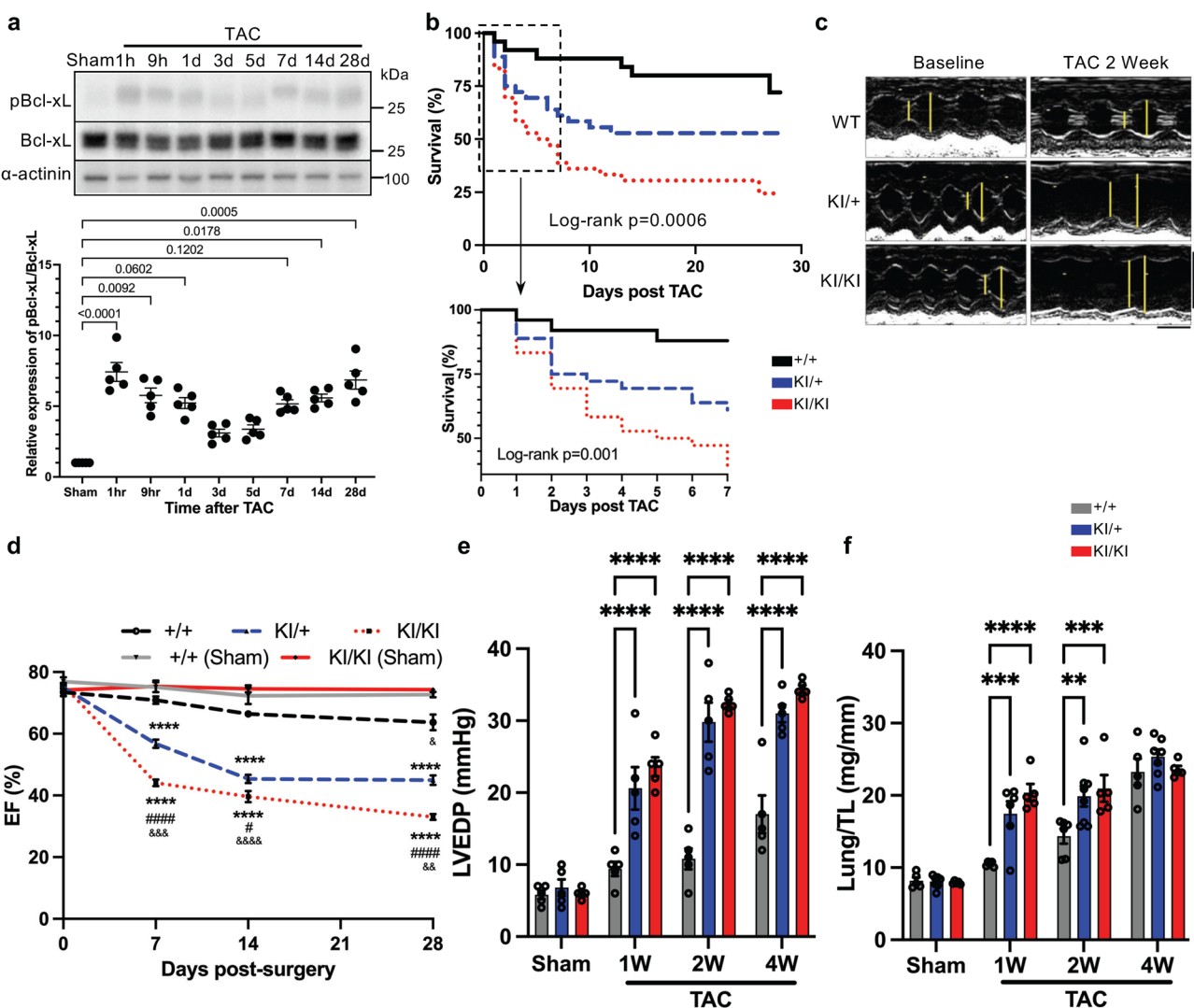

**Fig. 1 | Knock-in (KI) mice in which Serine (Ser) 14 of Bcl-xL is replaced with Alanine show worse phenotypes in response to pressure overload.**
**a** Representative immunoblots showing phosphorylation of Bcl-xL at Ser14 in the heart with time course after transverse aortic constriction (TAC). Sham is a heart sample collected one hour after sham surgery. h; hours, and d; days after TAC. Lower panel shows densitometric analysis of relative expression of pBcl-xL (Ser14)/Bcl-xL in the heart. Kruskal-Wallis test with sham as the control. *p* values are shown in the figure (*n* = 5). **b** Kaplan–Meier survival curves. Log-rank (Mantel-Cox) test *p* = 0.0006. *n* = 25–36. Lower panel shows the survival curves 1 week after TAC. **c** Representative pictures of M-mode echocardiography. Yellow lines indicate left ventricular (LV) end-systolic and -diastolic diameters. Vertical scale bar indicates 5 mm and horizontal scale bar indicates 100 ms. **d** Ejection fraction (EF) with time course after TAC or sham surgery. Two-way ANOVA with Tukey's multiple

comparison test. ****p* < 0.0001 compared to +/+ (wild type) control. #*p* = 0.0183 and ####*p* < 0.0001 compared to knock-in/+. &&&&*p* < 0.0001, &&&*p* = 0.0003, && *p* = 0.0025, and &*p* = 0.0141 compared to Sham (upaired *t* test or Mann–Whitney test, two-sided). *n* = 6–16 (TAC) and 5 (Sham). **e** LV end-diastolic pressure at the indicated time points after TAC or sham surgery, evaluated by hemodynamic study. Two-way ANOVA with Tukey's multiple comparison test. ****  *p* < 0.0001. *n* = 5. **f** Lung weight normalized by tibia length at the indicated time points after TAC or sham surgery. Two-way ANOVA with Tukey's multiple comparison test. ****p* < 0.0001, ***p* < 0.001, and **p* < 0.01. Adjusted *p* values for (**f**): 0.0004 (1 W, +/+ vs KI/ + ), 0.0018 (2 W, +/+ vs KI/ + ), 0.0008 (2 W, +/+ vs KI/KI). *n* = 5–8. *N* represents biologically independent replicates. Data are mean ± SEM. Source data are provided as a Source Data file.

phosphorylation-resistant knock-in mice, in which Ser14 has been replaced with Ala (S14A knock-in mice)[4], and control wild type (WT) mice were subjected to TAC surgery. Although the pressure gradient between the ascending aorta and the femoral artery 2 weeks after TAC tended to be less in homozygous S14A knock-in mice compared to in WT mice due to cardiac dysfunction ($p = 0.09$ between WT and homozygous S14A knock-in mice, unpaired $t$ test), there was no statistically significant difference among WT, heterozygous, and homozygous S14A knock-in mice (Supplementary Fig. 1b). Significantly less Bcl-xL Ser14 phosphorylation in S14A knock-in mouse hearts than in WT mice one hour after TAC confirms the specificity of our Bcl-xL-Ser14 phosphorylation-specific antibody (Supplementary Fig. 1c). In contrast to the protection conferred by the Bcl-xL S14A knock-in against myocardial I/R injury[4], S14A knock-in mice exhibited a significantly higher mortality rate after TAC than WT mice. Homozygous S14A knock-in mice exhibited the highest mortality, while heterozygous S14A knock-in mice exhibited an intermediate mortality rate (Log-rank (Mantel-Cox) test $p = 0.0006$) (Fig. 1b). Notably, most of the TAC-induced acute death in homozygous and heterozygous S14A knock-in mice was observed within the first week (Log-rank (Mantel-Cox) test $p = 0.001$). We evaluated cardiac function with echocardiography and hemodynamic analyses. There was no significant difference in cardiac function or chamber size among the groups following sham operation. However, both heterozygous and homozygous S14A knock-in mice exhibited a significantly decreased ejection fraction (EF), a measure of left ventricular contractile function, compared to WT mice, which was observed as early as 1 week post-TAC (Fig. 1c, d and Supplementary Fig. 1d). Left ventricular end-diastolic pressure (LVEDP) and lung weight normalized by tibia length, markers of congestive heart failure, were significantly elevated in S14A knock-in mice 1 week after TAC (Fig. 1e, f). These data indicate that S14A knock-in mice develop acute decompensated heart failure in response to hemodynamic stress, and that phosphorylation of Bcl-xL at Ser 14 is an adaptive response to acute pressure overload.

## Inhibition of Bcl-xL phosphorylation suppresses cardiac hypertrophy

To explore the mechanism by which pressure overload exacerbates cardiac dysfunction in S14A knock-in mice, we evaluated the degree of hypertrophy. Increases in the left ventricular wall thickness caused by cardiac hypertrophy reduce ventricular wall stress, thereby playing an adaptive role during the acute phase of pressure overload[6]. Both heterozygous and homozygous S14A knock-in mice exhibited significantly less hypertrophy than WT mice 1 to 4 weeks after TAC, as evidenced by a lower heart weight to tibia length ratio and thinner wall thickness (Fig. 2a and Supplementary Fig. 1d). Wheat Germ Agglutinin (WGA) staining showed smaller individual cardiomyocytes in S14A knock-in mice than in WT mice (Fig. 2b, c). qPCR analyses showed that TAC-induced upregulation of fetal type genes, including *NPPA* and *NPPB*, *MYH7*, and *Rcan1.4*, was suppressed in S14A knock-in mice (Fig. 2d). End-diastolic left ventricular wall stress 1 week post-TAC was significantly greater in both heterozygous and homozygous S14A knock-in mice than in WT mice (Fig. 2e). These results suggest that the pressure overload is not properly counterbalanced with compensatory hypertrophy in both heterozygous and homozygous S14A knock-in mice. We used homozygous S14A knock-in mice for the rest of the study. Since Ser14 phosphorylation of Bcl-xL inhibits the anti-apoptotic functions of Bcl-xL during myocardial I/R[3], we speculated that S14A knock-in may protect the heart against apoptosis during pressure overload. Pressure overload slightly increased apoptosis 2 weeks after TAC in WT mice. However, the percentage of TUNEL-positive nuclei was similar between WT and S14A knock-in mice after 2 weeks of TAC (Fig. 2f, g) and 4 weeks of TAC (Supplementary Fig. 1e). It is important to distinguish between the pathological roles of cardiomyocyte and non-cardiomyocyte apoptosis; thus, we further evaluated cardiomcyote apoptosis by co-staining the heart tissue with

cardiac troponin T and TUNEL 2 weeks after TAC. The rate of cardiomyocyte apoptosis was indistinguishable between WT and S14A knock-in mice (Supplementary Fig. 1f). Cleaved caspase 3 and 9 were slightly elevated 2 weeks post-TAC to a similar extent in both WT and S14A knock-in mice (Fig. 2h). We also isolated adult cardiomyocytes from WT and S14A knock-in mice and cultured them with or without phenylephrine (PE) for 24 hs. mRNA expression of *NPPA*, *MYH7*, and *Rcan1.4* in response to PE was significantly greater in WT cardiomyocytes than in those isolated from S14A knock-in mice (Supplementary Fig. 1g), suggesting that the effect of S14A knock-in upon hypertrophy may be at least partially cell-autonomous. These data suggest that Bcl-xL Ser14 phosphorylation is critical for the development of compensatory hypertrophy rather than promoting apoptosis during the acute phase of pressure overload.

## H-Ras-MEK-ERK signaling is crucial for the increase in Bcl-xL phosphorylation in response to hypertrophic stimuli, which promotes nuclear localization of NFAT3

Since Mst1 is activated in response to pressure overload on Day 3 at the earliest[5] but phosphorylation of Bcl-xL at Ser14 is observed within one day, the early Ser14 phosphorylation of Bcl-xL in response to acute pressure overload may be independent of Mst1. In order to better understand which signaling pathway is affected in the heart during the acute phase of pressure overload, we performed RNA-sequencing analyses using mouse hearts subjected to 9 hs of pressure overload (Fig. 3a). As in the qPCR analyses, hypertrophy marker genes, including *NPPA* and *MYH7*, were upregulated after TAC in WT, but not S14A knock-in, mouse hearts (Supplementary Fig. 2a). The gene set enrichment analysis (GSEA) indicated that Fcε receptor 1, ERK, and integrin pathways were upregulated in WT mouse hearts in response to pressure overload (Fig. 3b and Supplementary Table 1). Since these pathways all utilize the Ras-MEK-ERK pathway, a well established mechanism involved in the development of cardiac hypertrophy[8,11], we hypothesized that the Ras-MEK-ERK axis may be activated to promote Bcl-xL Ser14 phosphorylation in the heart during acute pressure overload.

Consistent with the results of the RNA-sequencing analysis, ERK1/2 phosphorylation was increased but the level of total ERK1/2 protein was unaltered in the heart after 9 h of TAC (Fig. 3c). Phosphorylation of ERK1/2, but not Mst1, was also rapidly induced in response to PE, an α1-adrenergic receptor agonist, in neonatal rat ventricular cardiomyocytes, along with Bcl-xL Ser14 phosphorylation (Fig. 3d, e). Phosphorylation of Akt, a well-known regulator of cardiac hypertrophy[6,12,13], was also increased in response to PE. To examine whether activation of either ERK or Akt is involved in Bcl-xL phosphorylation at Ser14 in response to hypertrophic stimuli, cardiomyocytes were pretreated with a MEK inhibitor, PD0325901, or an Akt inhibitor, Triciribine, and stimulated with PE. Inhibition of MEK, but not Akt, attenuated PE-induced increases in Bcl-xL phosphorylation at Ser14 (Fig. 3f, g and Supplementary Fig. 2b), suggesting that the MEK-ERK axis functions upstream of Bcl-xL Ser14 phosphorylation. To determine whether ERK1 directly phosphorylates Bcl-xL at Ser14, we performed in vitro kinase assays with active ERK1 and recombinant GST-Bcl-xL (Supplementary Fig. 2c). Subsequent LC-MS/MS analyses showed that ERK1 directly phosphorylates Bcl-xL at Ser14 (Fig. 3h).

Hypertrophic stimuli activate H-Ras, which, in turn, promotes nuclear translocation of NFAT3 in cardiomyocytes[3,14]. To investigate whether Bcl-xL phosphorylation at Ser14 is critical for activation of NFAT3, adenovirus harboring H-Ras was transduced into adult mouse cardiomyocytes isolated from WT and S14A knock-in mice (Supplementary Fig. 2d). H-Ras increased nuclear localization of NFAT3 in WT, but not S14A knock-in, cardiomyocytes (Fig. 3i, j). The BH4 domain of Bcl-2 family proteins interacts with H-Ras[15–17]. Co-immunoprecipitation assays showed that binding of H-Ras to Bcl-xL was greater in hearts

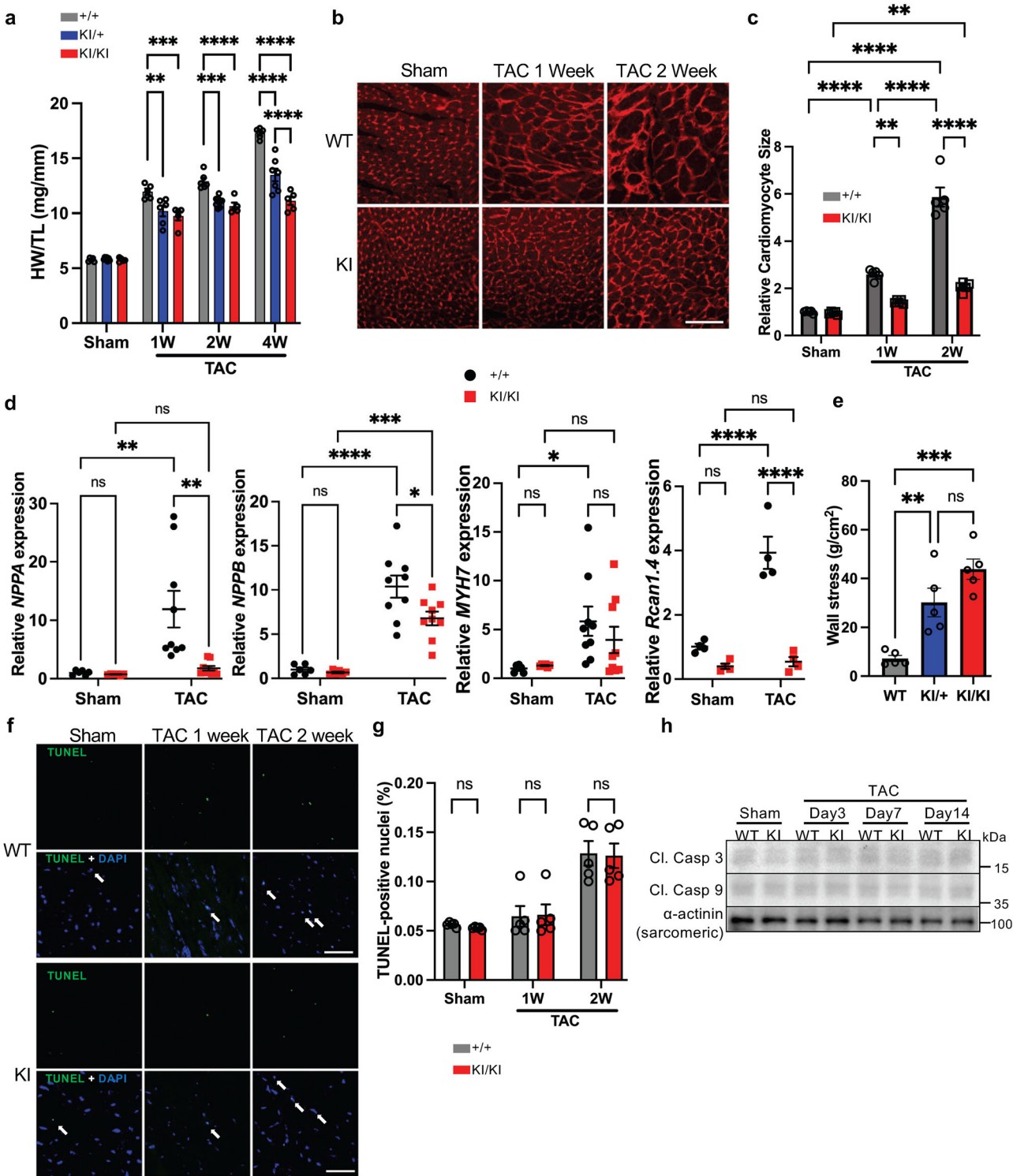

subjected to TAC than in those without TAC (Fig. 3k). The PE-induced increase in the interaction between Bcl-xL and H-Ras was also observed in cardiomyocytes expressing FLAG-Bcl-xL in FLAG-pull down-coimmunoprecipitation assays (Fig. 3l). These results suggest that the H-Ras-MEK-ERK pathway may promote Bcl-xL Ser14 phosphorylation, which in turn activates NFAT signaling in cardiomyocytes.

We further investigated the involvement of the H-Ras-MEK-ERK pathway in Ser14 Bcl-xL phosphorylation in the heart in vivo. WT and S14A knock-in mice were pretreated with a MEK inhibitor, PD0325901, or vehicle for three days. The mice were then subjected to either TAC or sham operation in the presence of PD035901 or vehicle. Inhibition

of MEK suppressed the development of hypertrophy in WT mice, as assessed by echocardiographic measurement of the wall thicknesses of the interventricular septum at end-diastole (IVSd) and the LV posterior wall at end-diastole (LVPWd), heart weight normalized by tibia length, and individual cardiomyocyte size, after 1 week of TAC (Supplementary Fig. 2e–h). The suppression of cardiac hypertrophy was accompanied by exacerbation of contractile dysfunction and heart failure, as assessed by echocardiographically measured EF and lung weight normalized by tibia length (Supplementary Fig. 2f, g). However, inhibition of MEK failed to elicit any additive effect on the S14A knock-in phenotype after 1 week of TAC. These data suggest that Ser14 Bcl-xL

**Fig. 2 | Phosphorylation of Bcl-xL at Serine 14 is essential for compensatory hypertrophy in response to pressure overload.** Both heterozygous and homozygous Serine (S14A) knock-in (KI) mice were used in **a** and **e** whereas homozygous mice were used in **b**–**d** and **f**–**h**. **a** Heart weight normalized by tibia length at the indicated time points after TAC or sham. Sham: +/+ $n = 5$, KI/+ $n = 7$, KI/KI $n = 5$; 1 W: +/+ $n = 5$, KI/+ $n = 6$, KI/KI $n = 5$; 2 W: +/+ $n = 6$, KI/+ $n = 8$, KI/KI $n = 5$; 4 W: +/+ $n = 5$, KI/+ $n = 7$, KI/KI $n = 5$. **b** Wheat Germ Agglutinin (WGA) staining of the indicated heart tissues. Scale bar; 100 μm. **c** Quantitative analysis of relative cardiomyocyte size. $n = 5$. **d** Relative *NPPA*, *NPPB*, *MYH7*, and *Rcan1.4* gene expressions. *NPPA*, *NPPB*, and *MYH7*: $n = 6$ (sham) and 9 (TAC). *Rcan1.4*: $n = 4$. **e** Calculated end-diastolic wall stress of the indicated mice 1-week post-TAC ($n = 5$). **f** Terminal deoxynucleotidyl transferase dUTP nick end labeling (TUNEL) staining of the indicated heart tissues. Arrows indicate TUNEL-positive nuclei. Scale bar; 100 μm. **g** Quantitative analysis of TUNEL-positive nuclei. $n = 5$. **h** Immunoblots showing cleaved caspase 3 and 9 expression levels in the heart after TAC or sham. Immunoblots were repeated at least three times using biologically independent replicates. In all graphs, WT is indicated by +/+ and knock-in (KI) is indicated by KI/+ (heterozygous) or KI/KI (homozygous). $n$ represents biologically independent replicates. Two-way ANOVA with Tukey's multiple comparison test. ****$p < 0.0001$, ***$p < 0.001$, **$p < 0.01$, *$p < 0.05$, ns not significant. Adjusted $p$ values for (**a**): 0.0015 (1 W, +/+ vs KI/+), 0.0001 (1 W, +/+ vs KI/KI), 0.0002 (2 W, +/+ vs KI/+). Adjusted $p$ values for (**c**): 0.0012 (TAC 1 W, +/+ vs KI/KI), 0.0021 (KI/KI, sham vs TAC 2 W). Adjusted $p$ values for (**d**): 0.0029 (*NPPA*: WT, Sham vs TAC), 0.002 (*NPPA*: TAC, +/+ vs KI/KI), 0.0251 (*NPPB*: TAC, +/+ vs KI/KI), 0.0004 (*NPPB*: KI/KI, Sham vs TAC), 0.0491 (*MYH7*: +/+, Sham vs TAC). Adjusted $p$ values for (**e**): 0.0061 (WT vs KI/+), 0.0001 (WT vs KI/KI). Data are mean ± SEM. Source data are provided as a Source Data file.

phosphorylation and the H-Ras-MEK-ERK pathway act on the same signaling pathway, thereby mediating TAC-induced cardiac hypertrophy. Furthermore, these results suggest that the H-Ras-MEK-ERK1/2 signaling acts upstream of Ser14 Bcl-xL phosphorylation during pressure overload-induced cardiac hypertrophy.

## Ser14 phosphorylation of Bcl-xL promotes Ca2+ signaling in response to pressure overload

We then explored the mechanism by which Ser14 Bcl-xL phosphorylation mediates cardiac hypertrophy in response to pressure overload. Since it has been suggested previously that $Ca^{2+}$ serves as a critical second messenger to induce hypertrophy and non-canonical interaction between the BH4 domain of Bcl-xL and sarcoplasmic reticulum (SR)/endoplasmic reticulum (ER)[18,19], we evaluated the $Ca^{2+}$ transient in cardiomyocytes isolated from S14A knock-in and WT mice. There was no significant difference in the $Ca^{2+}$ transient amplitude, SR $Ca^{2+}$ content, fractional $Ca^{2+}$ release, or $T_{50}$ between WT and S14A knock-in mice at baseline (Fig. 4a and Supplementary Fig. 3a). Pressure overload increased the $Ca^{2+}$ transient amplitude in cardiomyocytes isolated from WT mice, but decreased it in cardiomyocytes from S14A mice, after one day of TAC. Pressure overload increased the SR $Ca^{2+}$ content in cardiomyocytes isolated from WT mice, but not S14A mice. Pressure overload did not significantly affect fractional $Ca^{2+}$ release or $T_{50}$ compared to sham operation in cardiomyocytes isolated from WT mice. On the other hand, a significantly decreased fractional $Ca^{2+}$ release and increased $T_{50}$ were observed after TAC in cardiomyocytes isolated from S14A knock-in mice compared to in those from WT mice (Fig. 4a and Supplementary Fig. 3a). We further assessed the contractile function of individual cardiomyocytes after two days of TAC. Cardiomyocytes isolated from S14A knock-in mice exhibited significantly reduced contraction compared to those from WT mice, consistent with the impaired $Ca^{2+}$ signaling in S14A knock-in mice after pressure overload (Supplementary Fig. 3b, c). Along with suppressed compensatory hypertrophy, impaired contractility with a decrease in $Ca^{2+}$ transient amplitude may also contribute to the decompensated heart failure and high mortality of the S14A knock-in mice during the acute phase of pressure overload.

The inositol 1,4,5-triphosphate receptor (IP3R) plays a central role in the development of cardiac hypertrophy, with functional redundancy in all 3 types of IP3Rs[20,21]. To examine whether IP3Rs mediate the effect of Bcl-xL Ser14phosphorylation, 2-APB, a membrane permeable IP3R antagonist, was applied during $Ca^{2+}$ transient measurements in cardiomyocytes (Fig. 4b). 2-APB decreased the $Ca^{2+}$ transient amplitude in WT cardiomyocytes, but not in S14A knock-in cardiomyocytes, after one day of TAC (Fig. 4b, c). Although 2-APB slightly increased the $Ca^{2+}$ transient amplitude in S14A knock-in cardiomyocytes, underlying mechanisms are unknown. The differences in $Ca^{2+}$ transient amplitude, SR $Ca^{2+}$ content, and fractional $Ca^{2+}$ release between WT and the S14A knock-in cardiomyocytes after TAC were abolished in the presence of 2-APB (Fig. 4d). The difference in T50 was not abolished but became smaller in the presence of 2-APB. These results suggest that IP3R

functions downstream of Bcl-xL Ser14 phosphorylation in cardiomyocytes in response to pressure overload.

## Phosphorylation of Bcl-xL disrupts its inhibitory interaction with IP3R

The results presented thus far suggest that Bcl-xL Ser14 phosphorylation promotes pressure overload-induced SR $Ca^{2+}$ release through IP3Rs. Consistent with this notion, the GSEA showed that calcineurin-mediated signaling is significantly suppressed in S14A knock-in mouse hearts subjected to 9 h of pressure overload compared to in WT mouse hearts with pressure overload (Fig. 5a). The calcineurin-NFAT pathway is activated in response to increases in cytosolic $Ca^{2+}$ and required for the development of compensatory hypertrophy[9]. To validate the result of the pathway analysis, we conducted NFAT-luciferase (NFAT-Luc) assays in cardiomyocytes expressing Bcl-xL-WT, Bcl-xL-S14A (SA) or Bcl-xL-S14D (SD) (Supplementary Fig. 3d). PE significantly increased the NFAT-Luc activity, but this increase was attenuated in the presence of 2-APB in cardiomyocytes transduced with adenovirus harboring Bcl-xL-WT (Fig. 5b). The PE-induced increase in NFAT activity was significantly attenuated in cardiomyocytes transduced with adenovirus harboring Bcl-xL-S14A and was not further inhibited in the presence of 2-APB (Fig. 5b). These results suggest that Ser14 Bcl-xL phosphorylation is required for PE-induced activation of calcineurin-NFAT signaling through $Ca^{2+}$ release from IP3Rs.

We further explored the mechanism by which Bcl-xL Ser14 phosphorylation enhances IP3R-NFAT signaling in response to hypertrophic stimuli. Immunoprecipitation assays showed that pressure overload decreases the physical interaction between Bcl-xL and IP3R-type 2 in WT mouse hearts, but not S14A knock-in mouse hearts, in the presence of TAC (Fig. 5c), suggesting that Ser14 phosphorylation during TAC inhibits Bcl-xL-IP3R interaction. Recombinant GST-IP3R-fragment 3, which is known to bind to Bcl-xL[22], was pulled down more efficiently with flag-Bcl-xL-S14A than with flag-Bcl-xL-S14D, a phosphorylation-mimicking mutant (Fig. 5d). Thus, direct physical interaction between Bcl-xL and IP3R-fragment 3 is negatively regulated by Bcl-xL Ser14 phosphorylation. We further investigated the functional significance of the interaction between Bcl-xL and IP3Rs using a membrane permeable synthetic peptide harboring partial IP3R-fragment 3 (Fragment 3-P)[23]. The PE-induced increase in NFAT-Luc activity in cardiomyocytes expressing wild-type Bcl-xL was not significantly affected by Fragment 3-P. In contrast, although the PE-induced increase in NFAT-Luc activity was significantly attenuated in cardiomyocytes expressing Bcl-xL-S14A, the effect of PE was fully rescued in the presence of Fragment 3-P (Fig. 5e). These data suggest that phosphorylation of Bcl-xL at Ser14 plays a critical role in stimulating calcineurin-NFAT signaling by inhibiting the physical interaction between Bcl-xL and IP3R-fragment 3, a key mechanism checking $Ca^{2+}$ release from the SR[9]. We also investigated whether Bcl-xL S14D is sufficient to induce NFAT activation and hypertrophy or augment cardiac hypertrophy in response to PE treatment. Bcl-xL S14D neither activated NFAT nor induced or augmented hypertrophy in

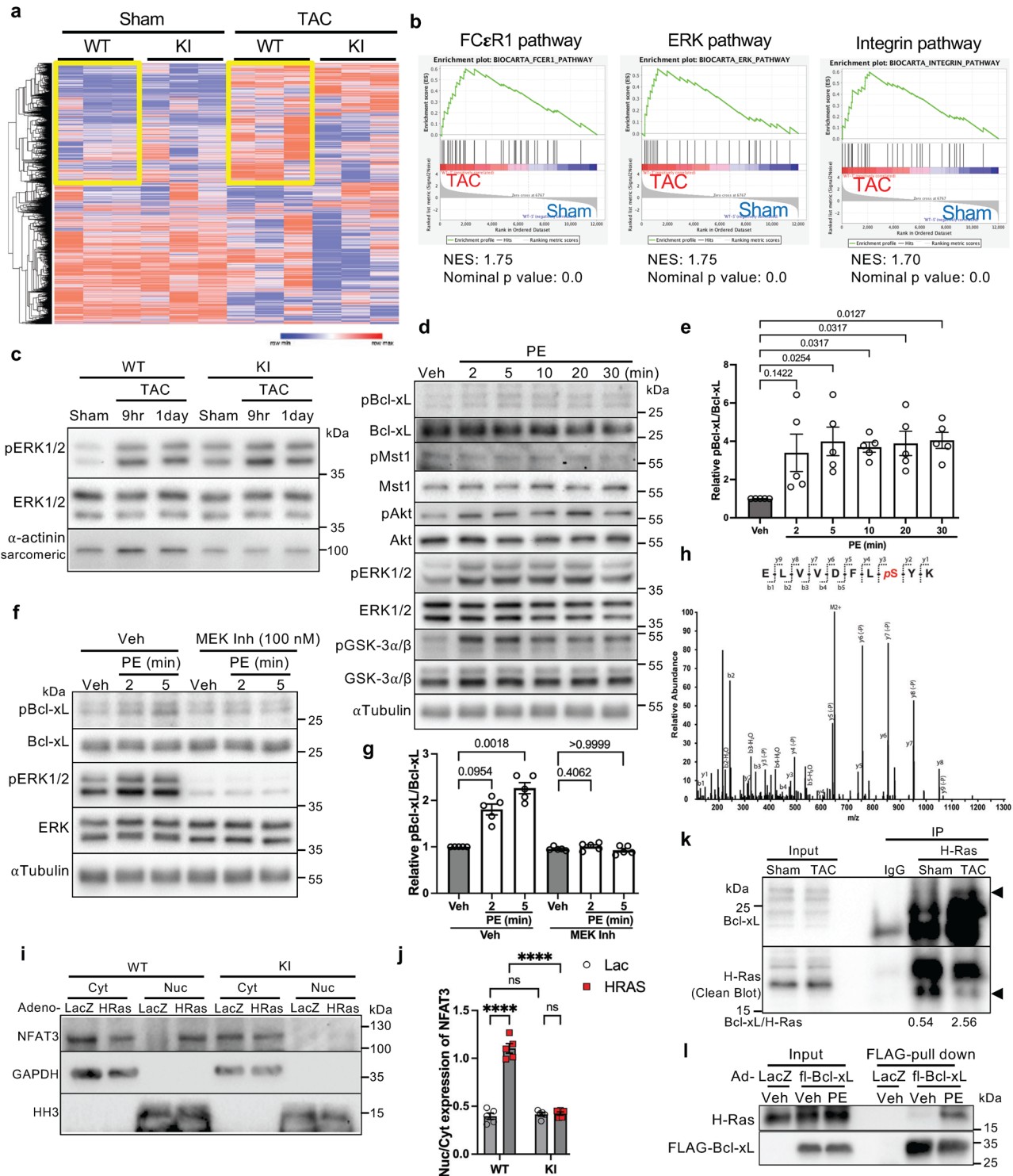

cardiomyocytes at baseline or in the presence of PE (Supplementary Fig. 3e-g), suggesting that overexpression of Bcl-xL S14D is not sufficient to activate NFAT or induce/enhance hypertrophy in cardiomyocytes. These results are consistent with the notion that Ser14 Bcl-xL phosphorylation mediates compensatory hypertrophy during the acute phase of pressure overload by controlling Ca$^{2+}$ release from IP3Rs (Supplementary Fig. 3h and 4).

Although Bcl-2 family proteins regulate autophagy activity by interacting with the BH3 domain of Beclin-1[24,25], S14A knock-in mice showed no overt change in the levels of Beclin-1, p62, or LC3-I/II (Supplementary Fig. 3i)[4], indicating that Bcl-xL Ser14 phosphorylation

does not affect autophagy activity at baseline. To investigate whether Bcl-xL phosphorylation-mediated Ca$^{2+}$ release takes place only in cardiomyocytes or in other types of cells in the heart as well, we isolated neonatal cardiac fibroblasts and measured cytosolic Ca$^{2+}$ levels. WT cardiac fibroblasts exhibited increased Ser14 phosphorylation and cytosolic Ca$^{2+}$ levels in response to angiotensin II stimulation, whereas S14A knock-in cardiac fibroblasts showed a significantly smaller elevation of cytosolic Ca$^{2+}$ (Supplementary Fig. 3j), indicating that Bcl-xL phosphorylation is critical for enhancing Ca$^{2+}$ release not only in cardiomyocytes but also in cardiac fibroblasts in response to hypertrophic stimuli.

**Fig. 3 | The Ras-MEK-ERK pathway is activated immediately after hypertrophic stimulation, phosphorylating Bcl-xL at Serine 14 and promoting nuclear translocation of NFAT3. a** Heat map of differentially expressed genes in the hearts of WT and Bcl-xL-S14A knock-in (KI) mice 9 hs after TAC or sham surgery. $n = 3$. **b** Gene set enrichment analysis plots of the FCεR1, ERK and Integrin pathways enriched after TAC compared to sham in WT mice (Supplementary Table 1). NES, normalized enrichment score. GSEA nominal $p$ value is the statistical significance of the enrichment score by using a phenotype-based permutation test with no adjustment. **c** Immunoblots showing the phosphorylation status of ERK1/2 in the hearts of WT and KI mice after TAC or sham surgery. Immunoblots were repeated at least three times using biologically independent replicates. **d** Immunoblots showing the phosphorylation status of Bcl-xL (Ser14) in cardiomyocytes after phenylephrine (PE). **e, g** Relative expression of pBcl-xL (Ser14)/Bcl-xL. Kruskal–Wallis test with vehicle as the control. $p$ values are shown in the figure ($n = 5$). **f** Immunoblots showing the effect of a MEK inhibitor (PD0325901) on PE-induced Bcl-xL-Ser14

phosphorylation in cardiomyocytes. **h** MS/MS spectrum of a doubly charged ion ($m/z$ 646.81) corresponding to the peptide sequence $^7$ELVVDFL$p$SYK$^{16}$ with a phosphorylation modification at S$^{14}$ in Bcl-xL. The observed $y$- and $b$-ion series confirmed the peptide sequence and phosphorylation modification site. **i, j** Immunoblots showing the nuclear and cytosolic localization of NFAT3 in WT and S14A KI adult mouse cardiomyocytes transduced with adenovirus harboring H-Ras or LacZ (**i**) and its quantification analysis (**j**, $n = 5$). Two-way ANOVA with Tukey's multiple comparison test. $^{****}p < 0.0001$, ns not significant. **k** Immunoprecipitation assay using α-H-Ras antibody with heart lysates from mice subjected to 9 hs of pressure overload. The numbers indicate the ratio of Bcl-xL (upper arrow) to H-Ras (lower arrow) by densitometric analysis. **l** FLAG-pull down assay using rat neonatal ventricular cardiomyocytes transduced with adenovirus harboring FLAG-Bcl-xL for two days in the presence of PE or vehicle for 20 mins. Immunoblots were repeated at least three times using independently prepared cardiomyocytes. $n$ represents biologically independent replicates. Data are mean ± SEM.

## Discussion

We show that activation of the H-Ras-MEK1-ERK1/2 pathway by hypertrophic stimuli promotes Bcl-xL-Ser14 phosphorylation, which disrupts the inhibitory interaction between Bcl-xL and IP3Rs, thereby augmenting SR Ca$^{2+}$ release and activating the calcineurin-NFAT pathway, a major signaling mechanism controlling cardiac hypertrophy. This mechanism is crucial for the development of adaptive hypertrophy and cardiomyocyte contraction, and suppression of this mechanism promotes acute decompensated heart failure. These results suggest that modulating the level of Bcl-xL Ser14 phosphorylation and downstream mechanisms, including the interaction between Bcl-xL and IP3R, could be a promising intervention against heart failure during acute pressure overload.

Increasing evidence indicates that Bcl-2 family proteins, including Bcl-2 and Bcl-xL, modulate Ca$^{2+}$ signaling through a non-canonical mechanism independent of their actions upon mitochondrial outer membrane permeabilization, namely interaction with IP3Rs located on the ER/SR. The BH4 domain of Bcl-2 interacts with IP3R fragment 3, which inhibits the IP3R channel gating[23,26,27]. Lysine (K) 17 in the BH4 domain of Bcl-2 plays a critical role in mediating the inhibitory interaction between Bcl-2 and IP3R fragment 3, although K17 in Bcl-2 is not conserved in the BH4 domain of Bcl-xL, which has D11 instead[28]. In contrast to the inhibitory action of Bcl-2 on IP3R, binding of Bcl-xL to two carboxyl terminal BH3-like domains of IP3R *sensitizes* IP3R to low concentrations of IP3, thereby stimulating Ca$^{2+}$ release, mitochondrial bioenergetics and cell survival[29,30]. However, Bcl-xL at high concentrations binds to the IP3R fragment 3, thereby inhibiting IP3R channel gating. Furthermore, a synthetic peptide targeting the IP3R fragment 3 completely blocks the inhibitory effect of Bcl-xL on IP3Rs[30], suggesting that the effect of Bcl-xL on IP3R is context-dependent. Another study also showed that Bcl-xL always inhibits, rather than activates, IP3R in vitro, primarily by binding to the fragment 3 as well as the ligand-binding domain of IP3Rs[22]. Our data indicate that Bcl-xL binds to IP3Rs and that the disruption of this binding enhances IP3R-mediated Ca$^{2+}$ release in the context of hypertrophic stimuli. We speculate that abundant expression of Bcl-xL in the heart may contribute to the inhibitory interaction with IP3Rs.

Importantly, we here show that the interaction between Bcl-xL and IP3Rs is regulated by Bcl-xL Ser14 phosphorylation. A prior study demonstrated that K17D mutation in the BH4 domain of Bcl-2 abrogates the ability of the BH4 domain to bind to and inhibit IP3R, whereas D11K mutation in the BH4 domain of Bcl-xL rendered the BH4 domain of Bcl-xL capable of binding to and inhibiting IP3R[28], suggesting the crucial role of a *negative charge* in preventing the interaction. Ser14 phosphorylation introduces a negative charge in the BH4 domain of Bcl-xL, which may contribute to disrupting the interaction with IP3R. We here show that interaction between Bcl-xL and IP3Rs and consequent activation of NFAT are negatively regulated by Bcl-xL phosphorylation at Ser14, located in the BH4 domain. Ser14 Bcl-xL

phosphorylation may disrupt the interaction with IP3R either by adding a negative charge to the BH4 domain or inducing an allosterical conformational change in another part of Bcl-xL, including the BH3 domain. K17 in Bcl-2 is not conserved in Bcl-xL, nor is S14 in Bcl-xL conserved in Bcl-2. However, both K17 in Bcl-2 and S14 in Bcl-xL allow acidic posttranslational modification in the BH4 domain. Thus, posttranslational modification of K17 and S14 may allow the BH4 domains of these Bcl-2 family proteins to control Ca$^{2+}$ signaling through IR3Rs in a regulated manner. We propose that the phosphorylation status of Ser14 in the BH4 domain of Bcl-xL could serve as a convenient access point to control cardiac contractility and hypertrophy by modulating the interaction between Bcl-xL and IP3R in the heart.

Cardiac hypertrophy is required to maintain contractile function and pump sufficient blood throughout the peripheral organs in the face of increased afterload, but persistent hyperactivation of hypertrophic signaling results in cardiac dysfunction, such as aberrant activation of mTOR and YAP[6,13,31]. Phenylephrine, an α1-adrenergic receptor agonist, is generally cardioprotective in humans[32]. We here show that inhibition of Bcl-xL Ser14 phosphorylation suppresses the hypertrophic signaling mechanism during phenylephrine treatment in vitro, consistent with the notion that Bcl-xL Ser14 phosphorylation mediates adaptive cardiac hypertrophy. The Ca$^{2+}$-calcineurin pathway plays a central role in developing hypertrophy[9]. Inhibition of calcineurin with cyclosporine attenuates pressure overload-induced hypertrophy and enhances susceptibility to decompensation and heart failure[33], a phenotype similar to that of Bcl-xL-S14A knock-in mice. We have also shown that a lack of sufficient hypertrophy during the acute phase of pressure overload induces heart failure in heterozygous cardiac-specific YAP KO mice[34]. Since we observed decreases in contractility in single ventricular cardiomyocytes isolated from S14A knock-in mouse hearts after TAC, however, other mechanisms besides hypertrophy regulated by Bcl-xL Ser14 phosphorylation may also contribute to the failing phenotype in S14A knock-in mice during TAC. Further investigation is needed to clarify this issue.

A previous study showed that overexpression of IP3R2 in cardiomyocytes significantly enhances TAC-induced hypertrophy, whereas 2-week TAC-induced hypertrophy is not inhibited in cardiomyocyte-specific IP3-sponge transgenic mice[21]. We have shown that S14A knock-in alone induces an intracellular Ca$^{2+}$ environment in cardiomyocytes similar to that in WT cardiomyocytes treated with 2-APB, an inhibitor of IP3R, and that disinhibition of IP3Rs with IP3R-fragment 3 rescued the Ca$^{2+}$-NFAT signaling defect in cardiomyocytes expressing Bcl-xL-S14A. Nevertheless, we cannot formally exclude the presence of unknown Ca$^{2+}$ handling mechanisms regulated by Bcl-xL-Ser14 phosphorylation. In addition, negative regulation of the Ca$^{2+}$ transient through Bcl-xL-Ser14 phosphorylation also exists in the cardiac fibroblast population. How changes in Ca$^{2+}$ handling in cardiac fibrobalsts contribute to the cardiac phenotype in response to TAC in S14A knock-in mice remain to be elucidated.

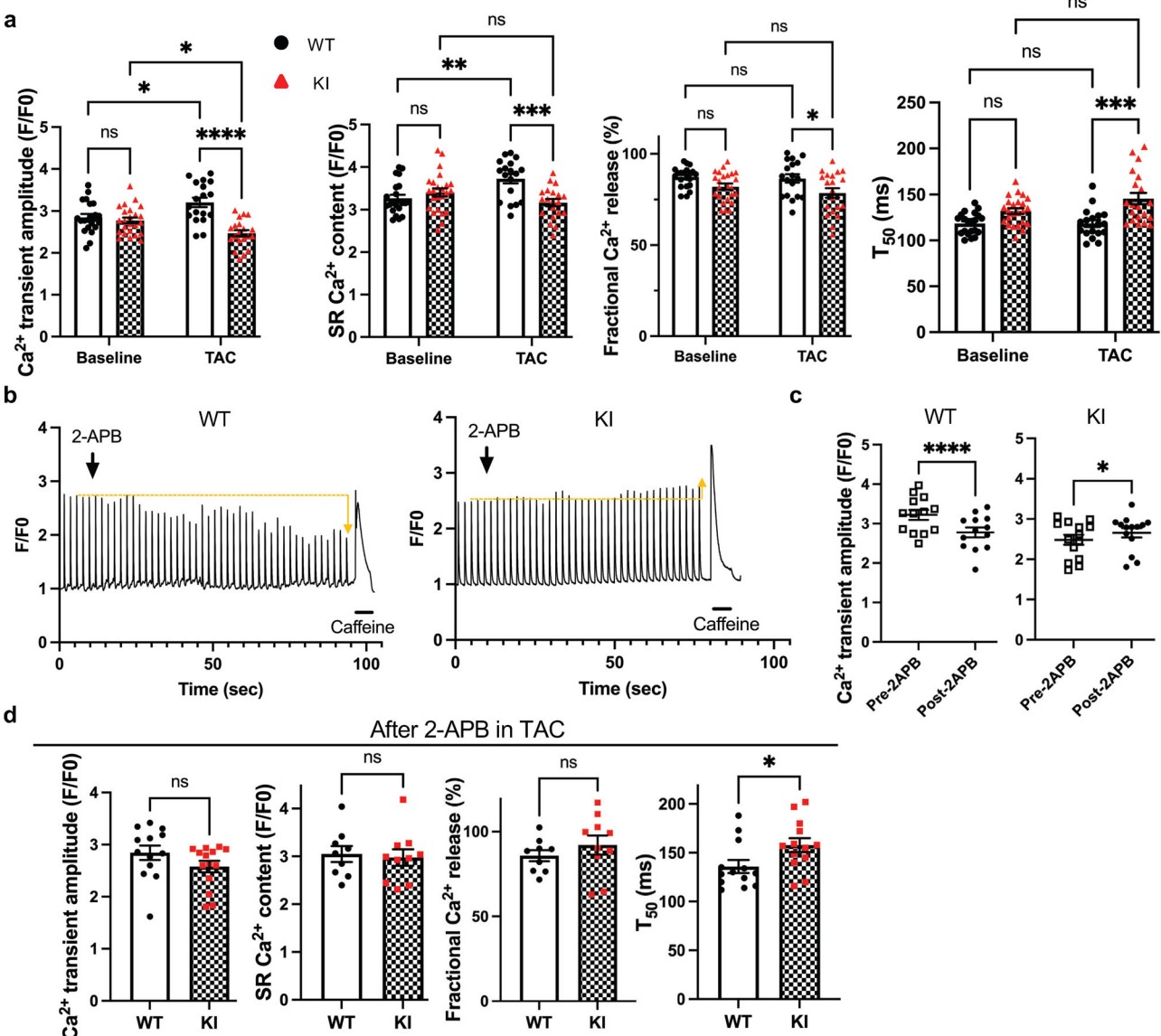

**Fig. 4 | Wild type (WT), but not Bcl-xL-S14A knock-in (KI), cardiomyocytes exhibit increases in Ca²⁺ transient amplitude and sarcoplasmic reticulum (SR) Ca²⁺ content after TAC-induced pressure overload, which are suppressed by IP3R inhibition.** **a** Hemodynamic stress differentially alters intracellular Ca²⁺ dynamics in WT and KI cardiomyocytes, as shown in the Ca²⁺ transient amplitude, SR Ca²⁺ content, fractional Ca²⁺ release, and $T_{50}$. Cardiomyocytes were isolated from the indicated mouse hearts 1 day after TAC. WT baseline $n = 21$; WT TAC $n = 18$; KI baseline $n = 24$; KI TAC $n = 20$ from 3 hearts/group. Two-way ANOVA with Tukey's multiple comparison test. **b** Representative traces of Ca²⁺ transient (*F/FO*) with 2-APB treatment in WT and KI cardiomyocytes 1 day after TAC. Orange dotted lines indicate the level of *F/FO* at the time of 2-APB administration and arrows indicate the direction of change in *F/FO*. **c** Ca²⁺ transient amplitude before and after 2-APB treatment in cardiomyocytes 1 day after TAC. WT $n = 13$; KI $n = 14$ cells from 3

hearts/group. Two-sided paired *t* test (WT) or Wilcoxon matched-pairs signed rank test (KI). **d** Quantification of intracellular Ca²⁺ dynamics after 2-APB treatment in WT and KI cardiomyocytes 1 day after TAC. WT $n = 13$ and KI $n = 14$ cells (Ca²⁺ transient amplitude, two-sided Mann-Whitney *U* test); WT $n = 9$ and KI $n = 10$ cells (SR Ca²⁺ content and fractional Ca²⁺ release, two-sided unpaired *t* test); WT and KI $n = 13$ cells ($T_{50}$, two-sided Mann-Whitney *U* test) from 3 hearts/group. ****$p < 0.0001$, ***$p < 0.001$, **$p < 0.01$, *$p < 0.05$, ns not significant. Adjusted *p* values for (**a**): 0.0216 (Ca²⁺ transient amplitude, WT, Baseline vs TAC), 0.0435 (Ca²⁺ transient amplitude, KI, Baseline vs TAC), 0.0071 (SR Ca²⁺ content, WT, Baseline vs TAC), 0.0007 (SR Ca²⁺ content, TAC, WT vs KI), 0.0423 (Fractional Ca²⁺ release, TAC, WT vs KI), 0.0001 ($T_{50}$, TAC, WT vs KI). Adjusted *p* values for (**c**): 0.0353 (KI, Pre-2APB vs Post-2APB). Adjusted *p* values for (**d**): 0.0373 ($T_{50}$, WT vs KI). Data are mean ± SEM. Source data are provided as a Source Data file.

Cardiac-specific overexpression of MEK1 and activation of ERK1/2 promote physiological hypertrophy, which is reversed by knockdown of ERK1/2[8,35], indicating the crucial role of the MEK1-ERK1/2 signaling pathway in adaptive hypertrophy[36]. The results presented here indicate that MEK1-ERK1/2 controls cardiac hypertrophy in part through regulation of Bcl-xL phosphorylation. Although ERK1/2 are proline-directed kinases, there is no proline residue near Ser14 of Bcl-xL. However, ERK1/2 can also phosphorylate a serine/threonine residue that does not immediately precede a proline residue[37,38]. Furthermore, a MEK inhibitor had no additive detrimental effect in S14A

knock-in mice in response to TAC in vivo, consistent with the notion that Bcl-xL Ser14 is directly phosphorylated by ERK1/2 in vivo. However, the possibility that the MEK1-ERK1/2 pathway indirectly phosphorylates Bcl-xL at Ser14 through other serine/threonine kinases cannot be formally excluded. Futher investigation is required to clarify this issue.

We have shown previously that Ser14 phosphorylation of Bcl-xL occurs during I/R through a K-Ras-Mst1-dependent mechanism, thereby stimulating apoptosis by promoting dissociation of Bcl-xL from Bax on the outer mitochondrial membrane[3,4]. Here, we show

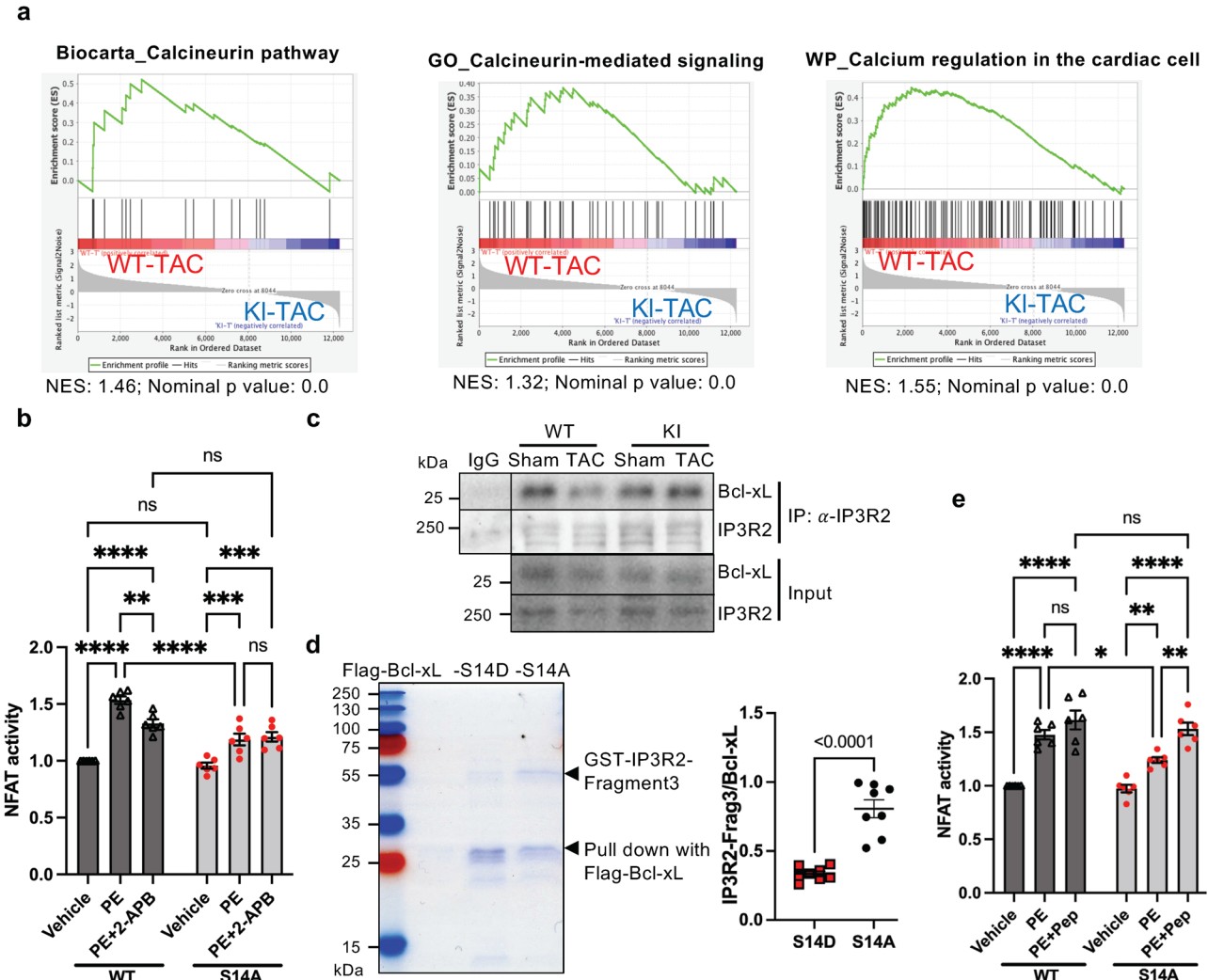

**Fig. 5 | Bcl-xL Serine (Ser) 14 phosphorylation disrupts its inhibitory interaction with IP3R, enhancing hypertrophic stimuli-induced calcineurin-NFAT signaling.** **a** Gene set enrichment analysis plots of calcineurin and calcium regulation pathways enriched in WT compared to knock-in (KI) mouse hearts after 9 hs of TAC. NES, normalized enrichment score. GSEA nominal *p* value is the statistical significance of the enrichment score by using a phenotype-based permutation test with no adjustment. **b** The relative NFAT transcriptional activity in response to phenylephrine (PE) with or without 2-APB in cardiomyocytes transfected with adenovirus harboring Bcl-xL-WT or -S14A mutant (*n* = 6 independently prepared cardiomyocyte preparations/cultures). **c** Immunoprecipitation assay with anti-IP3R type 2 antibody using WT and KI mouse hearts 1 day after TAC. Repeated three times. **d** Flag pull-down assay using Flag-Bcl-xL-S14D or -S14A and recombinant

GST-IP3R-fragment3, followed by Coomassie Brilliant Blue staining. Right panel shows quantification analysis of immunoprecipitated proteins (GST-IP3R-fragment 3 versus Flag-Bcl-xL-S14D or -S14A) (*n* = 8 independently prepared cardiomyocyte preparations/cultures, followed by in vitro binding assay). **e** The relative NFAT transcriptional activity in response to PE with or without synthetic peptide corresponding to the amino acid sequence in IP3R Fragment 3 (*n* = 6, independently prepared cardiomyocyte preparations/cultures). Two-way ANOVA with Tukey's multiple comparison test. ****$p < 0.0001$, ***$p < 0.001$, **$p < 0.01$, *$p < 0.05$, ns not significant. Adjusted *p* values for (**b**): 0.0041 (WT, PE vs PE + 2-APB), 0.001 (S14A, Vehicle vs PE), 0.0003 (S14A, Vehicle vs PE + 2-APB). Adjusted *p* values for (e): 0.0251 (PE, WT vs S14A), 0.0096 (S14A, Vehicle vs PE), 0.0037 (S14A, PE vs PE+Pep). Data are mean ± SEM. Source data are provided as a Source Data file.

that Ser14 phosphorylation of Bcl-xL during the acute phase of pressure overload occurs through an H-Ras-MEK-dependent mechanism, presumably in the ER/SR, and plays a salutary role by promoting compensatory cardiac hypertrophy. Interestingly, the level of apoptosis after pressure overload was similar between Bcl-xL-S14A knock-in and WT mice. Since both depressed cardiac function and increased LV wall stress could have induced higher levels of apoptosis, the fact that apoptosis is not increased in S14A knock-in mice suggests that a mechanism suppressing apoptosis may still be operative in these mice. Even so, it is puzzling to observe that Bcl-xL-S14A mice exhibited a more *detrimental* cardiac phenotype during acute pressure overload. One possible explanation for the discrepancy could be that regulation of the signaling complexes in which Bcl-xL takes part, its interacting partners, and its subcellular localization are distinct between pressure overload and I/R. For

example, I/R induces K-Ras-induced activation of Mst1 in mitochondria, which induces phosphorylation of Bcl-xL Ser14 and its dissociation from Bax at the mitochondrial outer membrane[3]. We speculate that Bcl-xL Ser14 phosphorylation by the H-Ras-MEK pathway occurs at a distinct subcellular location during the acute phase of pressure overload, namely at Bcl-xL bound to IP3R in the SR. Consistent with this hypothesis, Bcl-xL physically interacts with H-Ras (Fig. 3h). Furthermore, H-Ras and K-Ras induce distinct phenotypes in the heart in response to pressure overload[39]. Our results also suggest that activation of the signaling mechanism mediating compensatory hypertrophy is important during the early phase of pressure overload, even if it might promote apoptosis. Whether Bcl-xL Ser14 phosphorylation is regulated during the chronic phase of heart failure and, if so, where in the cell it is regulated and how it affects cell death remain to be clarified.

We have shown previously that overexpression of Bcl-xL(S14A) inhibits dissociation of Bcl-xL from the BH3 domain of Bax during myocardial reperfusion, thereby inhibiting apoptosis[4]. Thus, by inference, Bcl-xL (S14A) may exhibit increased binding to the BH3 domian of Beclin 1, thereby inhibiting autophagy. S14A knock-in mice showed no overt change in the levels of Beclin-1, p62, or LC3-I/II, indicating no or minimal effect of Bcl-xL S14 phosphorylation on autophagy activity in the heart at baseline. However, further investigation is required to test whether Bcl-xL Ser14 phosphorylation decreases the binding of Bcl-xL to Beclin 1, thereby stimulating autophagy in stress condition, such as pressure overload.

It has been shown that the combined use of a MEK inhibitor and ABT-263, a chemical inhibitor of Bcl-xL, promotes tumor regression in KRAS mutant cancer models[40]. MEK inhibitors alone may promote the anti-apoptotic actions of Bcl-xL by inhibiting Ser14 phosphorylation and stimulating Bcl-xL-Bax interaction, thereby diminishing their killing effects. We speculate that concomitant use of ABT-263 would enhance the anti-cancer effect of the MEK inhibitors by eliminating the anti-apoptotic actions of Bcl-xL.

There are some limitations to our study. We used rat neonatal ventricular cardiomyocytes in some in vitro experiments to evaluate the upstream kinases that phosphorylate Bcl-xL at Ser14 and determine the effect of Bcl-xL Ser14 phosphorylation upon NFAT activity. Since cultured neonatal cardiomyocytes may not fully recapitulate the stress response that occurs in adult hearts at baseline and in response to pressure overload, futher investigation is needed to elucidate the molecular mechanisms through which pressure overload leads to Bcl-xL Ser14 phosphorylation, and whether Bcl-xL Ser14 phosphorylation, in turn, regulates NFAT activity in vivo. Second, mitochondrial uptake of calcium released from IP3Rs is a critical determinant of cell survival, in part through regulation of apoptosis and autophagy. Thus, how the regulation of IP3R calcium release by Bcl-xL Ser14 phosphorylation affects the ER/SR-mitochondrial connection remains to be clarified. Finally, although we show that Bcl-xL Ser14 phosphorylation plays a salutary role during the acute phase of pressure overload through regulation of IP3R-mediated compensatory hypertrophy and increased contractility, considering the subcellular localization of Bcl-xL in mitochondria and the SR, contributions of additional mechanisms, including $Ca^{2+}$ handling mechanisms and mitochondrial mechanisms, to the failing cardiac phenotype in Bxl-xL(S14A) mice cannot be excluded.

In summary, the current study demonstrates that Bcl-xL Ser14 phosphorylation is essential for adaptive hypertrophy to prevent acute decompensated heart failure, in part through IP3R-mediated $Ca^{2+}$ release and calcineurin-NFAT signaling, during acute pressure overload.

## Statistics

All values are expressed as mean ± SEM. Statistical analyses were carried out by 2-tailed unpaired Student $t$ test for 2 groups or one- or two-way ANOVA followed by the Tukey post-hoc analysis for 3 groups or more unless otherwise stated. If the data distribution failed normality by the Shapiro−Wilk test or Kolmogorov−Smirnov test, the Mann-Whitney $U$ test for 2 groups was performed. The statistical analyses used for each figure are indicated in the corresponding figure legends. Survival curves were plotted by the Kaplan−Meier method, with statistical significance analyzed by log-rank test. Statistical analyses were conducted using Prism 9 (GraphPad Software). All experiments are represented by multiple biological replicates or independent experiments. The number of replicates per experiment are indicated in the legends. All experiments were conducted using at least two independent experimental materials or cohorts to reproduce similar results. No sample was excluded from analysis. A $p$ value of less than 0.05 was considered significant.

## Methods

### Mice

The Bcl-xL S14A knock-in mice (C57BL/6 J background) were generated as previously described[4]. Briefly, Bcl-xL genomic DNA was isolated from the BAC clone (RP23-106A3) to construct the Bcl-xL S14A knock-in targeting vector. PCR-based site-directed mutagenesis was performed to introduce a single mutation of T to G in codon 14 in exon 2 of Bcl-xL to change codon 14 from Serine to Alanine. The targeting vector was linearized with *PmeI* and subsequently electroporated into R1 ES cells. G418-resistant ES clones were screened for homologous recombination by *MfeI* digestion, followed by Southern blot analysis. Three independent homologous recombinant ES clones were microinjected into blastocytes from C57BL/6 J mice and transferred into pseudo-pregnant recipients to generate male chimeras. The chimeric male mice resulting from the microinjection were bred with C57BL/6 J female mice to generate germline-transmitted heterozygous S14A knock-in mice. The pGK neo cassette was deleted by crossing het knock-in mice with CMV-Cre mice (Jackson Labs, Strain # 006054). The mutant offspring were backcrossed into the C57BL/6J background for more than 8 generations. Male C57BL/6 J wild-type mice were purchased from Jackson Labs (Strain # 000664) at 5−8 weeks of age. The ERK inhibitor, PD0325901 (15 mg/kg/day), or vehicle (DMSO) was administered orally once a day for 3 days before surgery and 2 days after surgery. Mice were housed in a temperature and humidity-controlled environment within a range of 21−23 °C with 12-h light/dark cycles, in which they received food and water *ad libitum*. We used age-matched male mice in all animal experiments. The sample size required was estimated to be $n = 5-8$ per group according to the Power analysis based upon previous studies examining the effects of pressure overload on cardiac hypertrophy and hypertrophic signaling. All protocols concerning the use of animals were approved by the Institutional Animal Care and Use Committee at Rutgers New Jersey Medical School and all procedures conformed to NIH guidelines (Guide for the Care and Use of Laboratory Animals). Handling of mice and euthanasia with $CO_2$ in an appropriate chamber were conducted in accordance with guidelines on euthanasia of the American Veterinary Medical Association. Rutgers is accredited by AAALAC International, in compliance with Animal Welfare Act regulations and Public Health Service (PHS) Policy on Humane Care and Use of Laboratory Animals, and has a PHS Approved Animal Welfare Assurance with the NIH Office of Laboratory Animal Welfare (D16-00098 (A3158-01)).

### Cell line

HEK293 cells were purchased from the American Type Culture Collection (CRL-1573) and maintained at 37 °C with 5% $CO_2$ in Dulbecco's modified Eagle's Medium with 10% fetal bovine serum.

### Primary rat neonatal cardiomyocytes

Primary cultures of ventricular cardiomyocytes were prepared from 1-day-old Crl:(WI)BR-Wistar rats (Envigo, Somerville) and maintained in culture[12]. The neonatal rats of both sexes were deeply anesthetized with isoflurane. The chest was opened, and the heart was harvested. A cardiomyocyte-rich fraction was obtained by centrifugation through a discontinuous Percoll gradient. Cardiomyocytes were cultured in complete medium containing Dulbecco's modified Eagle's medium/F-12 supplemented with 5% horse serum, 4 μg/ml transferrin, 0.7 ng/ml sodium selenite, 2 g/l bovine serum albumin (fraction V), 3 mM pyruvate, 15 mM Hepes pH 7.1, 100 μM ascorbate, 100 mg/l ampicillin, 5 mg/l linoleic acid, and 100 μM 5-bromo-2′-deoxyuridine (Sigma). Culture dishes were coated with 0.3% gelatin or 2% gelatin for immunofluorescence staining on chamber slides.

### Isolation of adult cardiomyocytes for signaling experiments

Adult mouse cardiomyocytes were isolated as described previously with a modification[12,41]. Briefly, the heart of a male mouse was

perfused with 12 ml EDTA buffer [130 mM NaCl, 5 mM KCl, 0.5 mM NaH$_2$PO$_4$, 10 mM HEPES, 10 mM Glucose, 10 mM BDM, 10 mM Taurine, 5 mM EDTA] to stop the beating of the heart. Digestion was achieved using 30 ml perfusion buffer [130 mM NaCl, 5 mM KCl, 0.5 mM NaH$_2$PO$_4$, 10 mM HEPES, 10 mM Glucose, 10 mM BDM, 10 mM Taurine, 1 mM MgCl$_2$] containing Collagenase type II (0.5 mg/ml) and Protease XIV (0.05 mg/ml). Cellular dissociation was stopped by addition of 5 ml Perfusion buffer containing 5% FBS and 100 mM BSA-conjugated fatty acid cocktail (palmitic acid: oleic acid: linoleic acid = 2:1:1 and BSA: fatty acid = 1:5). Cardiomyocytes and non-cardiomyocytes were separated by 4 sequential rounds of gravity settling with calcium reintroduction medium containing 100 mM BSA-conjugated fatty acid cocktail.

## Transverse aortic constriction

Male 8–10-week-old animals were subjected to TAC or sham surgery. Mice were anesthetized with pentobarbital (60–70 mg/kg, intraperitoneal injection) and mechanically ventilated with a tidal volume of 0.2 ml and a respiratory rate of 110 breaths per minute. The mice were kept warm with heat lamps. It took around 5 min to establish full anesthesia. The chest and neck were shaved by clipper and the skin was cleaned using betadine and 70% ethanol 3 times. Sterile ophthalmic ointment was applied to the eyes. Mice were placed in a supine position. A lack of toe pinch/tail pinch reflex was checked before making the incision. Before making the surgical incision, we subcutaneously injected a very small volume of bupivacaine along the incision line. A midline cervical incision (15-20 mm) was made to assist intubation of the trachea and for access to the intercostal space. The left chest was opened at the second intercostal space. The intercostal incision was <0.5 cm and opened by self-designed stretchers made of 25-gauge needles connected to rubber bands and fixed on the surgical board by pins. With the aid of a dissecting microscope, aortic constriction was performed by ligating the transverse thoracic aorta between the innominate artery and left common carotid artery with a 28-gauge needle using a 7-0 prolene suture, and then removing the needle. Sham operation was performed without constricting the aorta. During surgery, the depth of anesthesia was monitored periodically by checking pedal reflex. Thoracotomy incision, overlying muscle layers and skin were closed in layers using 5-0 prolene sutures, and the pneumothorax was reduced. The TAC procedure was completed within 20–30 min per mouse. The mice were then treated with Buprenex-SR (1.0–1.2 mg/kg, SC) and monitored while being allowed to recover in a warm incubator. When recovered from anesthesia 1–2 h after the closure of the chest, the mice were extubated and returned to their cages. Upon completion of all experimental procedures, mice were euthanized by cervical dislocation followed by harvest of the hearts for biochemical studies, including signaling pathways.

## Echocardiography

Mice were anesthetized using 12 μl/g body weight of 2.5% avertin (Sigma-Aldrich), and echocardiography was performed using ultrasound (Vivid 7, GE Healthcare). It took around 10-20 min from the establishment of anesthesia to the completion of echocardiography and 1–2 h to fully recover from anesthesia after echocardiography. A 13-MHz linear ultrasound transducer was used. Mice were subjected to 2-dimension guided M-mode measurements of LV internal diameter at the papillary muscle level from the short-axis view to measure systolic function and wall thickness, which were taken from at least three beats and averaged. LV ejection fraction was calculated as follows: Ejection fraction = [(LVEDD)$^3$−(LVESD)$^3$]/(LVEDD)$^3$ × 100. End-diastolic wall stress was calculated using echocardiographic and hemodynamic parameters as follows: End-diastolic wall stress = LV end-diastolic pressure × LVDd/2 × LVPWd x (1 + LVPWd/2 × LVDd)[42].

## Hemodynamic measurements

Mice were anesthetized with Avertin (300 mg/kg, intraperitoneal injection) to measure arterial pressure gradients and LV end-diastolic pressures. The chest and neck were shaved by clipper and the skin was cleaned using betadine and 70% isopropyl alcohol three times. Mice were then placed in a supine position. The lack of pedal reflex was confirmed prior to making an incision. A small incision (5-10 mm) was made on the neck. Under a dissecting microscope, the common carotid artery was surgically isolated and clamped proximally and distally. A small incision (0.5-1 mm) was made in the carotid artery, and a high-fidelity micromanometer catheter (1.4 French; Millar Instruments Inc.) was inserted into the artery and advanced into the aorta to measure aortic pressure proximal to the constriction site and then into the LV cavity to measure LV pressure and its first derivatives. A separate high-fidelity micromanometer catheter was inserted via the femoral artery and advanced into the aorta to measure aortic pressure distal to the constriction simultaneously. During the procedure, the depth of anesthesia was monitored by checking pedal reflex periodically to ensure the surgical plane of anesthesia.

## Immunoblotting

Cardiomyocyte lysates and heart homogenates were prepared in RIPA buffer containing protease and phosphatase inhibitors (Sigma-Aldrich)[4,43]. Lysates were centrifuged at 16,100 x g at 4 °C for 15 min. Protein concentrations were determined using a standard BCA assay. Total protein lysates (10-30 μg) were incubated with SDS sample buffer (final concentration: 100 mM Tris (pH 6.8), 2% SDS, 5% glycerol, 2.5% 2-mercaptoethanol, and 0.05% bromophenol blue) at 95 °C for 5 min. The denatured protein samples were separated by SDS-PAGE, transferred to polyvinylidene difluoride membranes by wet electrotransfer, blocked in either 5% (w/v) BSA or 5% (w/v) non-fat dry milk in 1xTBS/0.5% Tween 20 at room temperature for 1 h, and probed with primary antibodies at 4 °C overnight. After washing with 1xTBS/0.5% Tween 20 for 20 min, the membranes were incubated with the corresponding secondary antibody at room temperature for 1 h. After washing with 1xTBS/0.5% Tween 20 for 45 min, the membranes were developed with ECL Western blotting substrate, followed by acquisition of digital images with the ChemiDoc MP Imaging System (Bio-Rad). The intensities of Western blot bands were quantified using ImageJ software. Uncropped blotting images with molecular markers are provided in Supplementary Fig. 5.

## Antibodies and reagents

The following commercial antibodies were used at the indicated dilutions: rabbit monoclonal Bcl-xL antibody (#2764) (1:6000), rabbit cleaved caspase-3 antibody (#9661) (1:2000), rabbit cleaved caspase-9 antibody (#9507) (1:2000), rabbit monoclonal p44/42 MAPK (Erk1/2) antibody (#9102) (1:5000), rabbit monoclonal phospho-p44/42 MAPK (Erk1/2) (Thr202/Tyr204) antibody (#4370) (1:5000), rabbit polyclonal phospho-GSK-3α/β (Ser21/9) antibody (#9331) (1:3000), rabbit monoclonal GSK-3α/β antibody (#5676) (1:5000), rabbit polyclonal phospho-Akt (Ser473) antibody (#9271) (1:4000), rabbit polyclonal Akt antibody (#9272) (1:8000), rabbit monoclonal phospho-MST1 (Thr183)/MST2 (Thr180) antibody (#49332) (1:1000), rabbit monoclonal NFAT3 antibody (#2183) (1:1,000), rabbit monoclonal GAPDH antibody (#5174) (1:8,000), rabbit monoclonal Histone H3 antibody (#4499) (1:10,000), anti-mouse or -rabbit IgG, HRP-linked antibodies (#7076 and #7074) (1:5,000) (Cell Signaling Technology); α-actinin (sarcomeric) antibody (#A7811) (1:4000), rabbit monoclonal α-tubulin antibody (T6199) (1:8,000) (Sigma-Aldrich); mouse monoclonal IP3R-II antibody (Santa Cruz Biotechnology #sc-398434) (1:1,000); rabbit polyclonal H-Ras antibody (C-20) (Santa Cruz Biotechnology #sc-520) (1:1000); and mouse monoclonal MST1 antibody (BD Transduction Laboratories #611052) (1:4,000). For detection of phosphorylation of Bcl-xL at Ser14, a polyclonal phosphorylation-specific antibody was

generated by immunizing rabbits with a phospho-peptide FL{pSER}YKLSQKGYSWSC by GenScript (dilution, 1:1,000)[3]. Antibodies were diluted in either 5% (w/v) BSA or 5% (w/v) non-fat dry milk in 1xTBS/0.5% Tween 20, depending on the level of background intensity. The following reagents were used: MEK inhibitor (PD0325901) and Akt inhibitor (Triciribine) (Sigma-Aldrich).

## Subcellular fractionation
Cultured neonatal rat cardiomyocytes were washed with PBS and collected with ice-cold PBS, followed by centrifugation at 600 x *g* for 5 min. Cardiomyocytes were then resuspended in hypotonic lysis buffer (10 mM K-HEPES pH 7.9, 1.5 mM MgCl$_2$, 10 mM KCl, 0.1 mM EGTA, 0.1 mM EDTA, 1% IGEPAL, 1% Phosphatase Inhibitor Cocktail, and 1% Protease Inhibitor Cocktail) and were incubated for 15 min on ice with intermittent pipetting. Whole-cell lysates were centrifuged at 1200×*g* for 5 min. The supernatant was collected for the cytosolic fraction, and the pellets were resuspended in lysis buffer (20 mM K-HEPES, 25% Glycerol, 0.45 M NaCl, 1.5 mM MgCl$_2$, 1 mM EGTA, 1 mM EDTA, 1% Phosphatase Inhibitor Cocktail, and 1% Protease Inhibitor Cocktail) and were incubated for 15 min on ice with intermittent pipetting. The total homogenate was centrifuged at 16,100 x g for 10 min to collect the nuclear fraction. The pelleted nuclei were resuspended in lysis buffer and protein content was determined for all fractions.

## Immunoprecipitation
Heart samples were lysed with lysis buffer containing 50 mM Tris-HCl pH 7.4, 150 mM NaCl, 1% Triton-X 100, 1% Sodium Deoxycholate, Protease Inhibitor Cocktail (Sigma), and Phosphatase Inhibitor Cocktail (Sigma). Primary antibody was covalently immobilized on protein A/G agarose using the Pierce Crosslink Immunoprecipitation Kit according to the manufacturer's instructions (Thermo Scientific). Samples were incubated with immobilized antibody beads overnight at 4 °C. After immunoprecipitation, the samples were washed with lysis buffer five times. They were then resuspended with lysis buffer and the immunoprecipitates were subjected to immunoblotting using specific primary antibodies and a conformation-specific secondary antibody (Clean-Blot IP Detection Reagent (ThermoFisher Scientific)).

## FLAG Pull-down assay
Cardiomyocytes were transduced with adenovirus harboring FLAG-Bcl-xL or LacZ for 2 days, followed by treatment with PE or vehicle for 20 min. The cardiomyocytes were collected with RIPA buffer containing protease and phosphatase inhibitors (Sigma-Aldrich) as described previously[4]. Lysates were centrifuged at 16,100×*g* at 4 °C for 15 min. After protein concentrations were determined using a standard BCA assay, protein lysates were incubated with anti-FLAG M2 Magnetic Beads (Sigma-Aldrich) overnight at 4 °C. After FLAG pull-down, the samples were washed with lysis buffer five times. They were then resuspended with lysis buffer and the immunoprecipitates were subjected to immunoblotting using specific primary and secondary antibodies.

## Adenovirus constructs
Recombinant adenovirus vector for overexpression was constructed, propagated and titered as previously described[3,4,12,43]. Briefly, pBHGloxΔE1,3Cre (Microbix), including the ΔE adenoviral genome, was co-transfected with the pDC shuttle vector containing the gene of interest into 293 cells. Replication-defective human adenovirus type 5 (devoid of E1) harboring full-length wild type or mutant Bcl-xL cDNA (Ad-Bcl-xL) or H-Ras cDNA (Ad-H-Ras) was generated by homologous recombination in 293 cells. Adenovirus harboring beta-galactosidase (Ad-LacZ) was used as a control.

## Recombinant proteins
The bacterial expression vectors for GST-fused Bcl-xL-full-length-wild type (WT) and -mutant (S14A) and IP3R fragment 3 were generated by insertion of human Bcl-xL and mouse IP3R fragment 3 cDNA amplified by PCR into the pCold-GST-vector. The BL21 E. coli strain was transformed with the expression vectors, grown in 3 ml LB medium containing ampicillin overnight at 37 °C, and then transferred to 250 ml LB medium containing ampicillin. Protein expression was induced by addition of 1 mM isopropylthio-b-galactoside. After overnight culture at 15 °C, the E. coli were lysed in lysis buffer (1% Triton X-100 and 1 mM DTT in PBS) with sonication. The lysate was incubated with 0.5 ml Glutathione-sepharose 4B (GE Healthcare) for 1 h at 4 °C. The sepharose was washed 3 times with 5 ml lysis buffer, and then suspended with 1 ml cleavage buffer (20 mM Tris pH 7, 150 mM NaCl, 1 mM DTT). A membrane-permeable synthetic peptide corresponding to the IP3R fragment 3 was generated by GeneScript as previously described with modification[23] using the following amino acid sequence (RKKRRQRRRGKNVYTEIKCNSLLPLDDIVRV).

## In vitro kinase assay
Recombinant active ERK1 was purchased from Millipore Sigma. Recombinant GST-tagged full-length Bcl-xL protein was generated using the pCold-GST-vector. Recombinant active ERK1 (10 ng, Millipore Sigma #14-439) was incubated with recombinant GST-Bcl-xL-WT (1 mg) in a kinase buffer (50 mM HEPES (pH 7.4), 15 mM MgCl$_2$ and 200 mM sodium vanadate) in the presence or absence of 100 mM ATP at 30 °C for 15 min. Recombinant phosphorylated GST-Bcl-xL protein was separated by SDS-PAGE, followed by immunoblots with anti-Bcl-xL Ser14 phospho-specific antibody or staining with Coomassie Brilliant Blue and LC-MS/MS analysis.

## Mass spectrometry sample preparation and analysis
A kinase reaction was performed using recombinant GST-tagged human Bcl-xL and recombinant active ERK1 protein. Phosphorylated protein was separated by SDS-PAGE and stained with Coomassie Brilliant Blue. The gel band of interest was excised for in-gel trypsin digestion as described[12,44]. Briefly, the gel bands of each sample were excised into ~1 cm$^3$ pieces, and washed four times with 1 μL each of a solution of 30% acetonitrile (ACN) and 70% of 100 mM NH$_4$HCO$_3$. Subsequently, 200 μL of 25 mM dithiothreitol (DTT) solution was used to reduce disulfides at 55 °C for 30 min, and 200 μL of 50 mM iodoacetamide solution was then added to alkylate thiols at room temperature in the dark for 30 min. The resulting gel peices were dehydrated with 200 μL of ACN to remove both DTT and iodoacetamide. For in-gel trypsin digestion, 100 μL of trypsin solution (5 μg/mL in 50 mM NH$_4$HCO$_3$) was added into each sample, and incubated at 37 °C overnight. Resulting peptides were extracted, desalted with Pierce C$_{18}$ spin columns (Thermo Scientific), based on the manufacturer's protocol, followed by Speed Vac prior to LC-MS/MS analysis on a Thermo Orbitrap Fusion Lumos MS instrument.

Peptides from each sample were reconstituted in Solvent A (consisting of 2% ACN in 0.1% formic acid, FA). Two microliters of peptides from each sample were subjected to LC-MS/MS analysis using an Orbitrap Fusion Lumos Mass Spectrometer coupled with an UltiMate 3000 UHPLC$^+$ system (Thermo Scientific). The separation of peptides occurred on an Acclaim PepMap C$_{18}$ column (75 μm × 50 cm, 2 μm, 100 Å) with a 2-h binary gradient ranging from 2% to 95% of Solvent B (85% ACN in 0.1% FA), at a flow rate of 300 nL/min. Eluted peptides were introduced into the MS system via a Nanospray Flex ion source with a spray voltage of 2 kV and a capillary temperature of 275 °C. MS spectra were acquired in positive ion mode using Xcalibur (1.5). Full MS scans were obtained in an m/z range of 375 to 1500 in profile mode, with an AGC value of approximately 1E6. Subsequent to each full MS scan, the data-dependent MS/MS mode was used to analyze the ions with charge states ranging from 2$^+$ to 7$^+$. The isolation window of 2 m/z was used for MS/MS analysis and the higher energy collision dissociation (HCD) was used for fragmentation with a normalized collision energy of 30%. The AGC for MS/MS

analysis was set to 5E4 and a dynamic exclusion time of 45 s was implemented.

The MS/MS spectra were searched against a Uniprot human database using the Sequest search engine on the Proteome Discoverer platform (PD V2.4.1.15). Trypsin was set as the enzyme with two miss cleavages. Methionine oxidation, STY phosphorylation were selected as variable modifications and C carbamidomethylation was set as a fixed modification. Peptide mass tolerance was set to 10 ppm, and the MS/MS mass tolerance was set to 0.1 Da. The false discovery rate of protein identification is less than 1%, using Percolator embedded in PD 2.4.1.15. To estimate phosphorylation site localization probability, ptmRS (2.2) embedded in PD2.4.1.15 was used. Only phosphorylation sites with probabilities of 90% or more were considered as confidently mapped.

## In vitro binding assays

Flag-Bcl-xL-S14D or -S14A protein was overexpressed using an adenovirus overexpression system in rat neonatal cardiomyocytes. Cardiomyocyte lysates were collected and Flag-tagged proteins were immunoprecipitated using Flag-agarose beads (Sigma Aldrich). Immunoprecipitated Flag-Bcl-xL-S14D or -S14A proteins were incubated with recombinant GST-fused IP3R fragment 3 in lysis buffer containing 50 mM Tris-HCl pH 7.4, 150 mM NaCl, 1% Triton-X 100, 1% Sodium Deoxycholate, Protease Inhibitor Cocktail (Sigma Aldrich), and Phosphatase Inhibitor Cocktail (Sigma Aldrich) with rotation for 1 h at 4 °C, followed by pull-down with Flag-agarose beads. After washing five times with lysis buffer, proteins were eluted with 5xSDS sample buffer, followed by SDS-PAGE and Coomassie Brilliant Blue staining.

## RNA-seq library preparation, sequencing, and data analysis

Total RNA was isolated from mouse hearts using TRIzol (Invitrogen). Isolated RNA was checked for integrity on an Agilent Bioanalyzer 2100; samples with RNA integrity number >7.0 were used for subsequent processing. Total RNA was subjected to two rounds of poly(A) selection using oligo-d(T)25 magnetic beads (New England Biolabs). A paired-end (strand specific) cDNA library was prepared using the NEB Next Ultra-directional RNA-seq protocol. Briefly, poly(A) + RNA was fragmented by heating at 94 °C for 10 min, followed by reverse transcription and second strand cDNA synthesis using the reagents provided in the NEB Next kit. End-repaired cDNA was then ligated with double stranded DNA adapters, followed by purification of ligated DNA with AmpureXP beads. cDNA was then amplified by PCR for 15 cycles with a universal forward primer and a reverse primer with bar code. The sequencing of the cDNA libraries was performed on the Illumina NextSeq 500 platform (Illumina, San Diego, CA) using the single-read 1 × 75 cycles configuration. The raw reads files have been deposited in the NCBI Gene Expression Omnibus.

Raw reads were quality trimmed using Trimmomatic-0.39 with leading and trailing Q score 20, minimum length 35 bp. The cleaned reads were mapped to *Mus musculus* genome GRCm38 using HISAT2 (Version 2.2.1). The reference genome sequence and annotation files were downloaded from ENSEMBL release 97 (Mus_musculus.GRCm38.101fa and Mus_musculus.GRCm38.101.gtf). The aligned read counts were obtained using htseq-count as part of the package HTSeq-0.6.1. The Bioconductor package edgeR (Version 3.18.1 with limma 3.32.10) was used to perform the differential gene expression analysis under R environment, R version 4.1.1. Expression patterns of regulated genes were graphically represented in a heat map. Hierarchical clustering was performed to group genes with similar features in the expression profile. Normalized expression data were also analyzed with GSEA version 4.1.0 software using the JAVA program (Broad Institute, Cambridge, MA). All gene sets were obtained from the Molecular Signatures Database version 7.3 distributed on the GSEA Web site.

## Quantitative RT-PCR

Total RNA was prepared from mouse hearts using TRIzol (Invitrogen)[43]. cDNA was generated using 300 ng total RNA and SuperScript III Reverse Transcriptase (ThermoFisher). Using Maxima SYBR Green qPCR master mix (Fermentas), real-time RT-PCR was performed under the following conditions: 94 °C for 10 min; 40 cycles of 94 °C for 15 s, 58 °C for 30 s, 72 °C for 30 s; and a final elongation at 72 °C for 15 min. Relative mRNA expression was determined by the ΔΔ-Ct method normalized to the ribosomal RNA (18S) level. The following oligonucleotide primers were used: *NPPA*, sense 5′-ATGGGCTCCTTCTCCATCAC-3′ and antisense 5′-ATCTTCGGTACCGGAAGCTG-3′; *NPPB*, sense 5′-AAGTCCTAGCCAGTC TCCAGA-3′ and antisense 5′-GAGCTGTCTCTGGGCCATTTC-3′; *MYH7*, sense 5′-GCCAACACCAACCTGTCCAAGTTC-3′ and antisense 5′- TGCAA AGGCTCCAGGTCTGAGGGC-3′; *Rcan1.4*, sense 5′-TCCAGCTTGGGCTTG ACTGAG-3′ and antisense 5′- ACTGGAAGGTGGTGTCCTTGT-3′; *18 S* rRNA, sense 5′-CGCGGTTCTATTTTGTTGGT-3′ and antisense 5′-AGTC GGCATCGTTTATGGTC-3′.

## Immunohistochemistry

The heart tissue was washed with PBS, fixed in 4% paraformaldehyde overnight, embedded in paraffin, and sectioned at 10-μm thickness onto a glass slide. After de-paraffinization, sections were stained with wheat germ agglutinin (WGA) for evaluation of the cross-sectional area of cardiomyocytes, or TUNEL for evaluation of apoptosis. The outline of 100–200 myocytes was traced in each section, using ImageJ software (NIH). For co-staining with TUNEL and troponin T, the heart sections were first stained with TUNEL and washed with PBS, followed by incubation with cardiac troponin T (Invitrogen #MA5-12960) at 4 °C overnight. After washing with PBS, the heart sections were incubated with Alexa 568 anti-mouse antibody at room temperature for 1 h to visualize cardiac troponin T. The heart sections were mounted with VECTASHIELD Mounting Media with DAPI. The tissues were observed under a fluorescence microscope (Eclipse T*i*, Nikon) with Nikon NIS-Elements imaging software.

## Reporter gene assay

NFAT reporter gene activity in rat neonatal cardiomyocytes was measured with a luciferase assay system (Promega). Cardiomyocytes were transfected with NFAT luciferase reporter plasmids (a gift from Dr. Toren Finkel, Addgene #10959) and Bcl-xL-WT, -S14A, or S14D in the pDC316 vector overnight (24-well plate) using LipofectAmine 2000 (Invitrogen). The NFAT reporter gene assay was performed after 4 h of phenylephrine treatment in the presence or absence of 2-APB (25 μM) or synthetic peptide (20 μM). Cardiomyocytes were lysed with 50 μl Reporter lysis buffer (24 wells each). The luminescence reaction was started by adding 5 μl lysate to 50 μl Reaction buffer, and luminescence was measured for 10 s using an OPTOCOMP I luminometer (MGM Instruments). The luminescence was normalized by protein content measured by protein assay kit (BioRad).

## Cardiomyocyte cell size measurement

Rat neonatal cardiomyocytes were cultured on coverslips, washed with PBS twice, fixed with 3.7% paraformaldehyde for 15 min, and washed with PBS three times. Samples were permeabilized with PBST (0.5% Triton-X in PBS) for 15 min, and blocked in 5% BSA, 5% goat serum in PBST for 30 min at 37 °C. Cardiomyocytes were stained with Alexa Fluor 555 phalloidin (Thermo Fisher Scientific, A34055). Samples were washed with PBS and mounted on glass slides with mounting medium (VECTASHIELD, Vector Laboratories). Cells were observed under a fluorescence microscope (Eclipse T*i*, Nikon) with Nikon NIS-Elements imaging software. Cell size was measured for 25–30 cells for each condition in each experiment and the mean value was taken as representative of the experiment. This experiment was performed independently five times (*n* = 5).

## Isolation of ventricular cardiomyocytes for cell shortening and Ca²⁺ transient experiments

Left ventricular myocytes were enzymatically isolated from hearts of Bcl-xL S14A knock-in and control WT male mice (2–3 months). Mice were deeply anesthetized with isoflurane in a covered beaker before hearts were removed and perfused retrogradely in Langendorff fashion at 37 °C with Ca²⁺-free Tyrode's solution containing 1.0 mg/ml collagenase type II (Worthington, Biochemical Corp., Lakewood, NJ, USA) and 0.1 mg/ml thermolysin (or protease type XIV, Sigma) for 10–12 min. After the enzyme solution was washed out, the hearts were transferred from the Langendorff apparatus to petri dishes. Left ventricles were gently teased apart with forceps. Ca²⁺ concentration was gradually increased to 1.0 mM. Finally, the cell suspension was filtered through a 200 μm nylon mesh. Cardiomyocytes were stored at room temperature and used within 8 hs after isolation. All electrophysiological experiments were performed at 35–37 °C.

## Mouse neonatal cardiac fibroblast isolation

Hearts were dissected from mice on postnatal day 1 to obtain neonatal fibroblasts. At this time, tails were also cut for genotyping. Neonatal cardiac fibroblasts were isolated by enzymatic digestion using enzyme digestion medium, containing 1.0 mg/ml collagenase type II (Worthington, Biochemical Corp., Lakewood, NJ, USA) in 25 ml of PBS and 0.25% (wt/vol) trypsin, at 37° on a rocker for 15 min and incubation with DMEM containing 20% FBS and penicillin-streptomycin in 6-well plates.

## Intracellular Ca2+ measurement

Ventricular myocytes were loaded with the Ca²⁺ indicator Fluo-4 AM (4 μm, Invitrogen, Grand Island, NY, USA) for 30 min. After washing and de-esterification (30 min), the myocytes were transferred to a heated chamber (37 °C) on a Nikon Eclipse TE200 inverted microscope (Nikon, Tokyo, Japan) with a Fluor x40 oil objective lens (numerical aperture 1.3). The intracellular Ca²⁺ fluorescence (excitation/emission wavelengths: 485/530 nm) was recorded with a spatial resolution of 500 × 400 pixels at 50 frames per second by an iXon Charge-Coupled Device (CCD) camera (Andor Technology, Concord, MA, USA) operated with Imaging Workbench software (INDEC BioSystems, Los Altos, CA, USA). The Ca²⁺ fluorescence intensity was expressed as the ratio $F/F_0$ (fluorescence (F) over the baseline diastolic fluorescence ($F_0$)).

## Measurement of single-cell shortening

Myocytes were placed in a heated chamber (37 °C) on a Nikon Eclipse TE200 inverted microscope and were subjected to 1-Hz field-pacing using a stimulator (Grass Instruments, West Warwick, Rhode Island, USA). Changes in cell length were monitored by a video-based edge detection system (model VED-105, Crescent Electronics, Sandy, UT, USA) with a 30-ms time resolution. Single-cell shortening was recorded and analyzed using commercially available pCLAMP 10 software (Molecular Devices, Sunnyvale, CA). The cell shortening was calculated as a percentage of shortening from the baseline cell length in the relaxed state.

## Reporting summary

Further information on research design is available in the Nature Portfolio Reporting Summary linked to this article.

## Data availability

All data generated or analyzed during this study are included in this published article and its Supplementary Information. RNA-sequencing data have been deposited at GEO and are publicly available as of the date of publication (Accession numbers: GSE199705). The mass spectrometry data have been deposited to the ProteomeXchange Consortium via the PRIDE partner repository with the accession code PXD045118. Source data are provided with this paper.

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

## Acknowledgements

We thank Daniela Zablocki for critical reading of the manuscript. This study was supported in part by U.S. Public Health Service grants HL155766 (M.N.) and HL67724, HL91469, HL102738, HL112330, HL138720, HL144626, HL150881, and AG23039 (J.S.). This work was also supported by an American Heart Association Scientist Development Grant (17SDG33660358) (M.N.) and Merit Award 20 Merit 35120374 (J.S.), and by the Fondation Leducq Transatlantic Network of Excellence 15CVD04 (J.S.). The mass spectrometry data were obtained using an Orbitrap mass spectrometer funded in part by NIH grants NS046593 and 1S10OD025047-01, for the support of proteomics research at the Rutgers Newark campus.

## Author contributions

M.N. and J.S. designed the experiments and wrote the paper; M.N. and M.A.K. conducted the in vitro and in vivo experiments; M.N. and P.Z. conducted the animal experiments and analyses; N.F. and L.H.X. conducted the Ca2+ transient experiments; T.L. and H.L. conducted the mass spectrometry analyses; M.N. conducted the gene expression analyses; Y.T. conducted immunohistochemistry analyses; D.D.R. provided the adenovirus harboring Bcl-xL mutants and H-Ras; S.I. provided technical support and suggestions regarding cell growth and death in the Hippo pathway; M.N. and J.S. generated project resources. All authors reviewed and commented on the manuscript.

## Competing interests

The authors declare no competing interests.
