## [Peer Review File · Nature Communications]

REVIEWER COMMENTS

Reviewer #1 (Remarks to the Author):

The authors addressed all of my major concerns and overall this is a strong manuscript. Some of the westerns are still underwhelming in my opinion. See my few comments below.

1. ED Fig 1c. Is there a reason why the Bcl-xL is so faint and why p-Bcl-xL is normalized to tubulin instead of total protein?
2. Any explanation why the WT mice subjected to TAC only have a reduced %EF of ~5% after 1 month.
3. Fig 3h examining the pulldown of Bcl-xL with H-Ras the IP blots are still not convincing, although a minor point
4. Figure 3d, it is slightly concerning there is no band for pMst1.
5. Page 4 the fourth line from the bottom reads 'In contrast to the protection proffered...' is this supposed to be offered?
6. Throughout the paper the +/- are in grey bars but the legend shows it as black.
7. Figure 5b and ED Fig 3e. Do the adenoviruses express to similar extents?

Reviewer #2 (Remarks to the Author):

The studies presented by Nakamura et al use a combination of in vitro and in vivo techniques to explore the role of BCL-xL S14 phosphorylation in the development of cardiac hypertrophy and failure. The strength of the study is the use of a knock-in mouse line that carries a S14A mutation removing the phosphorylation site. Overall the study is interesting and the phenotypes observed are strong. However some of the statistical analysis needs to be redone and the interpretation of the results is overreaching at times.

Major issues: The first issue is with the fact that this is a whole body knock in and the results need to be interpreted as such.

Fig 1 A (and the quantitation in Sup 1 A) shows baseline levels of BCL-xL phosphorylation. It is unclear of which timepoint this corresponds to. I assume from the supplemental figure this is correctly a sham and not a baseline sample, but especially when looking that close to surgery, the Sham sample from the corresponding time needs to be included. The authors assume the signal is coming from myocytes, however the sham surgery itself may result in a big inflammatory response, additionally it is unclear if

any circadian effects may be present. Finally the sham samples all have been normalized to one. This does not allow for the normal distribution of signal and exaggerates the statistical significance. Also the result section speaks of differences between the timepoints but no statistics are shown to support that.

Fig 1 B shows that over 50% of the animals have died within the seven day timepoint. This results in a survival bias for all the other results. Do they die within 1 day? 2 days? This is critical because the authors are trying to suggest that the loss of compensated hypertrophy is the cause of death. Hypertrophy takes time to develop and if the animals are dying that quickly something else may be contributing. Indeed there are a plethora of papers that show inhibition of hypertrophy after TAC is protective.

The S14 KI TAC hearts clearly hypertrophy, as evidenced by the HW changes, but it appears to be eccentric not concentric. Again, how quickly this develops is not clear.

The analysis in 2D-G is hampered by the fact that you are looking in the background of massive failure. Again earlier timepoints would be needed and perhaps other hypertrophy markers (BMHC RCAN1.4). Additionally it is not clear that the tunnel positive cells are myocytes, costaining is needed.

Fig 3A is good. 9 hours shows clear changes and differences between the groups. PE seems to maybe induce phosphorylation of Bcl-xl in a manner dependent of MEK, but without quantification it is not convincing. It is not clear why the authors didn't use this in vitro approach to look at hypertrophy signaling as well. This would help strengthen the interpretation of the in vivo results. I do see that extended figure 3 e-g uses transfection of gain of function mutants to address PE induced hypertrophy and NFAT3 localization.

- Figure 4 demonstrates that there is no increase in calcium transient in the KI line 1 day after TAC, but instead a decrease along with an increase of the half-life of decay. This is similar to what is observed in failing cells where Ca leak from the sarcoplasmic reticulum through the ryanodine receptor reduces the amplitude of the Ca transient and slows its rate of decay. However, no gain (or loss) in SR Ca content in the KI cells is observed. Inhibition with 2-ABP decreased the calcium transient amplitude in WT TAC myocytes but increased it in KI myocytes. These results are not adequately explained. Additionally, Molkenin has shown that inhibition of IP3R activation has little impact on TAC induced hypertrophy (Nakayama *Circ Res.* 2010 Sep 3;107(5):659-66.) How these results fit into that data is unclear. Likewise loss of ERK1/2 signaling does not result in a similar catastrophic phenotype after TAC (although it may support the concentric/eccentric hypertrophic changes. Kehat I et al . *Circ Res.* 2011 Jan 21;108(2):176-83) It would be important to look at mitochondrial phenotypes as they more often lead to the massive changes observed after TAC.

Overall there clearly is some interesting findings, ERK1 phosphorylation of BCL-xl is novel, as is the relationship between the IP3R and BCL-xl. I am not convinced of the relationship between the in vivo results and the in vitro results. Specifically the dramatic impact on cardiac contractility and mortality. Inhibition of IP3R alone is unlikely to account for the observations.

Reviewer #3 (Remarks to the Author):

All my concerns have been addressed, I have no further comment.

Point by point responses

Response to Reviewer #1:

The authors addressed all of my major concerns and overall this is a strong manuscript. Some of the westerns are still underwhelming in my opinion. See my few comments below.

Thank you very much for your comments and valuable suggestions. Below are our point-by-point responses to Reviewer 1's comments.

1. *ED Fig 1c. Is there a reason why the Bcl-xL is so faint and why p-Bcl-xL is normalized to tubulin instead of total protein?*

We have replaced the image of Bcl-xL with one with a stronger signal and normalized the pBcl-xL expression to that of Bcl-xL. We have now included the data in new Supplementary Fig. 1c.

2. *Any explanation why the WT mice subjected to TAG only have a reduced %EF of ~5% after 1 month.*

S14A knock-in (KI) mice developed more severe heart failure than wild type mice. Since their early mortality precluded thorough analyses of KI mice, we applied less severe aortic constriction in these experiments than in our ordinary TAC experiments. We confirmed that similar levels of aortic constriction were applied in wild type and S14A knock-in mice.

3. *Fig 3h examining the pull-down of Bcl-xL with H-Ras the IP blots are still not convincing, although a minor pointcm*

We have conducted another pull-down assay using cardiomyocytes transduced with adenovirus harboring FLAG-Bcl-xL in the presence or absence of phenylephrine. Phenylephrine treatment enhanced the interaction between H-Ras and Bcl-xL in cardiomyocytes *in vitro*. We have now included data on FLAG-pull down assays in new Fig. 3l.

4. *Figure 3d, it is slightly concerning there is no band for pMst1.*

We have repeated the Western blots with anti-phospho-Mst1 antibody and replaced the blots with ones with a stronger signal in Fig. 3d. We confirmed that the level of pMst1/Mst1 was not significantly affected by phenylephrine treatment.

5. *Page 4 the forth line from the bottom reads 'In contrast to the protection proffered...' is this suppose to be offered?*

We replaced the word with “conferred”.

6. *Throughout the paper the +/- are in grey bars but the legend shows it as black.*

We appreciate your attention to detail. We have changed the legend to grey.

7. *Figure 5b and ED Fig 3e. Do the adenoviruses express to similar extents?*

We have included immunoblots to show that Bcl-xL-wild type and the mutants are similarly expressed in new Supplementary Fig. 3d.

Response to Reviewer #2:

The studies presented by Nakamura et al use a combination of in vitro and in vivo techniques to explore the role of BCL-xL S14 phosphorylation in the development of cardiac hypertrophy and failure. The strength of the study is the use of a knock-in mouse line that carries a S14A mutation removing the phosphorylation site. Overall the study is interesting and the phenotypes observed are strong. However some of the statistical analysis needs to be redone and the interpretation of the results is overreaching at times.

Thank you very much for your interest in our paper and constructive criticisms. Our point-by-point responses to Reviewer #2 are listed below.

Major issues:

The first issue is with the fact that this is a whole body Knock in and the results need to be interpreted as such. Fig 1 A (and the quantitation in Sup 1 A) shows baseline levels of BCL-xL phosphorylation. It is unclear of which timepoint this corresponds to. I assume from the supplemental figure this is correctly a sham and not a baseline sample, but especially when looking that close to surgery, the Sham sample from the corresponding time needs to be included. The authors assume the signal is coming from myocytes, however the sham surgery itself may result in a big inflammatory response, additionally it is unclear if any circadian effects may be present. Finally the sham samples all have been normalized to one. This does not allow for the normal distribution of signal and exaggerates the statistical significance. Also the result section speaks of differences between the timepoints but no statistics are shown to support that.

The sham in new Fig. 1a represents data from the heart one hour after sham surgery. We have now added this information in the new Fig. 1a legend. We have also now included data on phosphorylated Bcl-xL expression in the heart after sham surgery *at the same time points as after TAC* in the **new Supplementary Fig. 1a**. In addition, we have replaced the densitometric analysis data with data obtained using the Kruskal-Wallis test (non-parametric analysis), with sham data serving as a control and expressed as 1. The data suggest biphasic increases of Bcl-xL phosphorylation after TAC.

Fig 1 B shows that over 50% of the animals have died within the seven day timepoint. This results in a survival bias for all the other results. Do they die within 1 day? 2 days? This is critical because the authors are trying to suggest that the loss of compensated hypertrophy is the cause of death. Hypertrophy takes time to develop and if the animals are dying that quickly something else may be contributing. Indeed there are a plethora of papers that show inhibition of hypertrophy after TAC is protective.

We have included new survival curves focusing on the first week after TAC in the new Fig. 1b (lower panel). The dysregulated signaling accompanying pathological hypertrophy induces maladaptive responses in the heart, including but not limited to cell death, fibrosis, oxidative stress, insufficient angiogenesis, mitochondrial dysfunction, and metabolic reprogramming. Inhibition of this signaling should be protective against pathological cardiac remodeling and dysfunction. However, before the heart decompensates, compensatory hypertrophy and its signaling are required for the heart to adapt to pressure overload through reduction of left ventricular wall stress and oxygen consumption (*Nat Rev Cardiol* 2018;15:387-407. PMID: 29674714). A lack or excessive inhibition of the hypertrophic response during the acute phase of pressure overload exacerbates acute heart failure, resulting in heart failure-related cardiac death (*Circ Res* 1999; 84:735-40. PMID: 10189362). We have shown recently that aerobic glycolysis also plays an important role in mediating compensatory hypertrophy during the acute phase of pressure overload (*JCI* 2022; 132: e150595). We believe our current findings support this notion.

The S14 KI TAC hearts clearly hypertrophy, as evidenced by the HW changes, but it appears to be eccentric not concentric. Again, how quickly this develops is not clear. The analysis in 2D-G is hampered by the fact that you are looking in the background of massive failure. Again earlier timepoints would be needed and perhaps other hypertrophy markers (BMHC RCAN1.4). Additionally it is not clear that the tunnel positive cells are myocytes, costaining is needed.

We have now conducted *b*-MHC and *Rcan1.4* gene expression analyses (Fig. 2d). Together with the earlier data showing the decreases in HW/TL in S14A KI mice, the new results are in line with our conclusion that inhibition of Bcl-xL phosphorylation suppresses the hypertrophic response to pressure overload. We have also performed co-staining of heart sections with cardiac troponin T and TUNEL to confirm that the labeled cells in Fig. 2f are cardiomyocytes. The results are included in Supplementary Fig. 1f.

Fig 3A is good. 9 hours shows clear changes and differences between the groups. PE seems to may be induce phosphorylation of Bcl-xl in a manner dependent of MEK, but without quantification it is not convincing. It is not clear why the authors didn't use this in vitro approach to look at hypertrophy signaling as well. This would help strengthen the interpretation of the in vivo results. I do see that extended figure 3 e-g uses transfection of gain of function mutants to address PE induced hypertrophy and NFAT3 localization.

We have added quantification analyses of pBcl-xL expression normalized by total Bcl-xL expression in the new Fig. 3e and 3g.

The aim of this study is to investigate the role of Bcl-xL phosphorylation in acute heart failure in response to pressure overload, which is used as a model of acute afterload

mismatch in the clinical setting. The reviewer's point is well taken. However, we believe that we provide sufficient *in vivo* data with appropriate supplementation with *in vitro* experiments to support our conclusions.

*Figure 4 demonstrates that there is no increase in calcium transient in the KI line 1 day after TAG, but instead a decrease along with an increase of the half-life of decay. This is similar to what is observed in failing cells where Ga leak from the sarcoplasmic reticulum through the ryanodine receptor reduces the amplitude of the Ga transient and slows its rate of decay. However, no gain (or loss) in SR Ga content in the KI cells is observed. Inhibition with 2-ABP decreased the calcium transient amplitude in WT TAG myocytes but increased it in KI myocytes. These results are not adequately explained. Additionally, Molkenin has shown that inhibition of IP3R activation has little impact on TAG induced hypertrophy (Nakayama *Girc Res.* 2010 Sep 3;107(5).659-66.) How these results fit into that data is unclear. Likewise loss of ERK1/2 signaling does not result in a similar catastrophic phenotype after TAG (although it may support the concentric/eccentric hypertrophic changes. Kehat I et al . *Girc Res.* 2011 Jan21;108(2).176-83) It would be important to look at mitochondrial phenotypes as they more often lead to the massive changes observed after TAG. Overall there clearly is some interesting findings, ERK1 phosphorylation of BGL-xl is novel, as is the relationship between the IP3R and BGL-xl. I am not convinced of the relationship between the *in vivo* results and the *in vitro* results. Specifically the dramatic impact on cardiac contractility and mortality. Inhibition of IP3R alone is unlikely to account for the observations.*

Although 2-week TAC-induced hypertrophy was not inhibited in Molkenin's cardiomyocyte-specific IP3-sponge transgenic mice, overexpression of IP3R2 in cardiomyocytes significantly enhances TAC-induced hypertrophy (*Girc Res* 2010; 107:659-66). Furthermore, IP3R2-mediated Ca^{2+} release and ERK1/2 are important for physiological hypertrophy signaling in cardiomyocytes (*Girc Res* 2010; 107:659-66; *Girc Res* 2011; 108:176-83). Another study showed that inhibition of calcineurin by cyclosporine suppresses pressure overload-induced cardiac hypertrophy but significantly exacerbates contractile dysfunction within 7 days after surgery, suggesting that a lack of appropriate hypertrophy caused by insufficient Ca^{2+} signaling is detrimental during the acute phase of pressure overload (*Girc Res* 1999; 84:735-40. PMID: 10189362). Based on these findings, it appears that ERK, IP3R, and calcineurin cooperatively induce hypertrophy. Thus, our proposed signaling mechanisms are not inconsistent with those previous findings or with Bcl-xL phosphorylation being important for compensatory hypertrophic signaling during the acute phase of pressure overload.

However, we do agree with the reviewer's comments regarding the downstream mechanisms through which Bcl-xL(S14A) inhibits hypertrophy and promotes heart failure. Our experiment showing that a peptide blocking the Bcl-xL-IP3R interaction alleviates Bcl-xL(S14A)-induced suppression of NFAT *in vitro* suggests the importance of the Bcl-xL phosphorylation-IP3R axis in compensatory cardiac hypertrophy during acute pressure overload. However, considering the subcellular localization of Bcl-xL in mitochondria and the SR, contributions of additional mechanisms, including mitochondrial mechanisms, to the failing cardiac phenotype in Bcl-xL(S14A) mice cannot be excluded. We have discussed this issue as an experimental limitation on page 17, lines 3-7.

Response to Reviewer #3:

All my concerns have been addressed, I have no further comment.

Thank you very much for taking the time to review our manuscript and your valuable suggestions.

**Ser14 phosphorylation of Bcl-xL mediates compensatory cardiac hypertrophy
by stimulating calcium release from the IP3 receptor**

Michinari Nakamura^{1*}, Mariko Aoyagi Keller¹, Nadezhda Fefelova¹, Peiyong Zhai¹, Tong Liu²,
Yimin Tian¹, Shohei Ikeda¹, Dominic P Del Re¹, Hong Li², Lai-Hua Xie¹, Junichi Sadoshima^{1*}

Affiliation:

1. Department of Cell Biology and Molecular Medicine, Cardiovascular Research Institute, Rutgers-New Jersey Medical School, 185 South Orange Ave, Newark, NJ 07103
2. Center for Advanced Proteomics Research, Department of Biochemistry & Molecular Biology, Rutgers New Jersey Medical School, Newark, NJ, 07103

*Correspondence to

Department of Cell Biology and Molecular Medicine, Cardiovascular Research Institute, Rutgers-New Jersey Medical School, 185 South Orange Ave, MSB G-609, Newark, NJ 07103

+1-973-972-8619

E-mail: nakamumi@njms.rutgers.edu (M.N.), sadoshju@njms.rutgers.edu (J.S.)

The authors have declared that no conflict of interest exists.

Abstract

The anti-apoptotic function of Bcl-xL in the heart during ischemia/reperfusion is diminished by K-Ras-Mst1-mediated phosphorylation of Ser14, which allows dissociation of Bcl-xL from Bax and promotes cardiomyocyte death. Here we show that Ser14 phosphorylation of Bcl-xL is also promoted by hemodynamic stress in the heart, through the H-Ras-ERK pathway. Bcl-xL Ser14 phosphorylation-resistant knock-in mice develop less cardiac hypertrophy and exhibit contractile dysfunction and increased mortality during acute pressure overload. Bcl-xL Ser14 phosphorylation enhances the Ca^{2+} transient by blocking the inhibitory interaction between Bcl-xL and IP3Rs, thereby promoting Ca^{2+} release and activation of the calcineurin-NFAT pathway, a Ca^{2+} -dependent mechanism that promotes cardiac hypertrophy. These results suggest that phosphorylation of Bcl-xL at Ser14 in response to acute pressure overload plays an essential role in mediating compensatory hypertrophy by inducing the release of Bcl-xL from IP3Rs, alleviating the negative constraint of Bcl-xL upon the IP3R-NFAT pathway.

Keywords

Bcl-xL; phosphorylation; IP3R; hypertrophy; calcium; heart failure; H-Ras, MEK1; ERK1/2

INTRODUCTION

The Bcl-2 family proteins mediate cell survival and death through apoptosis-dependent and -independent mechanisms ^{1, 2}. The anti-apoptotic property of Bcl-2 and Bcl-xL provides terminally differentiated organs, such as the heart and brain, with crucial protection against oxidative stress. We previously showed that mammalian Hippo kinase Mst1 phosphorylates Bcl-xL at Serine 14 (Ser14) in cardiomyocytes in response to oxidative stress, which disrupts its interaction with the pro-apoptotic protein Bax, thereby increasing the abundance of active Bax, apoptosis, and myocardial ischemia/reperfusion (I/R) injury in mice ^{3, 4}. In a series of intensive analyses of this signaling pathway in cardiac pathology, we found a biphasic increase of Ser14 phosphorylation in the mouse heart in response to acute pressure overload. This led us to investigate whether this post-translational modification contributes to cell survival and death in the context of hemodynamic stress in a manner similar to that during oxidative stress and, if so, how it affects cardiac structure and function. Mst1 is not activated in the heart ⁵ or the cardiomyocytes therein ³ during the acute phase of hypertrophic stimulation, raising the possibility that Bcl-xL Ser14 phosphorylation is mediated by an Mst1-independent mechanism during hypertrophy.

The heart adapts to increased blood pressure by increasing individual cardiomyocyte size, namely hypertrophy, to increase contractility and reduce ventricular wall stress and oxygen consumption ⁶. The MEK-ERK and Ca²⁺-NFAT signaling pathways have been intensively studied as crucial mechanisms for the development of cardiac hypertrophy ^{7, 8, 9}. An improper response to increased workload results in acute decompensated heart failure with high mortality, due in part to incompletely understood pathophysiology and limited available new therapies ¹⁰. It is important to unveil the missing piece(s) in hypertrophic signaling, normalization of which should prevent acute decompensated heart failure. The current study aims to investigate the functional significance and role of Bcl-xL Ser14

phosphorylation in response to pressure overload. Here we report the unexpected observation that phosphorylation of Bcl-xL at Ser14 plays a critical role in the development of compensatory hypertrophy by enhancing Ca^{2+} transients and calcineurin-NFAT signaling. We found that acute pressure overload induces phosphorylation of Bcl-xL at Ser14 through an H-Ras-ERK1/2-dependent mechanism, thereby alleviating the negative constraint of Bcl-xL upon Ca^{2+} release from IP3Rs.

RESULTS

Inhibition of endogenous Bcl-xL phosphorylation exacerbates acute decompensated heart failure in response to pressure overload

First, we examined the phosphorylation status of endogenous Bcl-xL at Ser14 in the heart in the presence of acute pressure overload by applying transverse aortic constriction (TAC) to mouse hearts. Ser14 phosphorylation was significantly increased in the heart within one hour of TAC, returning to baseline by Day 3 but increasing again at around one week and thereafter (Fig. 1 a), while there was little change in the phosphorylation level in the heart after sham surgery (Supplementary Fig. 1a). In order to demonstrate the functional significance of the increased phosphorylation of Bcl-xL at Ser14, phosphorylation-resistant knock-in mice, in which Ser14 has been replaced with Ala (S14A knock-in mice)⁴, and control wild type (WT) mice were subjected to TAC surgery. Although the pressure gradient between the ascending aorta and the femoral artery two weeks after TAC tended to be less in homozygous S14A knock-in mice compared to in WT mice due to cardiac dysfunction ($p=0.09$ between WT and homozygous S14A knock-in mice, unpaired t test), there was no statistically significant difference among WT, heterozygous, and homozygous S14A knock-in mice (Supplementary Fig. 1b). Significantly less Bcl-xL Ser14 phosphorylation in S14A knock-in mouse hearts than in WT mice one hour after TAC confirms the specificity of our Bcl-xL-Ser14 phosphorylation-specific antibody

(Supplementary Fig. 1c). In contrast to the protection conferred by the Bcl-xL S14A knock-in against myocardial I/R injury⁴, S14A knock-in mice exhibited a significantly higher mortality rate after TAC than WT mice. Homozygous S14A knock-in mice exhibited the highest mortality, while heterozygous S14A knock-in mice exhibited an intermediate mortality rate (Log-rank (Mantel-Cox) test $p = 0.0006$) (Fig. 1b). Notably, most of the TAC-induced acute death in homozygous and heterozygous S14A knock-in mice was observed within the first week. We evaluated cardiac function with echocardiography and hemodynamic analyses. There was no significant difference in cardiac function or chamber size among the groups following sham operation. However, both heterozygous and homozygous S14A knock-in mice exhibited a significantly decreased ejection fraction (EF), a measure of left ventricular contractile function, compared to WT mice, which was observed as early as one week post-TAC (Fig. 1c, 1d, and Supplementary Fig. 1d). Left ventricular end-diastolic pressure (LVEDP) and lung weight normalized by tibia length, markers of congestive heart failure, were significantly elevated in S14A knock-in mice one week after TAC (Fig. 1e and 1f). These data indicate that S14A knock-in mice develop acute decompensated heart failure in response to hemodynamic stress, and that phosphorylation of Bcl-xL at Ser 14 is an adaptive response to acute pressure overload.

Inhibition of Bcl-xL phosphorylation suppresses cardiac hypertrophy

To explore the mechanism by which pressure overload exacerbates cardiac dysfunction in S14A knock-in mice, we evaluated the degree of hypertrophy. Increases in the left ventricular wall thickness caused by cardiac hypertrophy reduce ventricular wall stress, thereby playing an adaptive role during the acute phase of pressure overload⁶. Both heterozygous and homozygous S14A knock-in mice exhibited significantly less hypertrophy than WT mice 1 to 4 weeks after TAC, as evidenced by a lower heart weight to tibia length ratio and thinner wall thickness (Fig. 2a and Supplementary Fig. 1d). Wheat Germ

Agglutinin (WGA) staining showed smaller individual cardiomyocytes in S14A knock-in mice than in WT mice (Fig. 2b and 2c). qPCR analyses showed that TAC-induced upregulation of fetal type genes, including *NPPA* and *NPPB*, *MYH7*, and *Rcan1.4*, was suppressed in S14A knock-in mice (Fig. 2d). End-diastolic left ventricular wall stress one week post-TAC was significantly greater in both heterozygous and homozygous S14A knock-in mice than in WT mice (Fig. 2e). These results suggest that the pressure overload is not properly counterbalanced with compensatory hypertrophy in both heterozygous and homozygous S14A knock-in mice. We used homozygous S14A knock-in mice for the rest of the study. Since Ser14 phosphorylation of Bcl-xL inhibits the anti-apoptotic functions of Bcl-xL during myocardial I/R³, we speculated that S14A knock-in may protect the heart against apoptosis during pressure overload. Pressure overload slightly increased apoptosis two weeks after TAC in WT mice. However, the percentage of TUNEL-positive nuclei was similar between WT and S14A knock-in mice after two weeks of TAC (Fig. 2f and 2g) and four weeks of TAC (Supplementary Fig. 1e). It is important to distinguish between the pathological roles of cardiomyocyte and non-cardiomyocyte apoptosis; thus, we further evaluated cardiomyocyte apoptosis by co-staining the heart tissue with cardiac troponin T and TUNEL two weeks after TAC. The rate of cardiomyocyte apoptosis was indistinguishable between WT and S14A knock-in mice (Supplementary Fig. 1f). Cleaved caspase 3 and 9 were slightly elevated two weeks post-TAC to a similar extent in both WT and S14A knock-in mice (Fig. 2h). These data suggest that Bcl-xL Ser14 phosphorylation is critical for the development of compensatory hypertrophy rather than promoting apoptosis during the acute phase of pressure overload.

H-Ras-MEK-ERK signaling is crucial for the increase in Bcl-xL phosphorylation in response to hypertrophic stimuli, which promotes nuclear localization of NFAT3

Since Mst1 is activated in response to pressure overload on Day 3 at the earliest⁵ but phosphorylation of

Bcl-xL at Ser14 is observed within one day, the early Ser14 phosphorylation of Bcl-xL in response to acute pressure overload may be independent of Mst1. In order to better understand which signaling pathway is affected in the heart during the acute phase of pressure overload, we performed RNA-sequencing analyses using mouse hearts subjected to 9 hours of pressure overload (Fig. 3a). As in the qPCR analyses, hypertrophy marker genes, including *NPPA* and *MYH7*, were upregulated after TAC in WT, but not S14A knock-in, mouse hearts (Supplementary Fig. 2a). The gene set enrichment analysis (GSEA) indicated that Fc γ receptor 1, ERK, and integrin pathways were upregulated in WT mouse hearts in response to pressure overload (Fig. 3b and Supplementary Table 1). Since these pathways all utilize the Ras-MEK-ERK pathway, a well established mechanism involved in the development of cardiac hypertrophy ^{8, 11}, we hypothesized that the Ras-MEK-ERK axis may be activated to promote Bcl-xL Ser14 phosphorylation in the heart during acute pressure overload.

Consistent with the results of the RNA-sequencing analysis, ERK1/2 phosphorylation was increased but the level of total ERK1/2 protein was unaltered in the heart after 9 hours of TAC (Fig. 3c). Phosphorylation of ERK1/2, but not Mst1, was also rapidly induced in response to phenylephrine (PE), an α 1-adrenergic receptor agonist, in neonatal rat ventricular cardiomyocytes, along with Bcl-xL Ser14 phosphorylation (Fig. 3d and 3e). Phosphorylation of Akt, a well-known regulator of cardiac hypertrophy ^{6, 12, 13}, was also increased in response to PE. To examine whether activation of either ERK or Akt is involved in Bcl-xL phosphorylation at Ser14 in response to hypertrophic stimuli, cardiomyocytes were pretreated with a MEK inhibitor, PD0325901, or an Akt inhibitor, Triciribine, and stimulated with PE. Inhibition of MEK, but not Akt, attenuated PE-induced increases in Bcl-xL phosphorylation at Ser14 (Fig. 3f, 3g, and Supplementary Fig. 2b), suggesting that the MEK-ERK axis functions upstream of Bcl-xL Ser14 phosphorylation. To determine whether ERK1 directly phosphorylates Bcl-xL at Ser14, we performed *in vitro* kinase assays with active ERK1 and recombinant

GST-Bcl-xL (Supplementary Fig. 2c). Subsequent LC-MS/MS analyses showed that ERK1 directly phosphorylates Bcl-xL at Ser14 (Fig. 3h).

Hypertrophic stimuli activate H-Ras, which, in turn, promotes nuclear translocation of NFAT3 in cardiomyocytes^{3, 14}. To investigate whether Bcl-xL phosphorylation at Ser14 is critical for activation of NFAT3, adenovirus harboring H-Ras was transduced into adult mouse cardiomyocytes isolated from WT and S14A knock-in mice (Supplementary Fig. 2d). H-Ras increased nuclear localization of NFAT3 in WT, but not S14A knock-in, cardiomyocytes (Fig. 3i and 3j). The BH4 domain of Bcl-2 family proteins interacts with H-Ras^{15, 16, 17}. Co-immunoprecipitation assays showed that binding of H-Ras to Bcl-xL was greater in hearts subjected to TAC than in those without TAC (Fig. 3k). The PE-induced increase in the interaction between Bcl-xL and H-Ras was also observed in cardiomyocytes expressing FLAG-Bcl-xL in FLAG-pull down-coimmunoprecipitation assays (Fig. 3l). These results suggest that the H-Ras-MEK-ERK pathway may promote Bcl-xL Ser14 phosphorylation, which in turn activates NFAT signaling in cardiomyocytes.

We further investigated the involvement of the H-Ras-MEK-ERK pathway in Ser14 Bcl-xL phosphorylation in the heart *in vivo*. WT and S14A knock-in mice were pretreated with a MEK inhibitor, PD0325901, or vehicle for three days. The mice were then subjected to either TAC or sham operation in the presence of PD0325901 or vehicle. Inhibition of MEK suppressed the development of hypertrophy in WT mice, as assessed by echocardiographic measurement of the wall thicknesses of the interventricular septum at end-diastole (IVSd) and the LV posterior wall at end-diastole (LVPWd), heart weight normalized by tibia length, and individual cardiomyocyte size, after one week of TAC (Supplementary Fig. 2e-2h). The suppression of cardiac hypertrophy was accompanied by exacerbation of contractile dysfunction and heart failure, as assessed by echocardiographically measured EF and lung weight normalized by tibia length (Supplementary Fig. 2f-2g). However, inhibition of MEK failed to elicit any

additive effect on the S14A knock-in phenotype after one week of TAC. These data suggest that Ser14 Bcl-xL phosphorylation and the H-Ras-MEK-ERK pathway act on the same signaling pathway, thereby mediating TAC-induced cardiac hypertrophy. Furthermore, these results suggest that the H-Ras-MEK-ERK1/2 signaling acts upstream of Ser14 Bcl-xL phosphorylation during pressure overload-induced cardiac hypertrophy.

Ser14 phosphorylation of Bcl-xL promotes Ca²⁺ signaling in response to pressure overload

We then explored the mechanism by which Ser14 Bcl-xL phosphorylation mediates cardiac hypertrophy in response to pressure overload. Since it has been suggested previously that Ca²⁺ serves as a critical second messenger to induce hypertrophy and non-canonical interaction between the BH4 domain of Bcl-xL and sarcoplasmic reticulum (SR)/endoplasmic reticulum (ER) ^{18, 19}, we evaluated the Ca²⁺ transient in cardiomyocytes isolated from S14A knock-in and WT mice. There was no significant difference in the Ca²⁺ transient amplitude, SR Ca²⁺ content, fractional Ca²⁺ release, or T₅₀ between WT and S14A knock-in mice at baseline (Fig. 4a and Supplementary Fig. 3a). Pressure overload increased the Ca²⁺ transient amplitude in cardiomyocytes isolated from WT mice, but decreased it in cardiomyocytes from S14A mice, after one day of TAC. Pressure overload increased the SR Ca²⁺ content in cardiomyocytes isolated from WT mice, but not S14A mice. Pressure overload did not significantly affect fractional Ca²⁺ release or T₅₀ compared to sham operation in cardiomyocytes isolated from WT mice. On the other hand, a significantly decreased fractional Ca²⁺ release and increased T₅₀ were observed after TAC in cardiomyocytes isolated from S14A knock-in mice compared to in those from WT mice (Fig. 4a and Supplementary Fig. 3a). We further assessed the contractile function of individual cardiomyocytes after two days of TAC. Cardiomyocytes isolated from S14A knock-in mice exhibited significantly reduced contraction compared to those from WT mice, consistent with the impaired Ca²⁺ signaling in S14A

knock-in mice after pressure overload (Supplementary Fig. 3b and 3c).

The inositol 1,4,5-triphosphate receptor (IP3R) plays a central role in the development of cardiac hypertrophy, with functional redundancy in all 3 types of IP3Rs^{20, 21}. To examine whether IP3Rs mediate the effect of Bcl-xL Ser14 phosphorylation, 2-APB, a membrane permeable IP3R antagonist, was applied during Ca²⁺ transient measurements in cardiomyocytes (Fig. 4b). 2-APB decreased the Ca²⁺ transient amplitude in WT cardiomyocytes, but not in S14A knock-in cardiomyocytes, after one day of TAC (Fig. 4b and 4c). Although 2-APB slightly increased the Ca²⁺ transient amplitude in S14A knock-in cardiomyocytes, underlying mechanisms are unknown. The differences in Ca²⁺ transient amplitude, SR Ca²⁺ content, and fractional Ca²⁺ release between WT and the S14A knock-in cardiomyocytes after TAC were abolished in the presence of 2-APB (Fig. 4d). The difference in T50 was not abolished but became smaller in the presence of 2-APB. These results suggest that IP3R functions downstream of Bcl-xL Ser14 phosphorylation in cardiomyocytes in response to pressure overload.

Phosphorylation of Bcl-xL disrupts its inhibitory interaction with IP3R

The results presented thus far suggest that Bcl-xL Ser14 phosphorylation promotes pressure overload-induced SR Ca²⁺ release through IP3Rs. Consistent with this notion, the GSEA showed that calcineurin-mediated signaling is significantly suppressed in S14A knock-in mouse hearts subjected to 9 hours of pressure overload compared to in WT mouse hearts with pressure overload (Fig. 5a). The calcineurin-NFAT pathway is activated in response to increases in cytosolic Ca²⁺ and required for the development of compensatory hypertrophy⁹. To validate the result of the pathway analysis, we conducted NFAT-luciferase (NFAT-Luc) assays in cardiomyocytes expressing Bcl-xL-WT, Bcl-xL-S14A (SA) or Bcl-xL-S14D (SD) (Supplementary Fig. 3d). PE significantly increased the NFAT-Luc activity, but this increase was attenuated in the presence of 2-APB in cardiomyocytes transduced with adenovirus harboring Bcl-

xL-WT (Fig. 5b). The PE-induced increase in NFAT activity was significantly attenuated in cardiomyocytes transduced with adenovirus harboring Bcl-xL-S14A and was not further inhibited in the presence of 2-APB (Fig. 5b). These results suggest that Ser14 Bcl-xL phosphorylation is required for PE-induced activation of calcineurin-NFAT signaling through Ca^{2+} release from IP3Rs.

We further explored the mechanism by which Bcl-xL Ser14 phosphorylation enhances IP3R-NFAT signaling in response to hypertrophic stimuli. Immunoprecipitation assays showed that pressure overload decreases the physical interaction between Bcl-xL and IP3R-type 2 in WT mouse hearts, but not S14A knock-in mouse hearts, in the presence of TAC (Fig. 5c), suggesting that Ser14 phosphorylation during TAC inhibits Bcl-xL-IP3R interaction. Recombinant GST-IP3R-fragment 3, which is known to bind to Bcl-xL²², was pulled down more efficiently with flag-Bcl-xL-S14A than with flag-Bcl-xL-S14D, a phosphorylation-mimicking mutant (Fig. 5d). Thus, direct physical interaction between Bcl-xL and IP3R-fragment 3 is negatively regulated by Bcl-xL Ser14 phosphorylation. We further investigated the functional significance of the interaction between Bcl-xL and IP3Rs using a membrane permeable synthetic peptide harboring partial IP3R-fragment 3 (Fragment 3-P)²³. The PE-induced increase in NFAT-Luc activity in cardiomyocytes expressing wild type Bcl-xL was not significantly affected by Fragment 3-P. In contrast, although the PE-induced increase in NFAT-Luc activity was significantly attenuated in cardiomyocytes expressing Bcl-xL-S14A, the effect of PE was fully rescued in the presence of Fragment 3-P (Fig. 5e). These data suggest that phosphorylation of Bcl-xL at Ser14 plays a critical role in stimulating calcineurin-NFAT signaling by inhibiting the physical interaction between Bcl-xL and IP3R-fragment 3, a key mechanism checking Ca^{2+} release from the SR⁹. We also investigated whether Bcl-xL S14D is sufficient to induce NFAT activation and hypertrophy or augment cardiac hypertrophy in response to PE treatment. Bcl-xL S14D neither activated NFAT nor induced or augmented hypertrophy in cardiomyocytes at baseline or in the presence of PE

(Supplementary Fig. 3e-g), suggesting that overexpression of Bcl-xL S14D is not sufficient to activate NFAT or induce/enhance hypertrophy in cardiomyocytes. These results are consistent with the notion that Ser14 Bcl-xL phosphorylation mediates compensatory hypertrophy during the acute phase of pressure overload by controlling Ca^{2+} release from IP3Rs (Supplementary Fig. 3h and 4).

Although Bcl-2 family proteins regulate autophagy activity by interacting with the BH3 domain of Beclin-1^{24, 25}, S14A knock-in mice showed no overt change in the levels of Beclin-1, p62, or LC3-I/II (Supplementary Fig. 3i)⁴, indicating that Bcl-xL Ser14 phosphorylation does not affect autophagy activity at baseline.

DISCUSSION

We show that activation of the H-Ras-MEK1-ERK1/2 pathway by hypertrophic stimuli promotes Bcl-xL-Ser14 phosphorylation, which disrupts the inhibitory interaction between Bcl-xL and IP3Rs, thereby augmenting SR Ca^{2+} release and activating the calcineurin-NFAT pathway, a major signaling mechanism controlling cardiac hypertrophy. This mechanism is crucial for the development of adaptive hypertrophy and suppression of this mechanism promotes acute decompensated heart failure. These results suggest that modulating the level of Bcl-xL Ser14 phosphorylation or disrupting the interaction between Bcl-xL and IP3R could be a promising intervention against heart failure during acute pressure overload.

Increasing evidence indicates that Bcl-2 family proteins, including Bcl-2 and Bcl-xL, modulate Ca^{2+} signaling through a non-canonical mechanism independent of their actions upon mitochondrial outer membrane permeabilization, namely interaction with IP3Rs located on the ER/SR. The BH4 domain of Bcl-2 interacts with IP3R fragment 3, which inhibits the IP3R channel gating^{23, 26, 27}. Lysine (K) 17 in the BH4 domain of Bcl-2 plays a critical role in mediating the inhibitory interaction between

Bcl-2 and IP3R fragment 3, although K17 in Bcl-2 is not conserved in the BH4 domain of Bcl-xL, which has D11 instead²⁸. In contrast to the inhibitory action of Bcl-2 on IP3R, binding of Bcl-xL to two carboxyl terminal BH3-like domains of IP3R *sensitizes* IP3R to low concentrations of IP3, thereby stimulating Ca²⁺ release, mitochondrial bioenergetics and cell survival^{29, 30}. However, Bcl-xL at high concentrations binds to the IP3R fragment 3, thereby inhibiting IP3R channel gating. Furthermore, a synthetic peptide targeting the IP3R fragment 3 completely blocks the inhibitory effect of Bcl-xL on IP3Rs³⁰, suggesting that the effect of Bcl-xL on IP3R is context-dependent. Another study also showed that Bcl-xL always inhibits, rather than activates, IP3R *in vitro*, primarily by binding to the fragment 3 as well as the ligand-binding domain of IP3Rs²². Our data indicate that Bcl-xL binds to IP3Rs and that the disruption of this binding enhances IP3R-mediated Ca²⁺ release in the context of hypertrophic stimuli. We speculate that abundant expression of Bcl-xL in the heart may contribute to the inhibitory interaction with IP3Rs.

Importantly, we here show that the interaction between Bcl-xL and IP3Rs is regulated by Bcl-xL Ser14 phosphorylation. A prior study demonstrated that K17D mutation in the BH4 domain of Bcl-2 abrogates the ability of the BH4 domain to bind to and inhibit IP3R, whereas D11K mutation in the BH4 domain of Bcl-xL rendered the BH4 domain of Bcl-xL capable of binding to and inhibiting IP3R²⁸, suggesting the crucial role of a *negative charge* in preventing the interaction. Ser14 phosphorylation introduces a negative charge in the BH4 domain of Bcl-xL, which may contribute to disrupting the interaction with IP3R. We here show that interaction between Bcl-xL and IP3Rs and consequent activation of NFAT are negatively regulated by Bcl-xL phosphorylation at Ser14, located in the BH4 domain. Ser14 Bcl-xL phosphorylation may disrupt the interaction with IP3R either by adding a negative charge to the BH4 domain or inducing an allosterical conformational change in another part of Bcl-xL, including the BH3 domain. K17 in Bcl-2 is not conserved in Bcl-xL, nor is S14 in Bcl-xL

conserved in Bcl-2. However, both K17 in Bcl-2 and S14 in Bcl-xL allow acidic posttranslational modification in the BH4 domain. Thus, posttranslational modification of K17 and S14 may allow the BH4 domains of these Bcl-2 family proteins to control Ca²⁺ signaling through IR3Rs in a regulated manner. We propose that the phosphorylation status of Ser14 in the BH4 domain of Bcl-xL could serve as a convenient access point to control cardiac hypertrophy by modulating the interaction between Bcl-xL and IP3R in the heart.

Cardiac hypertrophy is required to maintain contractile function and pump sufficient blood throughout the peripheral organs in the face of increased afterload, but persistent hyperactivation of hypertrophic signaling results in cardiac dysfunction, such as aberrant activation of mTOR and YAP ^{6, 13, 31}. Phenylephrine, an α 1-adrenergic receptor agonist, is generally cardioprotective in humans ³². We here show that inhibition of Bcl-xL Ser14 phosphorylation suppresses the hypertrophic signaling mechanism during phenylephrine treatment *in vitro*, consistent with the notion that Bcl-xL Ser14 phosphorylation mediates adaptive cardiac hypertrophy. The Ca²⁺-calcineurin pathway plays a central role in developing hypertrophy ⁹. Inhibition of calcineurin with cyclosporine attenuates pressure overload-induced hypertrophy and enhances susceptibility to decompensation and heart failure ³³, a phenotype similar to that of Bcl-xL-S14A knock-in mice. Cardiac-specific overexpression of MEK1 and activation of ERK1/2 promote physiological hypertrophy, which is reversed by knockdown of ERK1/2 ^{8, 34}, indicating the crucial role of the MEK1-ERK1/2 signaling pathway in adaptive hypertrophy ³⁵. The results presented here indicate that MEK1-ERK1/2 controls cardiac hypertrophy in part through regulation of Bcl-xL phosphorylation. Although ERK1/2 are proline-directed kinases, there is no proline residue near Ser14 of Bcl-xL. However, ERK1/2 can also phosphorylate a serine/threonine residue that does not immediately precede a proline residue ^{36, 37}. Furthermore, a MEK inhibitor had no additive detrimental effect in S14A knock-in mice in response to TAC *in vivo*, consistent with the notion that Bcl-xL Ser14 is

directly phosphorylated by ERK1/2 *in vivo*. However, the possibility that the MEK1-ERK1/2 pathway indirectly phosphorylates Bcl-xL at Ser14 through other serine/threonine kinases cannot be formally excluded. Further investigation is required to clarify this issue.

We have shown previously that Ser14 phosphorylation of Bcl-xL occurs during I/R through a K-Ras-Mst1-dependent mechanism, thereby stimulating apoptosis by promoting dissociation of Bcl-xL from Bax on the outer mitochondrial membrane^{3,4}. Here, we show that Ser14 phosphorylation of Bcl-xL during the acute phase of pressure overload occurs through an H-Ras-MEK-dependent mechanism, presumably in the ER/SR, and plays a salutary role by promoting compensatory cardiac hypertrophy. Interestingly, the level of apoptosis after pressure overload was similar between Bcl-xL-S14A knock-in and WT mice. Since both depressed cardiac function and increased LV wall stress could have induced higher levels of apoptosis, the fact that apoptosis is not increased in S14A knock-in mice suggests that a mechanism suppressing apoptosis may still be operative in these mice. Even so, it is puzzling to observe that Bcl-xL-S14A mice exhibited a more *detrimental* cardiac phenotype during acute pressure overload. One possible explanation for the discrepancy could be that regulation of the signaling complexes in which Bcl-xL takes part, its interacting partners, and its subcellular localization are distinct between pressure overload and I/R. For example, I/R induces K-Ras-induced activation of Mst1 in mitochondria, which induces phosphorylation of Bcl-xL Ser14 and its dissociation from Bax at the mitochondrial outer membrane³. We speculate that Bcl-xL Ser14 phosphorylation by the H-Ras-MEK pathway occurs at a distinct subcellular location during the acute phase of pressure overload, namely at Bcl-xL bound to IP3R in the SR. Consistent with this hypothesis, Bcl-xL physically interacts with H-Ras (Fig. 3h). Furthermore, H-Ras and K-Ras induce distinct phenotypes in the heart in response to pressure overload³⁸. Our results also suggest that activation of the signaling mechanism mediating compensatory hypertrophy is important during the early phase of pressure overload, even if it might promote apoptosis.

Whether Bcl-xL Ser14 phosphorylation is regulated during the chronic phase of heart failure and, if so, where in the cell it is regulated and how it affects cell death remain to be clarified.

We have shown previously that overexpression of Bcl-xL(S14A) inhibits dissociation of Bcl-xL from the BH3 domain of Bax during myocardial reperfusion, thereby inhibiting apoptosis⁴. Thus, by inference, Bcl-xL (S14A) may exhibit increased binding to the BH3 domain of Beclin 1, thereby inhibiting autophagy. S14A knock-in mice showed no overt change in the levels of Beclin-1, p62, or LC3-I/II, indicating no or minimal effect of Bcl-xL S14 phosphorylation on autophagy activity in the heart at baseline. However, further investigation is required to test whether Bcl-xL Ser14 phosphorylation decreases the binding of Bcl-xL to Beclin 1, thereby stimulating autophagy in stress condition, such as pressure overload.

It has been shown that the combined use of a MEK inhibitor and ABT-263, a chemical inhibitor of Bcl-xL, promotes tumor regression in KRAS mutant cancer models³⁹. MEK inhibitors alone may promote the anti-apoptotic actions of Bcl-xL by inhibiting Ser14 phosphorylation and stimulating Bcl-xL-Bax interaction, thereby diminishing their killing effects. We speculate that concomitant use of ABT-263 would enhance the anti-cancer effect of the MEK inhibitors by eliminating the anti-apoptotic actions of Bcl-xL.

There are some limitations to our study. We used rat neonatal ventricular cardiomyocytes in some *in vitro* experiments to evaluate the upstream kinases that phosphorylate Bcl-xL at Ser14 and determine the effect of Bcl-xL Ser14 phosphorylation upon NFAT activity. Since cultured neonatal cardiomyocytes may not fully recapitulate the stress response that occurs in adult hearts at baseline and in response to pressure overload, further investigation is needed to elucidate the molecular mechanisms through which pressure overload leads to Bcl-xL Ser14 phosphorylation, and whether Bcl-xL Ser14 phosphorylation, in turn, regulates NFAT activity *in vivo*. Second, mitochondrial uptake of calcium

released from IP3Rs is a critical determinant of cell survival, in part through regulation of apoptosis and autophagy. Thus, how the regulation of IP3R calcium release by Bcl-xL Ser14 phosphorylation affects the ER/SR-mitochondrial connection remains to be clarified. Finally, although we show that Bcl-xL

Ser14 phosphorylation plays a salutary role during the acute phase of pressure overload through regulation of IP3R-mediated compensatory hypertrophy, considering the subcellular localization of Bcl-xL in mitochondria and the SR, contributions of additional mechanisms, including mitochondrial mechanisms, to the failing cardiac phenotype in Bcl-xL(S14A) mice cannot be excluded.

In summary, the current study demonstrates that Bcl-xL Ser14 phosphorylation is essential for adaptive hypertrophy to prevent acute decompensated heart failure, in part through IP3R-mediated Ca^{2+} release and calcineurin-NFAT signaling, during acute pressure overload.

METHODS

Mice

The Bcl-xL S14A knock-in mice (C57BL/6 background) were generated as previously described⁴. Male C57BL/6J wild-type mice were purchased from Jackson Labs at 5-8 weeks of age. The ERK inhibitor, PD0325901 (15 mg/kg/day), or vehicle (DMSO) was administered orally once a day for 3 days before surgery and 2 days after surgery. Mice were housed in a temperature-controlled environment within a range of 21°C - 23°C with 12-hour light/dark cycles, in which they received food and water *ad libitum*. We used age-matched male mice in all animal experiments. The sample size required was estimated to be $n = 5-8$ per group according to the Power analysis based upon previous studies examining the effects of pressure overload on cardiac hypertrophy and hypertrophic signaling. All protocols concerning the use of animals were approved by the Institutional Animal Care and Use Committee at Rutgers New Jersey Medical School and all procedures conformed to NIH guidelines (Guide for the Care and Use of

Laboratory Animals).

Cell Line

HEK293 cells were maintained at 37°C with 5% CO₂ in Dulbecco's modified Eagle's Medium with 10% fetal bovine serum.

Primary Rat Neonatal Cardiomyocytes

Primary cultures of ventricular cardiomyocytes were prepared from 1-day-old Crl:(WI)BR-Wistar rats (Envigo, Somerville) and maintained in culture as described previously¹². The neonatal rats were deeply anesthetized with isoflurane. The chest was opened, and the heart was harvested. A cardiomyocyte-rich fraction was obtained by centrifugation through a discontinuous Percoll gradient. Cardiomyocytes were cultured in complete medium containing Dulbecco's modified Eagle's medium/F-12 supplemented with 5% horse serum, 4 µg/ml transferrin, 0.7 ng/ml sodium selenite, 2 g/l bovine serum albumin (fraction V), 3 mM pyruvate, 15 mM HEPES pH 7.1, 100 µM ascorbate, 100 mg/l ampicillin, 5 mg/l linoleic acid, and 100 µM 5-bromo-2'-deoxyuridine (Sigma). Culture dishes were coated with 0.3% gelatin or 2% gelatin for immunofluorescence staining on chamber slides.

Isolation of Adult Cardiomyocytes for Signaling Experiments

Adult mouse cardiomyocytes were isolated as described previously with a modification^{12, 40}. Briefly, the heart of a male mouse was perfused with 12 ml EDTA buffer [130 mM NaCl, 5 mM KCl, 0.5 mM NaH₂PO₄, 10 mM HEPES, 10 mM Glucose, 10 mM BDM, 10 mM Taurine, 5 mM EDTA] to stop the beating of the heart. Digestion was achieved using 30 ml perfusion buffer [130 mM NaCl, 5 mM KCl, 0.5 mM NaH₂PO₄, 10 mM HEPES, 10 mM Glucose, 10 mM BDM, 10 mM Taurine, 1 mM MgCl₂] containing Collagenase type II (0.5 mg/ml) and Protease XIV (0.05 mg/ml). Cellular dissociation was stopped by addition of 5 ml Perfusion buffer containing 5% FBS and 100 mM BSA-conjugated fatty acid cocktail (palmitic acid: oleic acid: linoleic acid = 2:1:1 and BSA: fatty acid = 1:5). Cardiomyocytes

and non-cardiomyocytes were separated by 4 sequential rounds of gravity settling with calcium reintroduction medium containing 100 mM BSA-conjugated fatty acid cocktail.

Transverse Aortic Constriction (TAC)

Male 8 to 10-week-old animals were subjected to TAC or sham surgery. Mice were anesthetized with pentobarbital (60-70 mg/kg, intraperitoneal injection) and mechanically ventilated with a tidal volume of 0.2 ml and a respiratory rate of 110 breaths per minute. The mice were kept warm with heat lamps. It took around 5 minutes to establish full anesthesia. The chest and neck were shaved by clipper and the skin was cleaned using betadine and 70% ethanol 3 times. Sterile ophthalmic ointment was applied to the eyes. Mice were placed in a supine position. A lack of toe pinch/tail pinch reflex was checked before making the incision. Before making the surgical incision, we subcutaneously injected a very small volume of bupivacaine along the incision line. A midline cervical incision (15-20 mm) was made to assist intubation of the trachea and for access to the intercostal space. The left chest was opened at the second intercostal space. The intercostal incision was less than 0.5 cm and opened by self-designed stretchers made of 25-gauge needles connected to rubber bands and fixed on the surgical board by pins. With the aid of a dissecting microscope, aortic constriction was performed by ligating the transverse thoracic aorta between the innominate artery and left common carotid artery with a 28-gauge needle using a 7-0 prolene suture, and then removing the needle. Sham operation was performed without constricting the aorta. During surgery, the depth of anesthesia was monitored periodically by checking pedal reflex. Thoracotomy incision, overlying muscle layers and skin were closed in layers using 5-0 prolene sutures, and the pneumothorax was reduced. The TAC procedure was completed within 20-30 minutes per mouse. The mice were then treated with Buprenex-SR (1.0-1.2 mg/kg, SC) and monitored while being allowed to recover in a warm incubator. When recovered from anesthesia 1-2 hours after the closure of the chest, the mice were extubated and returned to their cages. Upon completion of all

experimental procedures, mice were euthanized by cervical dislocation followed by harvest of the hearts for biochemical studies, including signaling pathways.

Echocardiography

Mice were anesthetized using 12 μ l/g body weight of 2.5% avertin (Sigma-Aldrich), and echocardiography was performed using ultrasound (Vivid 7, GE Healthcare). It took around 10-20 minutes from the establishment of anesthesia to the completion of echocardiography and 1-2 hours to fully recover from anesthesia after echocardiography. A 13-MHz linear ultrasound transducer was used. Mice were subjected to 2-dimension guided M-mode measurements of LV internal diameter at the papillary muscle level from the short-axis view to measure systolic function and wall thickness, which were taken from at least three beats and averaged. LV ejection fraction was calculated as follows: Ejection fraction = $[(LVEDD)^3 - (LVESD)^3] / (LVEDD)^3 \times 100$. End-diastolic wall stress was calculated using echocardiographic and hemodynamic parameters as follows: End-diastolic wall stress = LV end-diastolic pressure \times LVDd / $2 \times$ LVPWd \times (1 + LVPWd / $2 \times$ LVDd) ⁴¹.

Hemodynamic Measurements

Mice were anesthetized with Avertin (300 mg/kg, intraperitoneal injection) to measure arterial pressure gradients and LV end-diastolic pressures. The chest and neck were shaved by clipper and the skin was cleaned using betadine and 70% isopropyl alcohol three times. Mice were then placed in a supine position. The lack of pedal reflex was confirmed prior to making an incision. A small incision (5-10 mm) was made on the neck. Under a dissecting microscope, the common carotid artery was surgically isolated and clamped proximally and distally. A small incision (0.5-1 mm) was made in the carotid artery, and a high-fidelity micromanometer catheter (1.4 French; Millar Instruments Inc.) was inserted into the artery and advanced into the aorta to measure aortic pressure proximal to the constriction site and then into the LV cavity to measure LV pressure and its first derivatives. A separate high-fidelity

micromanometer catheter was inserted via the femoral artery and advanced into the aorta to measure aortic pressure distal to the constriction simultaneously. During the procedure, the depth of anesthesia was monitored by checking pedal reflex periodically to ensure the surgical plane of anesthesia.

Immunoblotting

Cardiomyocyte lysates and heart homogenates were prepared in RIPA buffer containing protease and phosphatase inhibitors (Sigma-Aldrich) as described previously^{4, 42}. Lysates were centrifuged at 13,200 rpm at 4°C for 15 minutes. Protein concentrations were determined using a standard BCA assay. Total protein lysates (10-30 µg) were incubated with SDS sample buffer (final concentration: 100 mM Tris (pH 6.8), 2% SDS, 5% glycerol, 2.5% 2-mercaptoethanol, and 0.05% bromophenol blue) at 95°C for 5 minutes. The denatured protein samples were separated by SDS-PAGE, transferred to polyvinylidene difluoride membranes by wet electrotransfer, blocked in either 5% (w/v) BSA or 5% (w/v) non-fat dry milk in 1xTBS/0.5% Tween 20 at room temperature for 1 hour, and probed with primary antibodies at 4°C overnight. After washing with 1xTBS/0.5% Tween 20 for 20 minutes, the membranes were incubated with the corresponding secondary antibody at room temperature for 1 hour. After washing with 1xTBS/0.5% Tween 20 for 45 minutes, the membranes were developed with ECL Western blotting substrate, followed by acquisition of digital images with the ChemiDoc MP Imaging System (Bio-Rad). The intensities of Western blot bands were quantified using ImageJ software. Uncropped blotting images with molecular markers are provided in Supplementary Fig. 5.

Antibodies and Reagents

The following commercial antibodies were used at the indicated dilutions: rabbit monoclonal Bcl-xL antibody (#2764) (1:6,000), rabbit cleaved caspase-3 antibody (#9661) (1:2,000), rabbit cleaved caspase-9 antibody (#9507) (1:2,000), rabbit monoclonal p44/42 MAPK (Erk1/2) antibody (#9102) (1:5,000), rabbit monoclonal phospho-p44/42 MAPK (Erk1/2) (Thr202/Tyr204) antibody (#4370)

(1:5,000), rabbit monoclonal phospho-GSK-3 α /J3 (Ser21/9) antibody (#9323) (1:3,000), rabbit monoclonal GSK-3 α /J3 antibody (#5676) (1:5,000), rabbit polyclonal phospho-Akt (Ser473) antibody (#9271) (1:4,000), rabbit polyclonal Akt antibody (#9272) (1:8,000), rabbit monoclonal phospho-MST1 (Thr183)/MST2 (Thr180) antibody (#49332) (1:1,000), rabbit monoclonal NFAT3 antibody (#2183) (1:1,000), rabbit monoclonal GAPDH antibody (#5174) (1:8,000), rabbit monoclonal Histone H3 antibody (#4499) (1:10,000), anti-mouse or -rabbit IgG, HRP-linked antibodies (#7076 and #7074) (1:5,000) (Cell Signaling Technology); α -actinin (sarcomeric) antibody (#A7811) (1:4,000), rabbit monoclonal α -tubulin antibody (T6199) (1:8,000) (Sigma-Aldrich); mouse monoclonal IP3R-II antibody (Santa Cruz Biotechnology #sc-398434) (1:1,000); rabbit polyclonal H-Ras antibody (C-20) (Santa Cruz Biotechnology #sc-520)(1:1,000); and mouse monoclonal MST1 antibody (BD Transduction Laboratories #611052) (1:4,000). For detection of phosphorylation of Bcl-xL at Ser14, a polyclonal phosphorylation-specific antibody was generated by immunizing rabbits with a phosphopeptide as described previously (1:1,000)³. Antibodies were diluted in either 5% (w/v) BSA or 5% (w/v) non-fat dry milk in 1xTBS/0.5% Tween 20, depending on the level of background intensity. The following reagents were used: MEK inhibitor (PD0325901) and Akt inhibitor (Triciribine) (Sigma-Aldrich).

Subcellular Fractionation

Cultured neonatal rat cardiomyocytes were washed with PBS and collected with ice-cold PBS, followed by centrifugation at 600g for 5 minutes. Cardiomyocytes were then resuspended in hypotonic lysis buffer (10 mM K-HEPES pH 7.9, 1.5 mM MgCl₂, 10 mM KCl, 0.1 mM EGTA, 0.1 mM EDTA, 1% IGEPAL, 1% Phosphatase Inhibitor Cocktail, and 1% Protease Inhibitor Cocktail) and were incubated for 15 minutes on ice with intermittent pipetting. Whole-cell lysates were centrifuged at 1200g for 5 minutes. The supernatant was collected for the cytosolic fraction, and the pellets were resuspended in

lysis buffer (20 mM K-HEPES, 25% Glycerol, 0.45 M NaCl, 1.5 mM MgCl₂, 1 mM EGTA, 1 mM EDTA, 1% Phosphatase Inhibitor Cocktail, and 1% Protease Inhibitor Cocktail) and were incubated for 15 minutes on ice with intermittent pipetting. The total homogenate was centrifuged at 13,000 rpm for 10 minutes to collect the nuclear fraction. The pelleted nuclei were resuspended in lysis buffer and protein content was determined for all fractions.

Immunoprecipitation

Heart samples were lysed with lysis buffer containing 50 mM Tris-HCl pH 7.4, 150 mM NaCl, 1% Triton-X 100, 1% Sodium Deoxycholate, Protease Inhibitor Cocktail (Sigma), and Phosphatase Inhibitor Cocktail (Sigma). Primary antibody was covalently immobilized on protein A/G agarose using the Pierce Crosslink Immunoprecipitation Kit according to the manufacturer's instructions (Thermo Scientific). Samples were incubated with immobilized antibody beads overnight at 4°C. After immunoprecipitation, the samples were washed with lysis buffer five times. They were then resuspended with lysis buffer and the immunoprecipitates were subjected to immunoblotting using specific primary antibodies and a conformation-specific secondary antibody (Clean-Blot IP Detection Reagent (ThermoFisher Scientific)).

FLAG Pull-down assay

Cardiomyocytes were transduced with adenovirus harboring FLAG-Bcl-xL or LacZ for 2 days, followed by treatment with PE or vehicle for 20 minutes. The cardiomyocytes were collected with RIPA buffer containing protease and phosphatase inhibitors (Sigma-Aldrich) as described previously⁴. Lysates were centrifuged at 13,200 rpm at 4°C for 15 minutes. After protein concentrations were determined using a standard BCA assay, protein lysates were incubated with anti-FLAG M2 Magnetic Beads (Sigma-Aldrich) overnight at 4°C. After FLAG pull-down, the samples were washed with lysis buffer five times. They were then resuspended with lysis buffer and the immunoprecipitates were subjected to

immunoblotting using specific primary and secondary antibodies.

Adenovirus Constructs

Recombinant adenovirus vector for overexpression was constructed, propagated and titered as previously described^{3, 4, 12, 42}. pBHGlox- Δ E1,3Cre (Microbix), including the Δ E adenoviral genome, was co-transfected with the pDC shuttle vector containing the gene of interest into 293 cells.

Replication-defective human adenovirus type 5 (devoid of E1) harboring full length wild type or mutant Bcl-xL cDNA (Ad-Bcl-xL) or H-Ras cDNA (Ad-H-Ras) was generated by homologous recombination in 293 cells. Adenovirus harboring beta-galactosidase (Ad-LacZ) was used as a control.

Recombinant Proteins

The bacterial expression vectors for GST-fused Bcl-xL-full length-wild type (WT) and -mutant (S14A) and IP3R fragment 3 were generated by insertion of mouse Bcl-xL and IP3R fragment 3 cDNA amplified by PCR into the pCold-GST-vector. The BL21 E. coli strain was transformed with the expression vectors, grown in 3 ml LB medium containing ampicillin overnight at 37°C, and then transferred to 250 ml LB medium containing ampicillin. Protein expression was induced by addition of 1 mM isopropylthio-b-galactoside. After overnight culture at 15°C, the E. coli were lysed in lysis buffer (1% Triton X-100 and 1 mM DTT in PBS) with sonication. The lysate was incubated with 0.5 ml Glutathione-sepharose 4B (GE Healthcare) for 1 hour at 4°C. The sepharose was washed 3 times with 5 ml lysis buffer, and then suspended with 1 ml cleavage buffer (20 mM Tris pH 7, 150 mM NaCl, 1 mM DTT). A membrane-permeable synthetic peptide corresponding to the IP3R fragment 3 was generated by GeneScript as previously described with modification²³ using the following amino acid sequence (RKKRRQRRRGKNVYTEIKCNSLLPLDDIVRV).

In Vitro Kinase Assay

Recombinant active ERK1 was purchased from Millipore Sigma. Recombinant GST-tagged full-length

Bcl-xL protein was generated using the pCold-GST-vector. Recombinant active ERK1 (10 ng, Millipore Sigma #14-439) was incubated with recombinant GST-Bcl-xL-WT (1 mg) in a kinase buffer (50 mM HEPES (pH 7.4), 15 mM MgCl₂ and 200 mM sodium vanadate) in the presence or absence of 100 mM ATP at 30°C for 15 min. Recombinant phosphorylated GST-Bcl-xL protein was separated by SDS-PAGE, followed by immunoblots with anti-Bcl-xL Ser14 phospho-specific antibody or staining with Coomassie Brilliant Blue and LC-MS/MS analysis.

Mass Spectrometry

A kinase reaction was performed using recombinant GST-tagged human Bcl-xL and recombinant active ERK1 protein. Phosphorylated protein was separated by SDS-PAGE and stained with Coomassie Brilliant Blue. The gel band of interest was excised for in-gel trypsin digestion. The resulting peptides were C18 desalted and analyzed directly by LC-MS/MS analysis on an Orbitrap Fusion Lumos MS instrument. The MS/MS spectra were searched against a Uniprot human database using the Sequest search engine on the Proteome Discoverer platform (V2.4). STY phosphorylation was set as the variable modification. The false discovery rate of protein identification is less than 1%.

In Vitro Binding Assays

Flag-Bcl-xL-S14D or -S14A protein was overexpressed using an adenovirus overexpression system in rat neonatal cardiomyocytes. Cardiomyocyte lysates were collected and Flag-tagged proteins were immunoprecipitated using Flag-agarose beads (Sigma Aldrich). Immunoprecipitated Flag-Bcl-xL-S14D or -S14A proteins were incubated with recombinant GST-fused IP3R fragment 3 in lysis buffer containing 50 mM Tris-HCl pH 7.4, 150 mM NaCl, 1% Triton-X 100, 1% Sodium Deoxycholate, Protease Inhibitor Cocktail (Sigma Aldrich), and Phosphatase Inhibitor Cocktail (Sigma Aldrich) with rotation for 1 hour at 4°C, followed by pull-down with Flag-agarose beads. After washing five times with lysis buffer, proteins were eluted with 5xSDS sample buffer, followed by SDS-PAGE and

Coomassie Brilliant Blue staining.

RNA-Seq Library Preparation, Sequencing, and Data Analysis

Total RNA was isolated from mouse hearts using TRIzol (Invitrogen). Isolated RNA was checked for integrity on an Agilent Bioanalyzer 2100; samples with RNA integrity number >7.0 were used for subsequent processing. Total RNA was subjected to two rounds of poly(A) selection using oligo-d(T)₂₅ magnetic beads (New England Biolabs). A paired-end (strand specific) cDNA library was prepared using the NEB Next Ultra-directional RNA-seq protocol. Briefly, poly(A)⁺ RNA was fragmented by heating at 94°C for 10 minutes, followed by reverse transcription and second strand cDNA synthesis using the reagents provided in the NEB Next kit. End-repaired cDNA was then ligated with double stranded DNA adapters, followed by purification of ligated DNA with AmpureXP beads. cDNA was then amplified by PCR for 15 cycles with a universal forward primer and a reverse primer with bar code. The sequencing of the cDNA libraries was performed on the Illumina NextSeq 500 platform (Illumina, San Diego, CA) using the single-read 1x75 cycles configuration. The raw reads files have been deposited in the NCBI Gene Expression Omnibus.

Raw reads were quality trimmed using Trimmomatic-0.39 with leading and trailing Q score 20, minimum length 35 bp. The cleaned reads were mapped to *Mus musculus* genome GRCm38 using HISAT2 (Version 2.2.1). The reference genome sequence and annotation files were downloaded from ENSEMBL release 97 (Mus_musculus.GRCm38.101fa and Mus_musculus.GRCm38.101.gtf). The aligned read counts were obtained using htseq-count as part of the package HTSeq-0.6.1. The Bioconductor package edgeR (Version 3.18.1 with limma 3.32.10) was used to perform the differential gene expression analysis under R environment, R version 4.1.1. Expression patterns of regulated genes were graphically represented in a heat map. Hierarchical clustering was performed to group genes with similar features in the expression profile. Normalized expression data were also analyzed with GSEA

version 4.1.0 software using the JAVA program (Broad Institute, Cambridge, MA). All gene sets were obtained from the Molecular Signatures Database version 7.3 distributed on the GSEA Web site. Quantitative RT-PCR

Total RNA was prepared from mouse hearts using TRIzol (Invitrogen) as previously described⁴². cDNA was generated using 300 ng total RNA and SuperScript III Reverse Transcriptase (ThermoFisher). Using Maxima SYBR Green qPCR master mix (Fermentas), real-time RT-PCR was performed under the following conditions: 94°C for 10 minutes; 40 cycles of 94°C for 15 seconds, 58°C for 30 seconds, 72°C for 30 seconds; and a final elongation at 72°C for 15 minutes. Relative mRNA expression was determined by the $\Delta\Delta$ -Ct method normalized to the ribosomal RNA (18S) level. The following oligonucleotide primers were used: NPPA, sense 5'-ATGGGCTCCTTCTCCATCAC-3' and antisense 5'-ATCTTCGGTACCGGAAGCTG-3'; NPPB, sense 5'-AAGTCCTAGCCAGTCTCCAGA-3' and antisense 5'-GAGCTGTCTCTGGGCCATTTC-3'; MYH7, sense 5'-GCCAACACCAACCTGTCCAAGTTC-3' and antisense 5'-TGCAAAGGCTCCAGGTCTGAGGGC-3'; Rcan1.4, sense 5'-TCCAGCTTGGGCTTGACTGAG-3' and antisense 5'-ACTGGAAGGTGGTGTCTTGT-3'; 18S rRNA, sense 5'-CGCGGTTCTATTTTGTTGGT-3' and antisense 5'-AGTCGGCATCGTTTATGGTC-3'.

Immunohistochemistry

The heart tissue was washed with PBS, fixed in 4% paraformaldehyde overnight, embedded in paraffin, and sectioned at 10- μ m thickness onto a glass slide. After de-paraffinization, sections were stained with wheat germ agglutinin (WGA) for evaluation of the cross-sectional area of cardiomyocytes, or TUNEL for evaluation of apoptosis. The outline of 100-200 myocytes was traced in each section, using ImageJ software (NIH). For co-staining with TUNEL and troponin T, the heart sections were first stained with TUNEL and washed with PBS, followed by incubation with cardiac troponin T (Invitrogen #MA5-

12960) at 4°C overnight. After washing with PBS, the heart sections were incubated with Alexa 568 anti-mouse antibody at room temperature for 1 hour to visualize cardiac troponin T. The heart sections were mounted with VECTASHIELD Mounting Media with DAPI.

Reporter Gene Assay

NFAT reporter gene activity in rat neonatal cardiomyocytes was measured with a luciferase assay system (Promega). Cardiomyocytes were transfected with NFAT luciferase reporter plasmids (a gift from Dr. Toren Finkel, Addgene #10959) and Bcl-xL-WT, -S14A, or S14D in the pDC316 vector overnight (24 well plate) using LipofectAmine 2000 (Invitrogen). The NFAT reporter gene assay was performed after 4 hours of phenylephrine treatment in the presence or absence of 2-APB (25 μ M) or synthetic peptide (20 μ M). Cardiomyocytes were lysed with 50 μ l Reporter lysis buffer (24 wells each). The luminescence reaction was started by adding 5 μ l lysate to 50 μ l Reaction buffer, and luminescence was measured for 10 seconds using an OPTOCOMP I luminometer (MGM Instruments). The luminescence was normalized by protein content measured by protein assay kit (BioRad).

Cardiomyocyte cell size measurement

Rat neonatal cardiomyocytes were cultured on coverslips, washed with PBS twice, fixed with 3.7% paraformaldehyde for 15 minutes, and washed with PBS three times. Samples were permeabilized with PBST (0.5% Triton-X in PBS) for 15 min, and blocked in 5% BSA, 5% goat serum in PBST for 30 minutes at 37°C. Cardiomyocytes were stained with Alexa Fluor 555 phalloidin (Thermo Fisher Scientific, A34055). Samples were washed with PBS and mounted on glass slides with mounting medium (VECTASHIELD, Vector Laboratories). Cells were observed under a fluorescence microscope. Cell size was measured for 25 - 30 cells for each condition in each experiment and the mean value was taken as representative of the experiment. This experiment was performed independently five times (n = 5).

Isolation of ventricular cardiomyocytes for cell shortening and Ca²⁺ transient experiments

Left ventricular myocytes were enzymatically isolated from hearts of Bcl-xL S14A knock-in and control WT male mice (2-3 months). Mice were deeply anesthetized with isoflurane in a covered beaker before hearts were removed and perfused retrogradely in Langendorff fashion at 37°C with Ca²⁺-free Tyrode's solution containing 1.0 mg/ml collagenase type II (Worthington, Biochemical Corp., Lakewood, NJ, USA) and 0.1 mg/ml thermolysin (or protease type XIV, Sigma) for 10-12 min. After the enzyme solution was washed out, the hearts were transferred from the Langendorff apparatus to petri dishes. Left ventricles were gently teased apart with forceps. Ca²⁺ concentration was gradually increased to 1.0 mM. Finally, the cell suspension was filtered through a 200 µm nylon mesh. Cardiomyocytes were stored at room temperature and used within 8 hours after isolation. All electrophysiological experiments were performed at 35-37°C.

Intracellular Ca²⁺ measurement

Ventricular myocytes were loaded with the Ca²⁺ indicator Fluo-4 AM (4 µm, Invitrogen, Grand Island, NY, USA) for 30 min. After washing and de-esterification (30 min), the myocytes were transferred to a heated chamber (37°C) on a Nikon Eclipse TE200 inverted microscope (Nikon, Tokyo, Japan) with a Fluor x40 oil objective lens (numerical aperture 1.3). The intracellular Ca²⁺ fluorescence (excitation/emission wavelengths: 485/530 nm) was recorded with a spatial resolution of 500 × 400 pixels at 50 frames per second by an iXon Charge-Coupled Device (CCD) camera (Andor Technology, Concord, MA, USA) operated with Imaging Workbench software (INDEC BioSystems, Los Altos, CA, USA). The Ca²⁺ fluorescence intensity was expressed as the ratio F/F₀ (fluorescence (F) over the baseline diastolic fluorescence (F₀)).

Measurement of single-cell shortening

Myocytes were placed in a heated chamber (37°C) on a Nikon Eclipse TE200 inverted microscope and

were subjected to 1-Hz field-pacing using a stimulator (Grass Instruments, West Warwick, Rhode Island, USA). Changes in cell length were monitored by a video-based edge detection system (model VED-105, Crescent Electronics, Sandy, UT, USA) with a 30-ms time resolution. Single-cell shortening was recorded and analyzed using commercially available pCLAMP 10 software (Molecular Devices, Sunnyvale, CA). The cell shortening was calculated as a percentage of shortening from the baseline cell length in the relaxed state.

STATISTICS and REPRODUCIBILITY

All values are expressed as mean \pm SEM. Statistical analyses were carried out by 2-tailed unpaired Student *t* test for 2 groups or one- or two-way ANOVA followed by the Tukey post-hoc analysis for 3 groups or more unless otherwise stated. If the data distribution failed normality by the Shapiro-Wilk test or Kolmogorov-Smirnov test, the Mann-Whitney *U* test for 2 groups was performed. The statistical analyses used for each figure are indicated in the corresponding figure legends. Survival curves were plotted by the Kaplan-Meier method, with statistical significance analyzed by log-rank test. Statistical analyses were conducted using Prism 9 (GraphPad Software). All experiments are represented by multiple biological replicates or independent experiments. The number of replicates per experiment are indicated in the legends. All experiments were conducted using at least two independent experimental materials or cohorts to reproduce similar results. No sample was excluded from analysis. A *p* value of less than 0.05 was considered significant.

DATA AVAILABILITY

All data generated or analyzed during this study are included in this published article and its Supplementary Information. RNA-sequencing data have been deposited at GEO and are publicly

available as of the date of publication (Accession numbers: GSE199705). No original code was generated in this study. Any additional information required to reanalyze the data reported in this paper is available from the lead contact upon request.

REFERENCES

1. Chong SJF, Marchi S, Petroni G, Kroemer G, Galluzzi L, Pervaiz S. Noncanonical Cell Fate Regulation by Bcl-2 Proteins. *Trends Cell Biol* **30**, 537-555 (2020).
2. Singh R, Letai A, Sarosiek K. Regulation of apoptosis in health and disease: the balancing act of BCL-2 family proteins. *Nat Rev Mol Cell Biol* **20**, 175-193 (2019).
3. Del Re DP, et al. Mst1 promotes cardiac myocyte apoptosis through phosphorylation and inhibition of Bcl-xL. *Mol Cell* **54**, 639-650 (2014).
4. Nakamura M, Zhai P, Del Re DP, Maejima Y, Sadoshima J. Mst1 -mediated phosphorylation of Bcl-xL is required for myocardial reperfusion injury. *JCI Insight* **1**, (2016).
5. Ikeda S, et al. Hippo Deficiency Leads to Cardiac Dysfunction Accompanied by Cardiomyocyte Dedifferentiation During Pressure Overload. *Circ Res* **124**, 292-305 (2019).
6. Nakamura M, Sadoshima J. Mechanisms of physiological and pathological cardiac hypertrophy. *Nat Rev Cardiol* **15**, 387-407 (2018).
7. Luo Y, et al. Cooperative Binding of ETS2 and NFAT Links Erk1/2 and Calcineurin Signaling in the Pathogenesis of Cardiac Hypertrophy. *Circulation* **144**, 34-51 (2021).
8. Bueno OF, et al. The MEK1-ERK1/2 signaling pathway promotes compensated cardiac hypertrophy in transgenic mice. *EMBO J* **19**, 6341-6350 (2000).
9. Molkenin JD, et al. A calcineurin-dependent transcriptional pathway for cardiac hypertrophy. *Cell* **93**, 215-228 (1998).
10. Njoroge JN, Teerlink JR. Pathophysiology and Therapeutic Approaches to Acute Decompensated Heart Failure. *Circ Res* **128**, 1468-1486 (2021).
11. Yue TL, et al. Extracellular signal-regulated kinase plays an essential role in hypertrophic agonists, endothelin-1 and phenylephrine-induced cardiomyocyte hypertrophy. *J Biol Chem* **275**, 37895-37901 (2000).

12. Nakamura M, et al. Glycogen Synthase Kinase-3alpha Promotes Fatty Acid Uptake and Lipotoxic Cardiomyopathy. *Cell Metab* **29**, 1119-1134 e1112 (2019).
13. Nakamura M, et al. Dietary carbohydrates restriction inhibits the development of cardiac hypertrophy and heart failure. *Cardiovasc Res* **117**, 2365-2376 (2021).
14. Ichida M, Finkel T. Ras regulates NFAT3 activity in cardiac myocytes. *J Biol Chem* **276**, 35243530 (2001).
15. Wang X, et al. RelB NF-kappaB represses estrogen receptor alpha expression via induction of the zinc finger protein Blimp1. *Mol Cell Biol* **29**, 3832-3844 (2009).
16. Denis GV, Yu Q, Ma P, Deeds L, Faller DV, Chen CY. Bcl-2, via its BH4 domain, blocks apoptotic signaling mediated by mitochondrial Ras. *J Biol Chem* **278**, 5775-5785 (2003).
17. Carne Trecesson S, et al. BCL-XL directly modulates RAS signalling to favour cancer cell stemness. *Nat Commun* **8**, 1123 (2017).
18. Pihán P, Carreras-Sureda A, Hetz C. BCL-2 family: integrating stress responses at the ER to control cell demise. *Cell Death & Differentiation* **24**, 1478-1487 (2017).
19. Kale J, Osterlund EJ, Andrews DW. BCL-2 family proteins: changing partners in the dance towards death. *Cell Death & Differentiation* **25**, 65-80 (2018).
20. Garcia MI, et al. Functionally redundant control of cardiac hypertrophic signaling by inositol 1,4,5-trisphosphate receptors. *J Mol Cell Cardiol* **112**, 95-103 (2017).
21. Nakayama H, et al. The IP3 receptor regulates cardiac hypertrophy in response to select stimuli. *Circ Res* **107**, 659-666 (2010).
22. Rosa N, et al. Bcl-xL acts as an inhibitor of IP3R channels, thereby antagonizing Ca(2+)-driven apoptosis. *Cell Death Differ*, (2021).
23. Rong YP, et al. Targeting Bcl-2-IP3 receptor interaction to reverse Bcl-2's inhibition of apoptotic calcium signals. *Mol Cell* **31**, 255-265 (2008).

24. Pattingre S, *et al.* Bcl-2 antiapoptotic proteins inhibit Beclin 1-dependent autophagy. *Cell* **122**, 927-939 (2005).
25. Maejima Y, *et al.* Mst1 inhibits autophagy by promoting the interaction between Beclin1 and Bcl-2. *Nat Med* **19**, 1478-1488 (2013).
26. Chen R, *et al.* Bcl-2 functionally interacts with inositol 1,4,5-trisphosphate receptors to regulate calcium release from the ER in response to inositol 1,4,5-trisphosphate. *J Cell Biol* **166**, 193-203 (2004).
27. Rong YP, *et al.* The BH4 domain of Bcl-2 inhibits ER calcium release and apoptosis by binding the regulatory and coupling domain of the IP3 receptor. *Proc Natl Acad Sci U S A* **106**, 14397-14402 (2009).
28. Monaco G, *et al.* Selective regulation of IP3-receptor-mediated Ca²⁺ signaling and apoptosis by the BH4 domain of Bcl-2 versus Bcl-Xl. *Cell Death Differ* **19**, 295-309 (2012).
29. White C, *et al.* The endoplasmic reticulum gateway to apoptosis by Bcl-X(L) modulation of the InsP3R. *Nat Cell Biol* **7**, 1021-1028 (2005).
30. Yang J, Vais H, Gu W, Foskett JK. Biphasic regulation of InsP3 receptor gating by dual Ca²⁺ release channel BH3-like domains mediates Bcl-xL control of cell viability. *Proc Natl Acad Sci U S A* **113**, E1953-1962 (2016).
31. Ikeda S, *et al.* Yes-Associated Protein (YAP) Facilitates Pressure Overload-Induced Dysfunction in the Diabetic Heart. *JACC Basic Transl Sci* **4**, 611-622 (2019).
32. Zhang J, Simpson PC, Jensen BC. Cardiac alpha1A-adrenergic receptors: emerging protective roles in cardiovascular diseases. *Am J Physiol Heart Circ Physiol* **320**, H725-H733 (2021).
33. Meguro T, *et al.* Cyclosporine attenuates pressure-overload hypertrophy in mice while enhancing susceptibility to decompensation and heart failure. *Circ Res* **84**, 735-740 (1999).
34. Purcell NH, *et al.* Genetic inhibition of cardiac ERK1/2 promotes stress-induced apoptosis and heart failure but has no effect on hypertrophy in vivo. *Proc Natl Acad Sci U S A* **104**, 14074-14079 (2007).

35. Maillet M, van Berlo JH, Molkentin JD. Molecular basis of physiological heart growth: fundamental concepts and new players. *Nat Rev Mol Cell Biol* **14**, 38-48 (2013).
36. Carlson SM, et al. Large-scale discovery of ERK2 substrates identifies ERK-mediated transcriptional regulation by ETV3. *Sci Signal* **4**, rs11 (2011).
37. Hornbeck PV, Zhang B, Murray B, Kornhauser JM, Latham V, Skrzypek E. PhosphoSitePlus, 2014: mutations, PTMs and recalibrations. *Nucleic Acids Res* **43**, D512-520 (2015).
38. Matsuda T, et al. H-Ras Isoform Mediates Protection Against Pressure Overload-Induced Cardiac Dysfunction in Part Through Activation of AKT. *Circ Heart Fail* **10**, (2017).
39. Corcoran RB, et al. Synthetic lethal interaction of combined BCL-XL and MEK inhibition promotes tumor regressions in KRAS mutant cancer models. *Cancer Cell* **23**, 121-128 (2013).
40. Ackers-Johnson M, Li PY, Holmes AP, O'Brien SM, Pavlovic D, Foo RS. A Simplified, Langendorff-Free Method for Concomitant Isolation of Viable Cardiac Myocytes and Nonmyocytes From the Adult Mouse Heart. *Circ Res* **119**, 909-920 (2016).
41. Grossman W, Jones D, McLaurin LP. Wall stress and patterns of hypertrophy in the human left ventricle. *J Clin Invest* **56**, 56-64 (1975).
42. Keller MA, Huang CY, Ivessa A, Singh S, Romanienko PJ, Nakamura M. Bcl-x short-isoform is essential for maintaining homeostasis of multiple tissues. *iScience* **26**, 106409 (2023).

ACKNOWLEDGEMENTS

We thank Daniela Zablocki for critical reading of the manuscript. This study was supported in part by U.S. Public Health Service grants HL155766 (M.N.) and HL67724, HL91469, HL102738, HL112330, HL138720, HL144626, HL150881, and AG23039 (J.S.). This work was also supported by an American Heart Association Scientist Development Grant (17SDG33660358) (M.N.) and Merit Award 20 Merit 35120374 (J.S.), and by the Fondation Leducq Transatlantic Network of Excellence 15CVD04 (J.S.). The mass spectrometry data were obtained using an Orbitrap mass spectrometer funded in part by NIH grants NS046593 and 1S10OD025047-01, for the support of proteomics research at the Rutgers Newark campus.

AUTHOR CONTRIBUTIONS

M.N. and J.S. designed the experiments and wrote the paper; M.N. and M.A.K. conducted the *in vitro* and *in vivo* experiments; M.N. and P.Z. conducted the animal experiments and analyses; N.F. and L.H.X. conducted the Ca²⁺ transient experiments; T.L. and H.L. conducted the mass spectrometry analyses; M.N. conducted the gene expression analyses; Y.T. conducted immunohistochemistry analyses; D.D.R. provided the adenovirus harboring Bcl-xL mutants and H-Ras; S.I. provided technical support and suggestions regarding cell growth and death in the Hippo pathway; M.N. and J.S. generated project resources. All authors reviewed and commented on the manuscript.

DECLARATION OF INTERESTS

The authors declare no competing interests.

Fig. 1

FIGURE LEGENDS

Fig. 1 Knock-in (KI) mice in which Serine (Ser) 14 of Bcl-xL is replaced with Alanine show worse phenotypes in response to pressure overload. (a) Representative immunoblots showing

phosphorylation of Bcl-xL at Ser14 in the heart with time course after transverse aortic constriction (TAC). Sham is a heart sample collected one hour after sham surgery. h; hours, and d; days after TAC.

Lower panel shows densitometric analysis of relative expression of pBcl-xL (Ser14)/Bcl-xL in the heart.

Kruskal-Wallis test with sham as the control. p values are shown in the figure (n=5). (b) Kaplan-Meier

survival curves. Log-rank (Mantel-Cox) test $p = 0.0006$. n=25-36. Lower panel shows the survival

curves 1 week after TAC. (c) Representative pictures of M-mode echocardiography. Yellow lines indicate left ventricular (LV) end-systolic and -diastolic diameters. Vertical scale bar indicates 5 mm

and horizontal scale bar indicates 100 ms. (d) Ejection fraction (EF) with time course after TAC or sham surgery. Two-way ANOVA with Tukey's multiple comparison test. **** $p < 0.0001$ compared to

+/+ (wild type) control. # $p < 0.05$ and ##### $p < 0.0001$ compared to knock-in/+. && $p < 0.01$ and & p

< 0.05 compared to Sham (unpaired t test or Mann-Whitney test). n=6-16 (TAC) and 5 (Sham). (e) LV

end-diastolic pressure at the indicated time points after TAC or sham surgery, evaluated by

hemodynamic study. Two-way ANOVA with Tukey's multiple comparison test. **** $p < 0.0001$. n=5.

(f) Lung weight normalized by tibia length at the indicated time points after TAC or sham surgery.

Two-way ANOVA with Tukey's multiple comparison test. **** $p < 0.0001$, *** $p < 0.001$, and ** $p <$

0.01. n=5-8. N represents biologically independent replicates. Data are mean \pm SEM.

Fig. 2

Fig. 2 Phosphorylation of Bcl-xL at Serine 14 is essential for compensatory hypertrophy in response to pressure overload. Both heterozygous and homozygous Serine (S14A) knock-in (KI) mice were used in a and e whereas homozygous mice were used in b-d and f-h. **(a)** Heart weight normalized by tibia length at the indicated time points after TAC or sham. n=5-8. **(b)** Wheat Germ Agglutinin (WGA) staining of the indicated heart tissues. Scale bar; 100 μ m. **(c)** Quantitative analysis of relative cardiomyocyte size. n=5. **(d)** Relative *NPPA*, *NPPB*, *MYH7*, and *Rcan1.4* gene expressions. *NPPA*, *NPPB*, and *MYH7*: n=6 (sham) and 9 (TAC). *Rcan1.4*: n=4. **(e)** Calculated end-diastolic wall stress of the indicated mice one-week post-TAC (n=5). **(f)** Terminal deoxynucleotidyl transferase dUTP nick end labeling (TUNEL) staining of the indicated heart tissues. Arrows indicate TUNEL-positive nuclei. Scale bar; 100 μ m. **(g)** Quantitative analysis of TUNEL-positive nuclei. n=5. **(h)** Immunoblots showing cleaved caspase 3 and 9 expression levels in the heart after TAC or sham. WT is indicated by +/+ and knock-in (KI) is indicated by KI/KI. *n* represents biologically independent replicates. Two-way ANOVA with Tukey's multiple comparison test. **** $p < 0.0001$, *** $p < 0.001$, ** $p < 0.01$, and * $p < 0.05$. Data are mean \pm SEM.

Fig. 3

Fig. 3 The Ras-MEK-ERK pathway is activated immediately after hypertrophic stimulation, phosphorylating Bcl-xL at Serine 14 and promoting nuclear translocation of NFAT3. (a) Heat map of differentially expressed genes in the hearts of WT and Bcl-xL-S14A knock-in (KI) mice 9 hours after TAC or sham surgery. n=3. (b) Gene set enrichment analysis plots of the FCεR1, ERK and Integrin pathways enriched after TAC compared to sham surgery in WT mice (Supplementary Table 1). (c) Immunoblots showing the phosphorylation status of ERK1/2 in the hearts of WT and KI mice after TAC or sham surgery. (d) Immunoblots showing the phosphorylation status of Bcl-xL (Ser14), Mst1 (Thr183), Akt (Ser473), ERK1/2 (Thr202/Tyr204), and GSK-3α/β (Ser21/9) in cardiomyocytes in response to phenylephrine (PE). (e) Relative expression of pBcl-xL (Ser14)/Bcl-xL. Kruskal-Wallis test with vehicle as the control. *p* values are shown in the figure (n=5). (f) Immunoblots showing the effect of a MEK inhibitor (PD0325901) on PE-induced Bcl-xL phosphorylation at Ser14 in cardiomyocytes. (g) Relative expression of pBcl-xL (Ser14)/Bcl-xL. Kruskal-Wallis test with vehicle as the control. *p* values are shown in the figure (n=5). (h) MS/MS spectrum of a doubly charged ion (*m/z* 646.81) corresponding to the peptide sequence ⁷ELVVDFL^pSYK¹⁶ with a phosphorylation modification at S¹⁴ in Bcl-2-like protein 1. The observed *y*- and *b*-ion series confirmed the peptide sequence and phosphorylation modification site. (i) Immunoblots showing the nuclear and cytosolic localization of NFAT3 in WT and S14A KI adult mouse cardiomyocytes transduced with adenovirus harboring H-Ras or LacZ. (j) Quantification analysis of nuclear and cytosolic expression ratios of NFAT3 (n=5). Two-way ANOVA with Tukey's multiple comparison test. **** *p* < 0.0001. (k) Immunoprecipitation assay using α-H-Ras antibody with heart lysates from mice subjected to 9 hours of pressure overload, followed by immunoblots with α-Bcl-xL antibody. The numbers indicate the ratio of Bcl-xL (upper arrow) to H-Ras (lower arrow) by densitometric analysis. (l) FLAG-pull down assay using rat neonatal ventricular cardiomyocytes transduced with adenovirus harboring FLAG-Bcl-xL for two days in the

presence of PE or vehicle for 20 mins. Immunoblots in *in vitro* experiments using rat neonatal ventricular cardiomyocytes were repeated at least three times using independently prepared cardiomyocytes unless otherwise indicated. *n* represents biologically independent replicates. Data are mean \pm SEM.

Fig. 4

Fig. 4 Wild type (WT), but not Bcl-xL-S14A knock-in (KI), cardiomyocytes exhibit increases in Ca^{2+} transient amplitude and sarcoplasmic reticulum (SR) Ca^{2+} content after TAC-induced pressure overload, which are suppressed by IP3R inhibition. (a) Hemodynamic stress differentially alters intracellular Ca^{2+} dynamics in WT and KI cardiomyocytes, as shown in the Ca^{2+} transient amplitude, SR Ca^{2+} content, fractional Ca^{2+} release, and T_{50} . Cardiomyocytes were isolated from the indicated mouse hearts 1 day after TAC (18-24 cells from 3 hearts/group). Two-way ANOVA with Tukey's multiple comparison test. **(b)** Representative traces of Ca^{2+} transient (F/F_0) with 2-APB treatment in WT and KI cardiomyocytes 1 day after TAC. Orange dotted lines indicate the level of F/F_0 at the time of 2-APB administration and arrows indicate the direction of change in F/F_0 . **(c)** Ca^{2+} transient amplitude before and after 2-APB treatment in cardiomyocytes 1 day after TAC (13-14 cells from 3 hearts/group). Paired t test (WT) or Wilcoxon matched-pairs signed rank test (KI). **(d)** Quantification of intracellular Ca^{2+} dynamics after 2-APB treatment in WT and KI cardiomyocytes 1 day after TAC. $n=13-14$ cells (Ca^{2+} transient amplitude and T_{50} , Mann-Whitney U test) and $9-10$ cells (SR Ca^{2+} content and fractional Ca^{2+} release, unpaired t test) from 3 hearts/group. **** $p < 0.0001$, *** $p < 0.001$, ** $p < 0.01$, and * $p < 0.05$. Data are mean \pm SEM.

Fig. 5

a

Biocarta_Calcineurin pathway

NES: 1.46; Nominal p value: 0.0

GO_Calcineurin-mediated signaling

NES: 1.32; Nominal p value: 0.0

WP_Calcium regulation in the cardiac cell

NES: 1.55; Nominal p value: 0.0

b

c

d

e

Fig. 5 Bcl-xL Serine (Ser) 14 phosphorylation disrupts its inhibitory interaction with IP3R, enhancing hypertrophic stimuli-induced calcineurin-NFAT signaling. (a) Gene set enrichment analysis plots of calcineurin and calcium regulation pathways enriched in WT compared to knock-in (KI) mouse hearts after 9 hours of TAC. (b) The NFAT transcriptional activity in response to phenylephrine (PE) with or without 2-APB in cardiomyocytes transfected with adenovirus harboring Bcl-xL-WT or -S14A mutant (n=6 independently prepared cardiomyocyte preparations/cultures). (c) Immunoprecipitation assay with anti-IP3R type 2 antibody using WT and KI mouse hearts 1 day after TAC. Repeated three times. (d) Flag pull-down assay using Flag-Bcl-xL-S14D or -S14A and recombinant GST-IP3R-fragment3, followed by Coomassie Brilliant Blue staining. Right panel shows quantification analysis of immunoprecipitated proteins (GST-IP3R-fragment 3 versus Flag-Bcl-xL-S14D or -S14A) (n=8 independently prepared cardiomyocyte preparations/cultures, followed by *in vitro* binding assay). (e) NFAT transcriptional activity in response to PE with or without synthetic peptide corresponding to the amino acid sequence in IP3R Fragment 3 (n=6, independently prepared cardiomyocyte preparations/cultures). Two-way ANOVA with Tukey's multiple comparison test. **** $p < 0.0001$, *** $p < 0.001$, ** $p < 0.01$, and * $p < 0.05$. Data are mean \pm SEM.

Supplementary Table 1

Biocarta pathways	SIZE	NES	NOM p-val	FDR q-val
1 BIOCARTA FCER1_PATHWAY	37	1.75	0	0.322
2 BIOCARTA ERK_PATHWAY	25	1.75	0	0.184
3 BIOCARTA IL1R_PATHWAY	26	1.74	0.087	0.138
4 BIOCARTA INTEGRIN_PATHWAY	31	1.7	0	0.126
5 BIOCARTA VIP_PATHWAY	23	1.66	0	0.138
6 BIOCARTA BCR_PATHWAY	31	1.64	0	0.144
7 BIOCARTA TFF_PATHWAY	19	1.63	0	0.13
8 BIOCARTA TCR_PATHWAY	37	1.62	0	0.126
9 BIOCARTA BARRESTIN_SRC_PATHWAY	16	1.6	0	0.149
10 BIOCARTA FMLP_PATHWAY	30	1.6	0	0.139
11 BIOCARTA HDAC_PATHWAY	22	1.59	0	0.136
12 BIOCARTA MET_PATHWAY	31	1.59	0	0.132
13 BIOCARTA IL2RB_PATHWAY	32	1.59	0	0.125
14 BIOCARTA BAD_PATHWAY	22	1.58	0	0.12
15 BIOCARTA HIVNEF_PATHWAY	52	1.57	0	0.115
16 BIOCARTA NFKB_PATHWAY	19	1.56	0.094	0.114
17 BIOCARTA GPCR_PATHWAY	28	1.53	0	0.125
18 BIOCARTA RAC1_PATHWAY	20	1.5	0	0.132
19 BIOCARTA PDGF_PATHWAY	28	1.49	0	0.131
20 BIOCARTA EIF_PATHWAY	16	1.49	0	0.13

SUPPLEMENTARY INFORMATION

Supplementary Table 1. Gene sets enriched in the heart after TAC compared to after sham

operation in WT mice. ERK1/2 were identified as significantly activated kinases in the heart during acute pressure overload. Among the top 4 gene sets enriched in TAC vs. Sham, ERK1/2 are involved in 3 Biocarta pathways, including the FCER1 pathway, ERK pathway, and Integrin pathway (as shown in bold).

Supplementary Figure 1

Supplementary Figure 1 Bcl-xL knock-in (KI) mice in which Serine (Ser) 14 of Bcl-xL is replaced with Alanine, develop more severe cardiac dysfunction in response to pressure overload than control wild type (WT) mice. (a) Representative immunoblots showing phosphorylation of Bcl-xL at Ser14 in the heart with time course after Sham. h; hours, and d; days after Sham surgery. (b) Pressure gradient between femoral artery and ascending aorta. One-way ANOVA (n=5). (c) Immunoblots showing phosphorylation of Bcl-xL at Ser14 and tubulin as a loading control in the WT and Bcl-xL (S14A) homozygous KI mouse heart one hour after TAC. Lower panel shows densitometric analysis (n=6). (d) Echocardiographic parameters of interventricular septum at end-diastole (IVSd), left ventricular (LV) posterior wall at end-diastole (LVPWd), and LV diameter at end-diastole (LVDd) and end-systole (LVDs) (n=6-16 for TAC and 5 for Sham). Two-way ANOVA with Tukey's multiple comparison test. *n* represents biologically independent replicates. **** $p < 0.0001$ and ** $p < 0.01$ compared to +/+ mice. ## $p < 0.01$ compared to KI/+ mice. &&& $p < 0.0001$, &&& $p < 0.001$, && $p < 0.01$ and & $p < 0.05$ compared to Sham (unpaired *t* test or Mann-Whitney test). (e) The percentage of TUNEL positive nuclei in the hearts of S14A KI and WT mice four weeks post-TAC. n=5. Unpaired *t* test. (f) Co-staining of WT and S14A KI mouse hearts 2 weeks after TAC with terminal deoxynucleotidyl transferase dUTP nick end labeling (TUNEL) and cardiac troponin T. Arrows indicate TUNEL-positive cardiomyocytes. Scale bar; 20 μm . Right panel shows quantitative analysis of TUNEL-positive cardiomyocytes. n=5. n represents biologically independent replicates. Data are mean \pm SEM.

Supplementary Figure 2

Supplementary Figure 2 Inhibition of the MEK-ERK pathway suppresses hypertrophy and decreases systolic function in wild type (WT) but not Bcl-xL-S14A homozygous knock-in (KI)

mice. (a) Heatmap of *NPPA* and *Myh7* gene expressions in WT and KI mouse hearts using the RNA-sequencing data (related to Fig. 3a). (b) Representative immunoblots showing pBcl-xL (Ser14) in cardiomyocytes treated with an Akt inhibitor (Triciribine) or vehicle following treatment with phenylephrine (PE) for the indicated times (minutes). Repeated three times. (c) *In vitro* kinase assay using active ERK1 and recombinant Bcl-xL-WT, S14A mutant, or GST alone, followed by blots with Bcl-xL and Bcl-xL Ser14 phospho-specific antibody. Repeated twice. (d) Immunoblots showing the expression of H-Ras and GAPDH as a loading control in adult cardiomyocytes transduced with adenovirus harboring H-Ras or LacZ. Repeated twice. (e) Representative immunoblots showing pERK1/2 and its quantification in WT mouse hearts after 9 hours of TAC. PD0325901 was used to inhibit MEK. One-way ANOVA followed by Tukey's multiple comparison test (n=5). (f) Echocardiographic parameters of the interventricular septum at end-diastole (IVSd), left ventricular (LV) posterior wall at end-diastole (LVPWd), LV diameter at end-diastole (LVDd) and end-systole (LVDs), and ejection fraction (EF) 1 week after TAC. n=7 (Vehicle) and 8 (MEK Inhibitor). (g) Heart weight (HW) normalized by tibia length (TL) and lung weight normalized by TL 1 week after TAC. n=7 (Vehicle) and 8 (MEK Inhibitor). (h) Wheat Germ Agglutinin (WGA) staining of the indicated heart tissues and quantification analysis of relative cardiomyocyte size. Scale bar; 100 μ m. n=7 (Vehicle) and 8 (MEK Inhibitor). Two-way ANOVA with Tukey's multiple comparison test unless otherwise stated. **** $p < 0.0001$, ** $p < 0.01$, and * $p < 0.05$. *n* represents biologically independent replicates unless otherwise indicated. Data are mean \pm SEM.

Supplementary Figure 3

Supplementary Figure 3 Wild type (WT) but not Bcl-xL-S14A homozygous knock-in (KI) cardiomyocytes exhibit increased Ca²⁺ transient amplitude and sarcoplasmic reticulum (SR) Ca²⁺ content after TAC-induced pressure overload. (a) Representative traces of Ca²⁺ transient (F/F_0) in WT and KI cardiomyocytes after 1 day of TAC or at baseline. **(b)** Representative traces of sarcomere shortening in cardiomyocytes isolated from WT and KI mouse hearts after two days of TAC. **(c)** Quantification analysis of sarcomere shortening (5-7 cells/heart from 2 hearts/group). **(d)** Immunoblots showing the expression of FLAG-Bcl-xL-WT and -S14A and -S14D mutants in cardiomyocytes. **(e)** Relative NFAT activity in cardiomyocytes expressing Bcl-xL-S14A, -S14D, or LacZ in response to phenylephrine (PE). (n=5 independently prepared cardiomyocyte preparations/cultures). **(f)** Phalloidin staining for determination of size of rat neonatal cardiomyocytes expressing Bcl-xL-S14A, -S14D, or LacZ in the presence or absence of PE. **(g)** Quantification analysis of cardiomyocyte size (n=5 independently prepared cardiomyocyte preparations/cultures). **(h)** Our proposed signaling pathway, involving Bcl-xL phosphorylation in hypertrophic stimuli-induced calcium-calcineurin-NFAT signaling. Phosphorylation of Bcl-xL at Ser14 by ERK disrupts its inhibitory interaction with IP3R, thereby augmenting Ca²⁺ signaling to promote compensatory hypertrophy in response to pressure overload. **(i)** Immunoblots showing the levels of autophagy-related proteins in the hearts of S14A homozygous KI and WT mice at baseline. Repeated three times. Two-way ANOVA with Tukey's multiple comparison test unless otherwise stated. **** $p < 0.0001$, *** $p < 0.001$, ** $p < 0.01$, and * $p < 0.05$. Data are mean \pm SEM.

Supplementary Figure 4

Supplementary Figure 4 Schema of our proposed mechanism. Activation of the MEK1-ERK1/2 pathway by hypertrophic stimuli promotes Bcl-xL-Ser14 phosphorylation, which disrupts its inhibitory interaction with IP3R, thereby augmenting Ca^{2+} release from the sarcoplasmic reticulum (SR) and calcineurin-NFAT signaling. This mechanism is crucial for the development of adaptive hypertrophy to suppress wall stress and prevent acute decompensated heart failure.

Supplementary Figure 5 Uncropped scan images of immunoblots membrane

Fig. 1a

Fig. 2h

Fig. 3f

Fig. 3c

Fig. 3d

Fig. 3i**Fig. 3k****Fig. 3l****Fig. 5c****Fig. 5d****Extended Data Fig. 1a****Extended Data Fig. 1c**
Extended Data Fig. 2b

Extended Data Fig. 2c

Extended Data Fig. 2d

Extended Data Fig. 2e

Extended Data Fig. 3i

Extended Data Fig. 3d

Supplementary Figure 5 Uncropped Western blot images

REVIEWER COMMENTS

Reviewer #1 (Remarks to the Author):

revisions were acceptable

Reviewer #2 (Remarks to the Author):

The authors answered some of my concerns but did not go far enough in answering all of them.

One of my major concerns was with the interpretation of a whole body knock-in phenotype. These questions could be addressed in part by extending the in vitro assays done to include looking at hypertrophy. I disagree that the in vivo data is sufficient without these in vitro experiments to confirm that the phenotype of the whole body knock-in is the result of cardiomyocyte specific signaling inhibiting hypertrophy.

An additional concern remains with proper controls for Fig 1A. I requested time matched sham controls for Figure 1. The authors added a single, non-quantified, western (supp Fig 1A) that started at 1 hour and lacked a sham zero time point control (in other words a control that was the same as the western in Fig 1A so readers could have something to compare it to). Indeed, when I look at phosphorylation signal from this western (supp Fig 1A) as a ratio of the loading control, the pattern seems to mimic perfectly the pattern of signal observed in the TAC samples(Fig 1A).

The third issue is that all of the signals they are looking at in the knock-in animals will show a survivor bias, since more than half of the animals are dead before the samples are taken. TAC surgeries (even if a “pressure gradient” is measured) have a natural variability, with some animals responding more strongly to the surgery than others. It is not clear that the decreased responses observed isn’t just due to the survivors having less relative stress response. The changes that occur already manifest in increased death by 3 days. Looking at 1 to 2 days post surgery could give a better insight into the mechanisms.

The fourth is the idea that loss of “compensated hypertrophy” is responsible for the phenotype. The authors site one study that used calcineurin inhibition to prevent hypertrophy that resulted in HF. However there was another study that demonstrated using a milder TAC surgery and calcineurin inhibition, hypertrophy was inhibited and neither HF, nor mortality occurred at any higher rate than baseline within the three week period. (Hill JA, Karimi M, Kutschke W, Davisson RL, Zimmerman K, Wang Z, Kerber RE, Weiss RM. Cardiac hypertrophy is not a required compensatory response to short-term pressure overload. *Circulation*. 2000 Jun 20;101) Indeed the authors themselves have shown that inhibition of hypertrophy is protective against heart failure (Usui S, Maejima Y, Pain J, Hong C, Cho J, Park JY, Zablocki D, Tian B, Glass DJ, Sadoshima J. Endogenous muscle atrophy F-box mediates pressure

overload-induced cardiac hypertrophy through regulation of nuclear factor-kappaB. *Circ Res.* 2011 Jul 8;109) suggesting in mice at least, you can have TAC and survive (certainly within the time frame observed in this manuscript) just fine without hypertrophy to deal with the increased demand.

What is happening in this model appears to be a rapid movement to heart failure and death. The authors see a bypassing of compensated hypertrophy, straight to decompensated hypertrophy in the surviving animals. If the quick movement to decompensated hypertrophy is driving the death, then it should be observable within 1-2 days. The interesting thing is what is driving this change. Is it an increase in cardiomyocyte cell death? Is it a change in energy dynamics? Is it even myocyte driven?

The authors did not address my concerns regarding the calcium transients.

The authors response my concerns that Inhibition of IP3R alone is unlikely to account for the observations were not convincing. As noted in my earlier review, Inhibition of IP3R has been shown to have little impact on hypertrophy. The fact that overexpressing it can induce hypertrophy and that it has some role in physiologic hypertrophy is not the point. The phenotype describe here is severe, loss of IP3R is not severe. Therefore inhibition of IP3R cannot account for the dramatic phenotype observed.

I stand by my original assessment, there are clearly some interesting findings, ERK1 phosphorylation of BCL-xl is novel, as is the relationship between the IP3R and BCL-xl. I am not convinced of the relationship between the in vivo results and the in vitro results. Specifically, the dramatic impact on cardiac contractility and mortality.

Point by point responses

Response to Reviewer #1:

Revisions were acceptable

Thank you very much for your time and invaluable suggestions.

Response to Reviewer #2:

Thank you very much for your constructive criticism and important suggestions. Below are our point-by-point responses to Reviewer 2's comments.

One of my major concerns was with the interpretation of a whole body knock-in phenotype. These questions could be addressed in part by extending the in vitro assays done to include looking at hypertrophy. I disagree that the in vivo data is sufficient without these in vitro experiments to confirm that the phenotype of the whole body knock-in is the result of cardiomyocyte specific signaling inhibiting hypertrophy.

We fully understand the importance of the reviewer's concerns regarding the interpretation of *in vivo* data obtained from systemic knock-in mice. We have conducted additional *in vitro* experiments to enhance mechanistic insights into the *in vivo* findings. Adult cardiomyocytes isolated from S14A knock-in mice showed significantly attenuated mRNA expression of *NPPA*, *MYH7*, and *Rcan1.4* in response to phenylephrine compared to those from wild-type (WT) mice (Supplementary Fig. 1g). We believe that this new experiment partially addresses the reviewer's concern and provides evidence that the effect of S14A knock-in upon

hypertrophy in cardiomyocytes may be at least partially cell-autonomous (page 6, lines 1722).

Furthermore, we investigated whether S14 phosphorylation of Bcl-xL occurs in response to hypertrophic stimuli, and, if so, whether it affects Ca^{2+} signaling in non-cardiomyocytes, using mouse neonatal cardiac fibroblasts isolated from WT mice. We found that angiotensin II (Ang II), a hypertrophic agonist, increased Ser14-phosphorylation of Bcl-xL and the cytosolic Ca^{2+} level. Interestingly, Ang II-induced increases in cytosolic Ca^{2+} were significantly lower in neonatal cardiac fibroblasts isolated from S14A knock-in mice (Supplementary Fig. 3j and 3k). These findings indicate that hypertrophic agonist-induced increases in Ca^{2+} signaling are suppressed in both cardiomyocytes and cardiac fibroblasts in S14A knock-in mice. This may in part explain the more severe TAC-induced cardiac phenotype in our systemic S14A knock-in mice compared to in cardiomyocyte-specific IP3-sponge transgenic mice (Nakayama H, *et al. Circ Res* 107, 659-666 (2010)), which the reviewer mentioned in his/her comment #4. We added discussion regarding this issue on page 15, lines 13-23.

An additional concern remains with proper controls for Fig 1A. I requested time matched sham controls for Figure 1. The authors added a single, non-quantified, western (supp Fig 1A) that started at 1 hour and lacked a sham zero time point control (in other words a control that was the same as the western in Fig 1A so readers could have something to compare it to). Indeed, when I look at phosphorylation signal from this western (supp Fig 1A) as a ratio of the loading control, the pattern seems to mimic perfectly the pattern of signal observed in the TAC samples(Fig 1A).

As per the reviewer's suggestions, we have replaced the old immunoblot with blots that include a sham zero time point control result and included the quantification analysis (Supplementary Figure 1a), which shows no significant change in the level of Ser14-phosphorylation of Bcl-xL in the heart after sham operation.

The third issue is that all of the signals they are looking at in the knock-in animals will show a survivor bias, since more than half of the animals are dead before the samples are taken. TAC

surgeries (even if a “pressure gradient” is measured) have a natural variability, with some animals responding more strongly to the surgery than others. It is not clear that the decreased responses observed isn’t just due to the survivors having less relative stress response. The changes that occur already manifest in increased death by 3 days. Looking at 1 to 2 days post surgery could give a better insight into the mechanisms.

We have obtained data from samples at early time points after hypertrophic stimulation. Our RNA-sequencing data were obtained from hearts 9 hours after TAC or sham surgery (Fig. 3 and 5). Calcium transient experiments were conducted using adult cardiomyocytes isolated from mouse hearts 1 day after TAC (Fig. 4). Sarcomere shortening in cardiomyocytes was measured using mouse hearts after 2 days of TAC (Supplementary Fig. 3).

Immunoprecipitation assays for the interaction between IP3R and Bcl-xL were conducted using mouse hearts 1 day after TAC (Fig. 5). The information, originally described in figure legends and the main text, has now been further emphasized in the main text.

The fourth is the idea that loss of “compensated hypertrophy” is responsible for the phenotype. The authors site one study that used calcineurin inhibition to prevent hypertrophy that resulted in HF. However there was another study that demonstrated using a milder TAC surgery and calcineurin inhibition, hypertrophy was inhibited and neither HF, nor mortality occurred at any higher rate than baseline within the three week period. (Hill JA, Karimi M, Kutschke W, Davisson RL, Zimmerman K, Wang Z, Kerber RE, Weiss RM. Cardiac hypertrophy is not a required compensatory response to short-term pressure overload. Circulation. 2000 Jun 20,101) Indeed the authors themselves have shown that inhibition of hypertrophy is protective against heart failure (Usui S, Maejima Y, Pain J, Hong C, Cho J, Park JY, Zablocki D, Tian B, Glass DJ, Sadoshima J. Endogenous muscle atrophy F-box mediates pressure overload-induced cardiac hypertrophy through regulation of nuclear factor-kappaB. Circ Res. 2011 Jul 8,109) suggesting in mice at least, you can have TAC and survive (certainly within the time frame observed in this manuscript) just fine without hypertrophy to deal with the increased demand. What is happening in this model appears to be a rapid movement to heart failure and death. The authors see a bypassing of compensated hypertrophy, straight to decompensated hypertrophy in the surviving animals. If the quick movement to decompensated hypertrophy is driving the death, then it should

be observable within 1-2 days. The interesting thing is what is driving this change. Is it an increase in cardiomyocyte cell death? Is it a change in energy dynamics? Is it even myocyte driven?

This is an important question. We agree with the reviewer and the current study may not exclude the possibilities raised by the reviewer as contributing mechanisms of heart failure in S14A knock-in mice. For example, in addition to the inhibition of compensatory hypertrophic responses within a few days after acute pressure overload, our calcium transient and sarcomere shortening experiments showed decreased contractility in cardiomyocytes isolated from S14A knock-in mice (Fig. 4 and supplementary Fig. 3b-c), which may also contribute to the increased mortality of the S14A knock-in mice after pressure overload. We have added discussion regarding the potential causes of the high mortality of S14A knock-in mice in response to acute pressure overload (Page 10 lines 5-8).

The authors did not address my concerns regarding the calcium transients. The authors response my concerns that Inhibition of IP3R alone is unlikely to account for the observations were not convincing. As noted in my earlier review, Inhibition of IP3R has been shown to have little impact on hypertrophy. The fact that overexpressing it can induce hypertrophy and that it has some role in physiologic hypertrophy is not the point. The phenotype describe here is severe, loss of IP3R is not severe. Therefore inhibition of IP3R cannot account for the dramatic phenotype observed. I stand by my original assessment, there are clearly some interesting findings, ERK1 phosphorylation of BCL-xl is novel, as is the relationship between the IP3R and BCL-xl. I am not convinced of the relationship between the in vivo results and the in vitro results. Specifically, the dramatic impact on cardiac contractility and mortality.

We agree with the reviewer's comments and it is important further discuss the mechanism by which systemic S14A knock-in induces "the dramatic impact on cardiac contractility and mortality." Together with the results obtained using isolated cardiomyocytes and fibroblasts from wild type and S14A knock-in mice, we offer the following discussion (page 15, lines 7-23).

Cardiac hypertrophy is required to maintain contractile function and pump sufficient blood throughout the peripheral organs in the face of increased afterload, but persistent hyperactivation of hypertrophic signaling results in cardiac dysfunction, such as aberrant activation of mTOR and YAP^{6, 13, 31}. Phenylephrine, an α 1-adrenergic receptor agonist, is generally cardioprotective in humans³². We here show that inhibition of Bcl-xL Ser14 phosphorylation suppresses the hypertrophic signaling mechanism during phenylephrine treatment *in vitro*, consistent with the notion that Bcl-xL Ser14 phosphorylation mediates adaptive cardiac hypertrophy. The Ca^{2+} -calcineurin pathway plays a central role in developing hypertrophy⁹. Inhibition of calcineurin with cyclosporine attenuates pressure overload-induced hypertrophy and enhances susceptibility to decompensation and heart failure³³, a phenotype similar to that of Bcl-xL-S14A knock-in mice. We have also shown that a lack of sufficient hypertrophy during the acute phase of pressure overload induces heart failure in heterozygous cardiac specific YAP KO mice³⁴. Since we observed decreases in contractility in single ventricular cardiomyocytes isolated from S14A knock-in mouse hearts after TAC, however, other mechanisms besides hypertrophy regulated by Bcl-xL Ser14 phosphorylation may also contribute to the failing phenotype in S14A knock-in mice during TAC. Further investigation is needed to clarify this issue.

A previous study showed that overexpression of IP3R2 in cardiomyocytes significantly enhances TAC-induced hypertrophy, whereas 2-week TAC-induced hypertrophy is not inhibited in cardiomyocyte-specific IP3-sponge transgenic mice²¹. We have shown that S14A knock-in alone induces an intracellular Ca^{2+} environment in cardiomyocytes similar to that in WT cardiomyocytes treated with 2-APB, an inhibitor of IP3R, and that disinhibition of IP3Rs with IP3R-fragment 3 rescued the Ca^{2+} -NFAT signaling defect in cardiomyocytes expressing Bcl-xL-S14A. Nevertheless, we cannot formally exclude the presence of unknown Ca^{2+} handling mechanisms regulated by Bcl-xL-Ser14 phosphorylation. In addition, negative regulation of the Ca^{2+} transient through Bcl-xL-Ser14 phosphorylation also exists in the cardiac fibroblast population. How changes in Ca^{2+} handling in cardiac fibroblasts contribute to the cardiac phenotype in response to TAC in S14A knock-in mice remain to be elucidated.

**Ser14 phosphorylation of Bcl-xL mediates compensatory cardiac hypertrophy
by stimulating calcium release from the IP3 receptor**

Michinari Nakamura^{1*}, Mariko Aoyagi Keller¹, Nadezhda Fefelova¹, Peiyong Zhai¹, Tong Liu²,
Yimin Tian¹, Shohei Ikeda¹, Dominic P Del Re¹, Hong Li², Lai-Hua Xie¹, Junichi Sadoshima^{1*}

Affiliation:

1. Department of Cell Biology and Molecular Medicine, Cardiovascular Research Institute, Rutgers-New Jersey Medical School, 185 South Orange Ave, Newark, NJ 07103
2. Center for Advanced Proteomics Research, Department of Biochemistry & Molecular Biology, Rutgers New Jersey Medical School, Newark, NJ, 07103

*Correspondence to

Department of Cell Biology and Molecular Medicine, Cardiovascular Research Institute, Rutgers-New Jersey Medical School, 185 South Orange Ave, MSB G-609, Newark, NJ 07103

+1-973-972-8619

E-mail: nakamumi@njms.rutgers.edu (M.N.), sadoshju@njms.rutgers.edu (J.S.)

The authors have declared that no conflict of interest exists.

Abstract

The anti-apoptotic function of Bcl-xL in the heart during ischemia/reperfusion is diminished by K-Ras-Mst1-mediated phosphorylation of Ser14, which allows dissociation of Bcl-xL from Bax and promotes cardiomyocyte death. Here we show that Ser14 phosphorylation of Bcl-xL is also promoted by hemodynamic stress in the heart, through the H-Ras-ERK pathway. Bcl-xL Ser14 phosphorylation-resistant knock-in mice develop less cardiac hypertrophy and exhibit contractile dysfunction and increased mortality during acute pressure overload. Bcl-xL Ser14 phosphorylation enhances the Ca²⁺ transient by blocking the inhibitory interaction between Bcl-xL and IP3Rs, thereby promoting Ca²⁺ release and activation of the calcineurin-NFAT pathway, a Ca²⁺-dependent mechanism that promotes cardiac hypertrophy. These results suggest that phosphorylation of Bcl-xL at Ser14 in response to acute pressure overload plays an essential role in mediating compensatory hypertrophy by inducing the release of Bcl-xL from IP3Rs, alleviating the negative constraint of Bcl-xL upon the IP3R-NFAT pathway.

Keywords

Bcl-xL; phosphorylation; IP3R; hypertrophy; calcium; heart failure; H-Ras, MEK1; ERK1/2

INTRODUCTION

The Bcl-2 family proteins mediate cell survival and death through apoptosis-dependent and -independent mechanisms ^{1, 2}. The anti-apoptotic property of Bcl-2 and Bcl-xL provides terminally differentiated organs, such as the heart and brain, with crucial protection against oxidative stress. We previously showed that mammalian Hippo kinase Mst1 phosphorylates Bcl-xL at Serine 14 (Ser14) in cardiomyocytes in response to oxidative stress, which disrupts its interaction with the pro-apoptotic protein Bax, thereby increasing the abundance of active Bax, apoptosis, and myocardial ischemia/reperfusion (I/R) injury in mice ^{3, 4}. In a series of intensive analyses of this signaling pathway in cardiac pathology, we found a biphasic increase of Ser14 phosphorylation in the mouse heart in response to acute pressure overload. This led us to investigate whether this post-translational modification contributes to cell survival and death in the context of hemodynamic stress in a manner similar to that during oxidative stress and, if so, how it affects cardiac structure and function. Mst1 is not activated in the heart ⁵ or the cardiomyocytes therein ³ during the acute phase of hypertrophic stimulation, raising the possibility that Bcl-xL Ser14 phosphorylation is mediated by an Mst1-independent mechanism during hypertrophy.

The heart adapts to increased blood pressure by increasing individual cardiomyocyte size, namely hypertrophy, to increase contractility and reduce ventricular wall stress and oxygen consumption ⁶. The MEK-ERK and Ca²⁺-NFAT signaling pathways have been intensively studied as crucial mechanisms for the development of cardiac hypertrophy ^{7, 8, 9}. An improper response to increased workload results in acute decompensated heart failure with high mortality, due in part to incompletely understood pathophysiology and limited available new therapies ¹⁰. It is important to unveil the missing piece(s) in hypertrophic signaling, normalization of which should prevent acute decompensated heart failure. The current study aims to investigate the functional significance and role of Bcl-xL Ser14

phosphorylation in response to pressure overload. Here we report the unexpected observation that phosphorylation of Bcl-xL at Ser14 plays a critical role in the development of compensatory hypertrophy by enhancing Ca^{2+} transients and calcineurin-NFAT signaling. We found that acute pressure overload induces phosphorylation of Bcl-xL at Ser14 through an H-Ras-ERK1/2-dependent mechanism, thereby alleviating the negative constraint of Bcl-xL upon Ca^{2+} release from IP3Rs.

RESULTS

Inhibition of endogenous Bcl-xL phosphorylation exacerbates acute decompensated heart failure in response to pressure overload

First, we examined the phosphorylation status of endogenous Bcl-xL at Ser14 in the heart in the presence of acute pressure overload by applying transverse aortic constriction (TAC) to mouse hearts. Ser14 phosphorylation was significantly increased in the heart within one hour of TAC, returning to baseline by Day 3 but increasing again at around one week and thereafter (Fig. 1a), while there was little change in the phosphorylation level in the heart after sham surgery (Supplementary Fig. 1a). In order to demonstrate the functional significance of the increased phosphorylation of Bcl-xL at Ser14, phosphorylation-resistant knock-in mice, in which Ser14 has been replaced with Ala (S14A knock-in mice)⁴, and control wild type (WT) mice were subjected to TAC surgery. Although the pressure gradient between the ascending aorta and the femoral artery two weeks after TAC tended to be less in homozygous S14A knock-in mice compared to in WT mice due to cardiac dysfunction ($p=0.09$ between WT and homozygous S14A knock-in mice, unpaired t test), there was no statistically significant difference among WT, heterozygous, and homozygous S14A knock-in mice (Supplementary Fig. 1b). Significantly less Bcl-xL Ser14 phosphorylation in S14A knock-in mouse hearts than in WT mice one hour after TAC confirms the specificity of our Bcl-xL-Ser14 phosphorylation-specific antibody

(Supplementary Fig. 1c). In contrast to the protection conferred by the Bcl-xL S14A knock-in against myocardial I/R injury⁴, S14A knock-in mice exhibited a significantly higher mortality rate after TAC than WT mice. Homozygous S14A knock-in mice exhibited the highest mortality, while heterozygous S14A knock-in mice exhibited an intermediate mortality rate (Log-rank (Mantel-Cox) test $p = 0.0006$) (Fig. 1b). Notably, most of the TAC-induced acute death in homozygous and heterozygous S14A knock-in mice was observed within the first week (Log-rank (Mantel-Cox) test $p = 0.001$). We evaluated cardiac function with echocardiography and hemodynamic analyses. There was no significant difference in cardiac function or chamber size among the groups following sham operation. However, both heterozygous and homozygous S14A knock-in mice exhibited a significantly decreased ejection fraction (EF), a measure of left ventricular contractile function, compared to WT mice, which was observed as early as one week post-TAC (Fig. 1c, 1d, and Supplementary Fig. 1d). Left ventricular end-diastolic pressure (LVEDP) and lung weight normalized by tibia length, markers of congestive heart failure, were significantly elevated in S14A knock-in mice one week after TAC (Fig. 1e and 1f). These data indicate that S14A knock-in mice develop acute decompensated heart failure in response to hemodynamic stress, and that phosphorylation of Bcl-xL at Ser 14 is an adaptive response to acute pressure overload.

Inhibition of Bcl-xL phosphorylation suppresses cardiac hypertrophy

To explore the mechanism by which pressure overload exacerbates cardiac dysfunction in S14A knock-in mice, we evaluated the degree of hypertrophy. Increases in the left ventricular wall thickness caused by cardiac hypertrophy reduce ventricular wall stress, thereby playing an adaptive role during the acute phase of pressure overload⁶. Both heterozygous and homozygous S14A knock-in mice exhibited significantly less hypertrophy than WT mice 1 to 4 weeks after TAC, as evidenced by a lower heart weight to tibia length ratio and thinner wall thickness (Fig. 2a and Supplementary Fig. 1d). Wheat Germ

Agglutinin (WGA) staining showed smaller individual cardiomyocytes in S14A knock-in mice than in WT mice (Fig. 2b and 2c). qPCR analyses showed that TAC-induced upregulation of fetal type genes, including *NPPA* and *NPPB*, *MYH7*, and *Rcan1.4*, was suppressed in S14A knock-in mice (Fig. 2d). End-diastolic left ventricular wall stress one week post-TAC was significantly greater in both heterozygous and homozygous S14A knock-in mice than in WT mice (Fig. 2e). These results suggest that the pressure overload is not properly counterbalanced with compensatory hypertrophy in both heterozygous and homozygous S14A knock-in mice. We used homozygous S14A knock-in mice for the rest of the study. Since Ser14 phosphorylation of Bcl-xL inhibits the anti-apoptotic functions of Bcl-xL during myocardial I/R³, we speculated that S14A knock-in may protect the heart against apoptosis during pressure overload. Pressure overload slightly increased apoptosis two weeks after TAC in WT mice. However, the percentage of TUNEL-positive nuclei was similar between WT and S14A knock-in mice after two weeks of TAC (Fig. 2f and 2g) and four weeks of TAC (Supplementary Fig. 1e). It is important to distinguish between the pathological roles of cardiomyocyte and non-cardiomyocyte apoptosis; thus, we further evaluated cardiomyocyte apoptosis by co-staining the heart tissue with cardiac troponin T and TUNEL two weeks after TAC. The rate of cardiomyocyte apoptosis was indistinguishable between WT and S14A knock-in mice (Supplementary Fig. 1f). Cleaved caspase 3 and 9 were slightly elevated two weeks post-TAC to a similar extent in both WT and S14A knock-in mice (Fig. 2h). We also isolated adult cardiomyocytes from WT and S14A knock-in mice and cultured them with or without phenylephrine (PE) for 24 hours. mRNA expression of *NPPA*, *MYH7*, and *Rcan1.4* in response to PE was significantly greater in WT cardiomyocytes than in those isolated from S14A knock-in mice (Supplementary Fig. 1g), suggesting that the effect of S14A knock-in upon hypertrophy may be at least partially cell-autonomous. These data suggest that Bcl-xL Ser14 phosphorylation is critical for the development of compensatory hypertrophy rather than promoting apoptosis during the acute phase of

pressure overload.

H-Ras-MEK-ERK signaling is crucial for the increase in Bcl-xL phosphorylation in response to hypertrophic stimuli, which promotes nuclear localization of NFAT3

Since Mst1 is activated in response to pressure overload on Day 3 at the earliest⁵ but phosphorylation of Bcl-xL at Ser14 is observed within one day, the early Ser14 phosphorylation of Bcl-xL in response to acute pressure overload may be independent of Mst1. In order to better understand which signaling pathway is affected in the heart during the *acute phase* of pressure overload, we performed RNA-sequencing analyses using mouse hearts subjected to 9 hours of pressure overload (Fig. 3a). As in the qPCR analyses, hypertrophy marker genes, including *NPPA* and *MYH7*, were upregulated after TAC in WT, but not S14A knock-in, mouse hearts (Supplementary Fig. 2a). The gene set enrichment analysis (GSEA) indicated that Fce receptor 1, ERK, and integrin pathways were upregulated in WT mouse hearts in response to pressure overload (Fig. 3b and Supplementary Table 1). Since these pathways all utilize the Ras-MEK-ERK pathway, a well established mechanism involved in the development of cardiac hypertrophy^{8, 11}, we hypothesized that the Ras-MEK-ERK axis may be activated to promote Bcl-xL Ser14 phosphorylation in the heart during acute pressure overload.

Consistent with the results of the RNA-sequencing analysis, ERK1/2 phosphorylation was increased but the level of total ERK1/2 protein was unaltered in the heart after 9 hours of TAC (Fig. 3c). Phosphorylation of ERK1/2, but not Mst1, was also rapidly induced in response to PE, an α 1-adrenergic receptor agonist, in neonatal rat ventricular cardiomyocytes, along with Bcl-xL Ser14 phosphorylation (Fig. 3d and 3e). Phosphorylation of Akt, a well-known regulator of cardiac hypertrophy^{6, 12, 13}, was also increased in response to PE. To examine whether activation of either ERK or Akt is involved in Bcl-xL phosphorylation at Ser14 in response to hypertrophic stimuli, cardiomyocytes were pretreated with a

MEK inhibitor, PD0325901, or an Akt inhibitor, Triciribine, and stimulated with PE. Inhibition of MEK, but not Akt, attenuated PE-induced increases in Bcl-xL phosphorylation at Ser14 (Fig. 3f, 3g, and Supplementary Fig. 2b), suggesting that the MEK-ERK axis functions upstream of Bcl-xL Ser14 phosphorylation. To determine whether ERK1 directly phosphorylates Bcl-xL at Ser14, we performed *in vitro* kinase assays with active ERK1 and recombinant GST-Bcl-xL (Supplementary Fig. 2c).

Subsequent LC-MS/MS analyses showed that ERK1 directly phosphorylates Bcl-xL at Ser14 (Fig. 3h).

Hypertrophic stimuli activate H-Ras, which, in turn, promotes nuclear translocation of NFAT3 in cardiomyocytes^{3, 14}. To investigate whether Bcl-xL phosphorylation at Ser14 is critical for activation of NFAT3, adenovirus harboring H-Ras was transduced into adult mouse cardiomyocytes isolated from WT and S14A knock-in mice (Supplementary Fig. 2d). H-Ras increased nuclear localization of NFAT3 in WT, but not S14A knock-in, cardiomyocytes (Fig. 3i and 3j). The BH4 domain of Bcl-2 family proteins interacts with H-Ras^{15, 16, 17}. Co-immunoprecipitation assays showed that binding of H-Ras to Bcl-xL was greater in hearts subjected to TAC than in those without TAC (Fig. 3k). The PE-induced increase in the interaction between Bcl-xL and H-Ras was also observed in cardiomyocytes expressing FLAG-Bcl-xL in FLAG-pull down-coimmunoprecipitation assays (Fig. 3l). These results suggest that the H-Ras-MEK-ERK pathway may promote Bcl-xL Ser14 phosphorylation, which in turn activates NFAT signaling in cardiomyocytes.

We further investigated the involvement of the H-Ras-MEK-ERK pathway in Ser14 Bcl-xL phosphorylation in the heart *in vivo*. WT and S14A knock-in mice were pretreated with a MEK inhibitor, PD0325901, or vehicle for three days. The mice were then subjected to either TAC or sham operation in the presence of PD0325901 or vehicle. Inhibition of MEK suppressed the development of hypertrophy in WT mice, as assessed by echocardiographic measurement of the wall thicknesses of the interventricular septum at end-diastole (IVSd) and the LV posterior wall at end-diastole (LVPWd), heart weight

normalized by tibia length, and individual cardiomyocyte size, after one week of TAC (Supplementary Fig. 2e-2h). The suppression of cardiac hypertrophy was accompanied by exacerbation of contractile dysfunction and heart failure, as assessed by echocardiographically measured EF and lung weight normalized by tibia length (Supplementary Fig. 2f-2g). However, inhibition of MEK failed to elicit any additive effect on the S14A knock-in phenotype after one week of TAC. These data suggest that Ser14 Bcl-xL phosphorylation and the H-Ras-MEK-ERK pathway act on the same signaling pathway, thereby mediating TAC-induced cardiac hypertrophy. Furthermore, these results suggest that the H-Ras-MEK-ERK1/2 signaling acts upstream of Ser14 Bcl-xL phosphorylation during pressure overload-induced cardiac hypertrophy.

Ser14 phosphorylation of Bcl-xL promotes Ca²⁺ signaling in response to pressure overload

We then explored the mechanism by which Ser14 Bcl-xL phosphorylation mediates cardiac hypertrophy in response to pressure overload. Since it has been suggested previously that Ca²⁺ serves as a critical second messenger to induce hypertrophy and non-canonical interaction between the BH4 domain of Bcl-xL and sarcoplasmic reticulum (SR)/endoplasmic reticulum (ER) 18, 19, we evaluated the Ca²⁺ transient in cardiomyocytes isolated from S14A knock-in and WT mice. There was no significant difference in the Ca²⁺ transient amplitude, SR Ca²⁺ content, fractional Ca²⁺ release, or T₅₀ between WT and S14A knock-in mice at baseline (Fig. 4a and Supplementary Fig. 3a). Pressure overload increased the Ca²⁺ transient amplitude in cardiomyocytes isolated from WT mice, but decreased it in cardiomyocytes from S14A mice, after one day of TAC. Pressure overload increased the SR Ca²⁺ content in cardiomyocytes isolated from WT mice, but not S14A mice. Pressure overload did not significantly affect fractional Ca²⁺ release or T₅₀ compared to sham operation in cardiomyocytes isolated from WT mice. On the other hand, a significantly decreased fractional Ca²⁺ release and increased T₅₀ were observed after TAC in

cardiomyocytes isolated from S14A knock-in mice compared to in those from WT mice (Fig. 4a and Supplementary Fig. 3a). We further assessed the contractile function of individual cardiomyocytes after two days of TAC. Cardiomyocytes isolated from S14A knock-in mice exhibited significantly reduced contraction compared to those from WT mice, consistent with the impaired Ca^{2+} signaling in S14A knock-in mice after pressure overload (Supplementary Fig. 3b and 3c). Along with suppressed compensatory hypertrophy, impaired contractility with a decrease in Ca^{2+} transient amplitude may also contribute to the decompensated heart failure and high mortality of the S14A knock-in mice during the acute phase of pressure overload.

The inositol 1,4,5-triphosphate receptor (IP3R) plays a central role in the development of cardiac hypertrophy, with functional redundancy in all 3 types of IP3Rs^{20, 21}. To examine whether IP3Rs mediate the effect of Bcl-xL Ser14 phosphorylation, 2-APB, a membrane permeable IP3R antagonist, was applied during Ca^{2+} transient measurements in cardiomyocytes (Fig. 4b). 2-APB decreased the Ca^{2+} transient amplitude in WT cardiomyocytes, but not in S14A knock-in cardiomyocytes, after one day of TAC (Fig. 4b and 4c). Although 2-APB slightly increased the Ca^{2+} transient amplitude in S14A knock-in cardiomyocytes, underlying mechanisms are unknown. The differences in Ca^{2+} transient amplitude, SR Ca^{2+} content, and fractional Ca^{2+} release between WT and the S14A knock-in cardiomyocytes after TAC were abolished in the presence of 2-APB (Fig. 4d). The difference in T50 was not abolished but became smaller in the presence of 2-APB. These results suggest that IP3R functions downstream of Bcl-xL Ser14 phosphorylation in cardiomyocytes in response to pressure overload.

Phosphorylation of Bcl-xL disrupts its inhibitory interaction with IP3R

The results presented thus far suggest that Bcl-xL Ser14 phosphorylation promotes pressure overload-induced SR Ca^{2+} release through IP3Rs. Consistent with this notion, the GSEA showed that calcineurin-

mediated signaling is significantly suppressed in S14A knock-in mouse hearts subjected to 9 hours of pressure overload compared to in WT mouse hearts with pressure overload (Fig. 5a). The calcineurin-NFAT pathway is activated in response to increases in cytosolic Ca^{2+} and required for the development of compensatory hypertrophy⁹. To validate the result of the pathway analysis, we conducted NFAT-luciferase (NFAT-Luc) assays in cardiomyocytes expressing Bcl-xL-WT, Bcl-xL-S14A (SA) or Bcl-xL-S14D (SD) (Supplementary Fig. 3d). PE significantly increased the NFAT-Luc activity, but this increase was attenuated in the presence of 2-APB in cardiomyocytes transduced with adenovirus harboring Bcl-xL-WT (Fig. 5b). The PE-induced increase in NFAT activity was significantly attenuated in cardiomyocytes transduced with adenovirus harboring Bcl-xL-S14A and was not further inhibited in the presence of 2-APB (Fig. 5b). These results suggest that Ser14 Bcl-xL phosphorylation is required for PE-induced activation of calcineurin-NFAT signaling through Ca^{2+} release from IP3Rs.

We further explored the mechanism by which Bcl-xL Ser14 phosphorylation enhances IP3R-NFAT signaling in response to hypertrophic stimuli. Immunoprecipitation assays showed that pressure overload decreases the physical interaction between Bcl-xL and IP3R-type 2 in WT mouse hearts, but not S14A knock-in mouse hearts, in the presence of TAC (Fig. 5c), suggesting that Ser14 phosphorylation during TAC inhibits Bcl-xL-IP3R interaction. Recombinant GST-IP3R-fragment 3, which is known to bind to Bcl-xL²², was pulled down more efficiently with flag-Bcl-xL-S14A than with flag-Bcl-xL-S14D, a phosphorylation-mimicking mutant (Fig. 5d). Thus, direct physical interaction between Bcl-xL and IP3R-fragment 3 is negatively regulated by Bcl-xL Ser14 phosphorylation. We further investigated the functional significance of the interaction between Bcl-xL and IP3Rs using a membrane permeable synthetic peptide harboring partial IP3R-fragment 3 (Fragment 3-P)²³. The PE-induced increase in NFAT-Luc activity in cardiomyocytes expressing wild type Bcl-xL was not significantly affected by Fragment 3-P. In contrast, although the PE-induced increase in NFAT-Luc

activity was significantly attenuated in cardiomyocytes expressing Bcl-xL-S14A, the effect of PE was fully rescued in the presence of Fragment 3-P (Fig. 5e). These data suggest that phosphorylation of Bcl-xL at Ser14 plays a critical role in stimulating calcineurin-NFAT signaling by inhibiting the physical interaction between Bcl-xL and IP3R-fragment 3, a key mechanism checking Ca^{2+} release from the SR⁹. We also investigated whether Bcl-xL S14D is sufficient to induce NFAT activation and hypertrophy or augment cardiac hypertrophy in response to PE treatment. Bcl-xL S14D neither activated NFAT nor induced or augmented hypertrophy in cardiomyocytes at baseline or in the presence of PE (Supplementary Fig. 3e-g), suggesting that overexpression of Bcl-xL S14D is not sufficient to activate NFAT or induce/enhance hypertrophy in cardiomyocytes. These results are consistent with the notion that Ser14 Bcl-xL phosphorylation mediates compensatory hypertrophy during the acute phase of pressure overload by controlling Ca^{2+} release from IP3Rs (Supplementary Fig. 3h and 4).

Although Bcl-2 family proteins regulate autophagy activity by interacting with the BH3 domain of Beclin-1^{24,25}, S14A knock-in mice showed no overt change in the levels of Beclin-1, p62, or LC3-I/II (Supplementary Fig. 3i)⁴, indicating that Bcl-xL Ser14 phosphorylation does not affect autophagy activity at baseline. To investigate whether Bcl-xL phosphorylation-mediated Ca^{2+} release takes place only in cardiomyocytes or in other types of cells in the heart as well, we isolated neonatal cardiac fibroblasts and measured cytosolic Ca^{2+} levels. WT cardiac fibroblasts exhibited increased Ser14 phosphorylation and cytosolic Ca^{2+} levels in response to angiotensin II stimulation, whereas S14A knock-in cardiac fibroblasts showed a significantly smaller elevation of cytosolic Ca^{2+} (Supplementary Fig. 3j), indicating that Bcl-xL phosphorylation is critical for enhancing Ca^{2+} release not only in cardiomyocytes but also in cardiac fibroblasts in response to hypertrophic stimuli.

DISCUSSION

We show that activation of the H-Ras-MEK1-ERK1/2 pathway by hypertrophic stimuli promotes Bcl-xL-Ser14 phosphorylation, which disrupts the inhibitory interaction between Bcl-xL and IP3Rs, thereby augmenting SR Ca²⁺ release and activating the calcineurin-NFAT pathway, a major signaling mechanism controlling cardiac hypertrophy. This mechanism is crucial for the development of adaptive hypertrophy and cardiomyocyte contraction, and suppression of this mechanism promotes acute decompensated heart failure. These results suggest that modulating the level of Bcl-xL Ser14 phosphorylation and downstream mechanisms, including the interaction between Bcl-xL and IP3R, could be a promising intervention against heart failure during acute pressure overload.

Increasing evidence indicates that Bcl-2 family proteins, including Bcl-2 and Bcl-xL, modulate Ca²⁺ signaling through a non-canonical mechanism independent of their actions upon mitochondrial outer membrane permeabilization, namely interaction with IP3Rs located on the ER/SR. The BH4 domain of Bcl-2 interacts with IP3R fragment 3, which inhibits the IP3R channel gating^{23, 26, 27}. Lysine (K) 17 in the BH4 domain of Bcl-2 plays a critical role in mediating the inhibitory interaction between Bcl-2 and IP3R fragment 3, although K17 in Bcl-2 is not conserved in the BH4 domain of Bcl-xL, which has D11 instead²⁸. In contrast to the inhibitory action of Bcl-2 on IP3R, binding of Bcl-xL to two carboxyl terminal BH3-like domains of IP3R *sensitizes* IP3R to low concentrations of IP3, thereby stimulating Ca²⁺ release, mitochondrial bioenergetics and cell survival^{29, 30}. However, Bcl-xL at high concentrations binds to the IP3R fragment 3, thereby inhibiting IP3R channel gating. Furthermore, a synthetic peptide targeting the IP3R fragment 3 completely blocks the inhibitory effect of Bcl-xL on IP3Rs³⁰, suggesting that the effect of Bcl-xL on IP3R is context-dependent. Another study also showed that Bcl-xL always inhibits, rather than activates, IP3R *in vitro*, primarily by binding to the fragment 3 as well as the ligand-binding domain of IP3Rs²². Our data indicate that Bcl-xL binds to IP3Rs and that

the disruption of this binding enhances IP3R-mediated Ca^{2+} release in the context of hypertrophic stimuli. We speculate that abundant expression of Bcl-xL in the heart may contribute to the inhibitory interaction with IP3Rs.

Importantly, we here show that the interaction between Bcl-xL and IP3Rs is regulated by Bcl-xL Ser14 phosphorylation. A prior study demonstrated that K17D mutation in the BH4 domain of Bcl-2 abrogates the ability of the BH4 domain to bind to and inhibit IP3R, whereas D11K mutation in the BH4 domain of Bcl-xL rendered the BH4 domain of Bcl-xL capable of binding to and inhibiting IP3R²⁸, suggesting the crucial role of a *negative charge* in preventing the interaction. Ser14 phosphorylation introduces a negative charge in the BH4 domain of Bcl-xL, which may contribute to disrupting the interaction with IP3R. We here show that interaction between Bcl-xL and IP3Rs and consequent activation of NFAT are negatively regulated by Bcl-xL phosphorylation at Ser14, located in the BH4 domain. Ser14 Bcl-xL phosphorylation may disrupt the interaction with IP3R either by adding a negative charge to the BH4 domain or inducing an allosterical conformational change in another part of Bcl-xL, including the BH3 domain. K17 in Bcl-2 is not conserved in Bcl-xL, nor is S14 in Bcl-xL conserved in Bcl-2. However, both K17 in Bcl-2 and S14 in Bcl-xL allow acidic posttranslational modification in the BH4 domain. Thus, posttranslational modification of K17 and S14 may allow the BH4 domains of these Bcl-2 family proteins to control Ca^{2+} signaling through IP3Rs in a regulated manner. We propose that the phosphorylation status of Ser14 in the BH4 domain of Bcl-xL could serve as a convenient access point to control cardiac contractility and hypertrophy by modulating the interaction between Bcl-xL and IP3R in the heart.

Cardiac hypertrophy is required to maintain contractile function and pump sufficient blood throughout the peripheral organs in the face of increased afterload, but persistent hyperactivation of hypertrophic signaling results in cardiac dysfunction, such as aberrant activation of mTOR and YAP^{6, 13},

³¹. Phenylephrine, an $\alpha 1$ -adrenergic receptor agonist, is generally cardioprotective in humans ³². We here show that inhibition of Bcl-xL Ser14 phosphorylation suppresses the hypertrophic signaling mechanism during phenylephrine treatment *in vitro*, consistent with the notion that Bcl-xL Ser14 phosphorylation mediates adaptive cardiac hypertrophy. The Ca^{2+} -calcineurin pathway plays a central role in developing hypertrophy ⁹. Inhibition of calcineurin with cyclosporine attenuates pressure overload-induced hypertrophy and enhances susceptibility to decompensation and heart failure ³³, a phenotype similar to that of Bcl-xL-S14A knock-in mice. We have also shown that a lack of sufficient hypertrophy during the acute phase of pressure overload induces heart failure in heterozygous cardiac specific YAP KO mice³⁴. Since we observed decreases in contractility in single ventricular cardiomyocytes isolated from S14A knock-in mouse hearts after TAC, however, other mechanisms besides hypertrophy regulated by Bcl-xL Ser14 phosphorylation may also contribute to the failing phenotype in S14A knock-in mice during TAC. Further investigation is needed to clarify this issue.

A previous study showed that overexpression of IP3R2 in cardiomyocytes significantly enhances TAC-induced hypertrophy, whereas 2-week TAC-induced hypertrophy is not inhibited in cardiomyocyte-specific IP3-sponge transgenic mice²¹. We have shown that S14A knock-in alone induces an intracellular Ca^{2+} environment in cardiomyocytes similar to that in WT cardiomyocytes treated with 2-APB, an inhibitor of IP3R, and that disinhibition of IP3Rs with IP3R-fragment 3 rescued the Ca^{2+} -NFAT signaling defect in cardiomyocytes expressing Bcl-xL-S14A. Nevertheless, we cannot formally exclude the presence of unknown Ca^{2+} handling mechanisms regulated by Bcl-xL-Ser14 phosphorylation. In addition, negative regulation of the Ca^{2+} transient through Bcl-xL-Ser14 phosphorylation also exists in the cardiac fibroblast population. How changes in Ca^{2+} handling in cardiac fibroblasts contribute to the cardiac phenotype in response to TAC in S14A knock-in mice remain to be elucidated.

Cardiac-specific overexpression of MEK1 and activation of ERK1/2 promote physiological hypertrophy, which is reversed by knockdown of ERK1/2^{8,35}, indicating the crucial role of the MEK1-ERK1/2 signaling pathway in adaptive hypertrophy³⁶. The results presented here indicate that MEK1-ERK1/2 controls cardiac hypertrophy in part through regulation of Bcl-xL phosphorylation. Although ERK1/2 are proline-directed kinases, there is no proline residue near Ser14 of Bcl-xL. However, ERK1/2 can also phosphorylate a serine/threonine residue that does not immediately precede a proline residue^{37,38}. Furthermore, a MEK inhibitor had no additive detrimental effect in S14A knock-in mice in response to TAC *in vivo*, consistent with the notion that Bcl-xL Ser14 is directly phosphorylated by ERK1/2 *in vivo*. However, the possibility that the MEK1-ERK1/2 pathway indirectly phosphorylates Bcl-xL at Ser14 through other serine/threonine kinases cannot be formally excluded. Further investigation is required to clarify this issue.

We have shown previously that Ser14 phosphorylation of Bcl-xL occurs during I/R through a K-Ras-Mst1-dependent mechanism, thereby stimulating apoptosis by promoting dissociation of Bcl-xL from Bax on the outer mitochondrial membrane^{3,4}. Here, we show that Ser14 phosphorylation of Bcl-xL during the acute phase of pressure overload occurs through an H-Ras-MEK-dependent mechanism, presumably in the ER/SR, and plays a salutary role by promoting compensatory cardiac hypertrophy. Interestingly, the level of apoptosis after pressure overload was similar between Bcl-xL-S14A knock-in and WT mice. Since both depressed cardiac function and increased LV wall stress could have induced higher levels of apoptosis, the fact that apoptosis is not increased in S14A knock-in mice suggests that a mechanism suppressing apoptosis may still be operative in these mice. Even so, it is puzzling to observe that Bcl-xL-S14A mice exhibited a more *detrimental* cardiac phenotype during acute pressure overload. One possible explanation for the discrepancy could be that regulation of the signaling complexes in which Bcl-xL takes part, its interacting partners, and its subcellular localization are distinct between

pressure overload and I/R. For example, I/R induces K-Ras-induced activation of Mst1 in mitochondria, which induces phosphorylation of Bcl-xL Ser14 and its dissociation from Bax at the mitochondrial outer membrane³. We speculate that Bcl-xL Ser14 phosphorylation by the H-Ras-MEK pathway occurs at a distinct subcellular location during the acute phase of pressure overload, namely at Bcl-xL bound to IP3R in the SR. Consistent with this hypothesis, Bcl-xL physically interacts with H-Ras (Fig. 3h). Furthermore, H-Ras and K-Ras induce distinct phenotypes in the heart in response to pressure overload³⁹. Our results also suggest that activation of the signaling mechanism mediating compensatory hypertrophy is important during the early phase of pressure overload, even if it might promote apoptosis. Whether Bcl-xL Ser14 phosphorylation is regulated during the chronic phase of heart failure and, if so, where in the cell it is regulated and how it affects cell death remain to be clarified.

We have shown previously that overexpression of Bcl-xL(S14A) inhibits dissociation of Bcl-xL from the BH3 domain of Bax during myocardial reperfusion, thereby inhibiting apoptosis⁴. Thus, by inference, Bcl-xL (S14A) may exhibit increased binding to the BH3 domain of Beclin 1, thereby inhibiting autophagy. S14A knock-in mice showed no overt change in the levels of Beclin-1, p62, or LC3-I/II, indicating no or minimal effect of Bcl-xL S14 phosphorylation on autophagy activity in the heart at baseline. However, further investigation is required to test whether Bcl-xL Ser14 phosphorylation decreases the binding of Bcl-xL to Beclin 1, thereby stimulating autophagy in stress condition, such as pressure overload.

It has been shown that the combined use of a MEK inhibitor and ABT-263, a chemical inhibitor of Bcl-xL, promotes tumor regression in KRAS mutant cancer models⁴⁰. MEK inhibitors alone may promote the anti-apoptotic actions of Bcl-xL by inhibiting Ser14 phosphorylation and stimulating Bcl-xL-Bax interaction, thereby diminishing their killing effects. We speculate that concomitant use of ABT-263 would enhance the anti-cancer effect of the MEK inhibitors by eliminating the anti-apoptotic actions

of Bcl-xL.

There are some limitations to our study. We used rat neonatal ventricular cardiomyocytes in some *in vitro* experiments to evaluate the upstream kinases that phosphorylate Bcl-xL at Ser14 and determine the effect of Bcl-xL Ser14 phosphorylation upon NFAT activity. Since cultured neonatal cardiomyocytes may not fully recapitulate the stress response that occurs in adult hearts at baseline and in response to pressure overload, further investigation is needed to elucidate the molecular mechanisms through which pressure overload leads to Bcl-xL Ser14 phosphorylation, and whether Bcl-xL Ser14 phosphorylation, in turn, regulates NFAT activity *in vivo*. Second, mitochondrial uptake of calcium released from IP3Rs is a critical determinant of cell survival, in part through regulation of apoptosis and autophagy. Thus, how the regulation of IP3R calcium release by Bcl-xL Ser14 phosphorylation affects the ER/SR-mitochondrial connection remains to be clarified. Finally, although we show that Bcl-xL Ser14 phosphorylation plays a salutary role during the acute phase of pressure overload through regulation of IP3R-mediated compensatory hypertrophy and increased contractility, considering the subcellular localization of Bcl-xL in mitochondria and the SR, contributions of additional mechanisms, including Ca²⁺ handling mechanisms and mitochondrial mechanisms, to the failing cardiac phenotype in Bcl-xL(S14A) mice cannot be excluded.

In summary, the current study demonstrates that Bcl-xL Ser14 phosphorylation is essential for adaptive hypertrophy to prevent acute decompensated heart failure, in part through IP3R-mediated Ca²⁺ release and calcineurin-NFAT signaling, during acute pressure overload.

METHODS

Mice

The Bcl-xL S14A knock-in mice (C57BL/6 background) were generated as previously described⁴. Male

C57BL/6J wild-type mice were purchased from Jackson Labs at 5-8 weeks of age. The ERK inhibitor, PD0325901 (15 mg/kg/day), or vehicle (DMSO) was administered orally once a day for 3 days before surgery and 2 days after surgery. Mice were housed in a temperature-controlled environment within a range of 21°C - 23°C with 12-hour light/dark cycles, in which they received food and water *ad libitum*. We used age-matched male mice in all animal experiments. The sample size required was estimated to be n = 5-8 per group according to the Power analysis based upon previous studies examining the effects of pressure overload on cardiac hypertrophy and hypertrophic signaling. All protocols concerning the use of animals were approved by the Institutional Animal Care and Use Committee at Rutgers New Jersey Medical School and all procedures conformed to NIH guidelines (Guide for the Care and Use of Laboratory Animals).

Cell Line

HEK293 cells were maintained at 37°C with 5% CO₂ in Dulbecco's modified Eagle's Medium with 10% fetal bovine serum.

Primary Rat Neonatal Cardiomyocytes

Primary cultures of ventricular cardiomyocytes were prepared from 1-day-old Crl:(WI)BR-Wistar rats (Envigo, Somerville) and maintained in culture as described previously¹². The neonatal rats were deeply anesthetized with isoflurane. The chest was opened, and the heart was harvested. A cardiomyocyte-rich fraction was obtained by centrifugation through a discontinuous Percoll gradient. Cardiomyocytes were cultured in complete medium containing Dulbecco's modified Eagle's medium/F-12 supplemented with 5% horse serum, 4 µg/ml transferrin, 0.7 ng/ml sodium selenite, 2 g/l bovine serum albumin (fraction V), 3 mM pyruvate, 15 mM HEPES pH 7.1, 100 µM ascorbate, 100 mg/l ampicillin, 5 mg/l linoleic acid, and 100 µM 5-bromo-2'-deoxyuridine (Sigma). Culture dishes were coated with 0.3% gelatin or 2% gelatin for immunofluorescence staining on chamber slides.

Isolation of Adult Cardiomyocytes for Signaling Experiments

Adult mouse cardiomyocytes were isolated as described previously with a modification^{12, 41}. Briefly, the heart of a male mouse was perfused with 12 ml EDTA buffer [130 mM NaCl, 5 mM KCl, 0.5 mM NaH₂PO₄, 10 mM HEPES, 10 mM Glucose, 10 mM BDM, 10 mM Taurine, 5 mM EDTA] to stop the beating of the heart. Digestion was achieved using 30 ml perfusion buffer [130 mM NaCl, 5 mM KCl, 0.5 mM NaH₂PO₄, 10 mM HEPES, 10 mM Glucose, 10 mM BDM, 10 mM Taurine, 1 mM MgCl₂] containing Collagenase type II (0.5 mg/ml) and Protease XIV (0.05 mg/ml). Cellular dissociation was stopped by addition of 5 ml Perfusion buffer containing 5% FBS and 100 mM BSA-conjugated fatty acid cocktail (palmitic acid: oleic acid: linoleic acid = 2:1:1 and BSA: fatty acid = 1:5). Cardiomyocytes and non-cardiomyocytes were separated by 4 sequential rounds of gravity settling with calcium reintroduction medium containing 100 mM BSA-conjugated fatty acid cocktail.

Transverse Aortic Constriction (TAC)

Male 8 to 10-week-old animals were subjected to TAC or sham surgery. Mice were anesthetized with pentobarbital (60-70 mg/kg, intraperitoneal injection) and mechanically ventilated with a tidal volume of 0.2 ml and a respiratory rate of 110 breaths per minute. The mice were kept warm with heat lamps. It took around 5 minutes to establish full anesthesia. The chest and neck were shaved by clipper and the skin was cleaned using betadine and 70% ethanol 3 times. Sterile ophthalmic ointment was applied to the eyes. Mice were placed in a supine position. A lack of toe pinch/tail pinch reflex was checked before making the incision. Before making the surgical incision, we subcutaneously injected a very small volume of bupivacaine along the incision line. A midline cervical incision (15-20 mm) was made to assist intubation of the trachea and for access to the intercostal space. The left chest was opened at the second intercostal space. The intercostal incision was less than 0.5 cm and opened by self-designed stretchers made of 25-gauge needles connected to rubber bands and fixed on the surgical board by pins.

With the aid of a dissecting microscope, aortic constriction was performed by ligating the transverse thoracic aorta between the innominate artery and left common carotid artery with a 28-gauge needle using a 7-0 prolene suture, and then removing the needle. Sham operation was performed without constricting the aorta. During surgery, the depth of anesthesia was monitored periodically by checking pedal reflex. Thoracotomy incision, overlying muscle layers and skin were closed in layers using 5-0 prolene sutures, and the pneumothorax was reduced. The TAC procedure was completed within 20-30 minutes per mouse. The mice were then treated with Buprenex-SR (1.0-1.2 mg/kg, SC) and monitored while being allowed to recover in a warm incubator. When recovered from anesthesia 1-2 hours after the closure of the chest, the mice were extubated and returned to their cages. Upon completion of all experimental procedures, mice were euthanized by cervical dislocation followed by harvest of the hearts for biochemical studies, including signaling pathways.

Echocardiography

Mice were anesthetized using 12 μ l/g body weight of 2.5% avertin (Sigma-Aldrich), and echocardiography was performed using ultrasound (Vivid 7, GE Healthcare). It took around 10-20 minutes from the establishment of anesthesia to the completion of echocardiography and 1-2 hours to fully recover from anesthesia after echocardiography. A 13-MHz linear ultrasound transducer was used. Mice were subjected to 2-dimension guided M-mode measurements of LV internal diameter at the papillary muscle level from the short-axis view to measure systolic function and wall thickness, which were taken from at least three beats and averaged. LV ejection fraction was calculated as follows:
$$\text{Ejection fraction} = [(LVEDD)^3 - (LVESD)^3] / (LVEDD)^3 \times 100.$$
 End-diastolic wall stress was calculated using echocardiographic and hemodynamic parameters as follows:
$$\text{End-diastolic wall stress} = \text{LV end-diastolic pressure} \times LVDd / 2 \times LVPWd \times (1 + LVPWd / 2 \times LVDd)^{42}.$$

Hemodynamic Measurements

Mice were anesthetized with Avertin (300 mg/kg, intraperitoneal injection) to measure arterial pressure gradients and LV end-diastolic pressures. The chest and neck were shaved by clipper and the skin was cleaned using betadine and 70% isopropyl alcohol three times. Mice were then placed in a supine position. The lack of pedal reflex was confirmed prior to making an incision. A small incision (5-10 mm) was made on the neck. Under a dissecting microscope, the common carotid artery was surgically isolated and clamped proximally and distally. A small incision (0.5-1 mm) was made in the carotid artery, and a high-fidelity micromanometer catheter (1.4 French; Millar Instruments Inc.) was inserted into the artery and advanced into the aorta to measure aortic pressure proximal to the constriction site and then into the LV cavity to measure LV pressure and its first derivatives. A separate high-fidelity micromanometer catheter was inserted via the femoral artery and advanced into the aorta to measure aortic pressure distal to the constriction simultaneously. During the procedure, the depth of anesthesia was monitored by checking pedal reflex periodically to ensure the surgical plane of anesthesia.

Immunoblotting

Cardiomyocyte lysates and heart homogenates were prepared in RIPA buffer containing protease and phosphatase inhibitors (Sigma-Aldrich) as described previously^{4,43}. Lysates were centrifuged at 13,200 rpm at 4°C for 15 minutes. Protein concentrations were determined using a standard BCA assay. Total protein lysates (10-30 µg) were incubated with SDS sample buffer (final concentration: 100 mM Tris (pH 6.8), 2% SDS, 5% glycerol, 2.5% 2-mercaptoethanol, and 0.05% bromophenol blue) at 95°C for 5 minutes. The denatured protein samples were separated by SDS-PAGE, transferred to polyvinylidene difluoride membranes by wet electrotransfer, blocked in either 5% (w/v) BSA or 5% (w/v) non-fat dry milk in 1xTBS/0.5% Tween 20 at room temperature for 1 hour, and probed with primary antibodies at 4°C overnight. After washing with 1xTBS/0.5% Tween 20 for 20 minutes, the membranes were incubated with the corresponding secondary antibody at room temperature for 1 hour. After washing

with 1xTBS/0.5% Tween 20 for 45 minutes, the membranes were developed with ECL Western blotting substrate, followed by acquisition of digital images with the ChemiDoc MP Imaging System (Bio-Rad). The intensities of Western blot bands were quantified using ImageJ software. Uncropped blotting images with molecular markers are provided in Supplementary Fig. 5.

Antibodies and Reagents

The following commercial antibodies were used at the indicated dilutions: rabbit monoclonal Bcl-xL antibody (#2764) (1:6,000), rabbit cleaved caspase-3 antibody (#9661) (1:2,000), rabbit cleaved caspase-9 antibody (#9507) (1:2,000), rabbit monoclonal p44/42 MAPK (Erk1/2) antibody (#9102) (1:5,000), rabbit monoclonal phospho-p44/42 MAPK (Erk1/2) (Thr202/Tyr204) antibody (#4370) (1:5,000), rabbit monoclonal phospho-GSK-3 α /3 (Ser21/9) antibody (#9323) (1:3,000), rabbit monoclonal GSK-3 α /3 antibody (#5676) (1:5,000), rabbit polyclonal phospho-Akt (Ser473) antibody (#9271) (1:4,000), rabbit polyclonal Akt antibody (#9272) (1:8,000), rabbit monoclonal phospho-MST1 (Thr183)/MST2 (Thr180) antibody (#49332) (1:1,000), rabbit monoclonal NFAT3 antibody (#2183) (1:1,000), rabbit monoclonal GAPDH antibody (#5174) (1:8,000), rabbit monoclonal Histone H3 antibody (#4499) (1:10,000), anti-mouse or -rabbit IgG, HRP-linked antibodies (#7076 and #7074) (1:5,000) (Cell Signaling Technology); α -actinin (sarcomeric) antibody (#A7811) (1:4,000), rabbit monoclonal α -tubulin antibody (T6199) (1:8,000) (Sigma-Aldrich); mouse monoclonal IP3R-II antibody (Santa Cruz Biotechnology #sc-398434) (1:1,000); rabbit polyclonal H-Ras antibody (C-20) (Santa Cruz Biotechnology #sc-520)(1:1,000); and mouse monoclonal MST1 antibody (BD Transduction Laboratories #611052) (1:4,000). For detection of phosphorylation of Bcl-xL at Ser14, a polyclonal phosphorylation-specific antibody was generated by immunizing rabbits with a phosphopeptide as described previously (1:1,000)³. Antibodies were diluted in either 5% (w/v) BSA or 5% (w/v) non-fat dry milk in 1xTBS/0.5% Tween 20, depending on the level of background intensity. The

following reagents were used: MEK inhibitor (PD0325901) and Akt inhibitor (Triciribine) (Sigma-Aldrich).

Subcellular Fractionation

Cultured neonatal rat cardiomyocytes were washed with PBS and collected with ice-cold PBS, followed by centrifugation at 600g for 5 minutes. Cardiomyocytes were then resuspended in hypotonic lysis buffer (10 mM K-HEPES pH 7.9, 1.5 mM MgCl₂, 10 mM KCl, 0.1 mM EGTA, 0.1 mM EDTA, 1% IGEPAL, 1% Phosphatase Inhibitor Cocktail, and 1% Protease Inhibitor Cocktail) and were incubated for 15 minutes on ice with intermittent pipetting. Whole-cell lysates were centrifuged at 1200g for 5 minutes. The supernatant was collected for the cytosolic fraction, and the pellets were resuspended in lysis buffer (20 mM K-HEPES, 25% Glycerol, 0.45 M NaCl, 1.5 mM MgCl₂, 1 mM EGTA, 1 mM EDTA, 1% Phosphatase Inhibitor Cocktail, and 1% Protease Inhibitor Cocktail) and were incubated for 15 minutes on ice with intermittent pipetting. The total homogenate was centrifuged at 13,000 rpm for 10 minutes to collect the nuclear fraction. The pelleted nuclei were resuspended in lysis buffer and protein content was determined for all fractions.

Immunoprecipitation

Heart samples were lysed with lysis buffer containing 50 mM Tris-HCl pH 7.4, 150 mM NaCl, 1% Triton-X 100, 1% Sodium Deoxycholate, Protease Inhibitor Cocktail (Sigma), and Phosphatase Inhibitor Cocktail (Sigma). Primary antibody was covalently immobilized on protein A/G agarose using the Pierce Crosslink Immunoprecipitation Kit according to the manufacturer's instructions (Thermo Scientific). Samples were incubated with immobilized antibody beads overnight at 4°C. After immunoprecipitation, the samples were washed with lysis buffer five times. They were then resuspended with lysis buffer and the immunoprecipitates were subjected to immunoblotting using specific primary antibodies and a conformation-specific secondary antibody (Clean-Blot IP Detection Reagent

(ThermoFisher Scientific)).

FLAG Pull-down assay

Cardiomyocytes were transduced with adenovirus harboring FLAG-Bcl-xL or LacZ for 2 days, followed by treatment with PE or vehicle for 20 minutes. The cardiomyocytes were collected with RIPA buffer containing protease and phosphatase inhibitors (Sigma-Aldrich) as described previously ⁴. Lysates were centrifuged at 13,200 rpm at 4°C for 15 minutes. After protein concentrations were determined using a standard BCA assay, protein lysates were incubated with anti-FLAG M2 Magnetic Beads (Sigma-Aldrich) overnight at 4°C. After FLAG pull-down, the samples were washed with lysis buffer five times. They were then resuspended with lysis buffer and the immunoprecipitates were subjected to immunoblotting using specific primary and secondary antibodies.

Adenovirus Constructs

Recombinant adenovirus vector for overexpression was constructed, propagated and titered as previously described ^{3, 4, 12, 43}. pBHGloxDE1,3Cre (Microbix), including the DE adenoviral genome, was co-transfected with the pDC shuttle vector containing the gene of interest into 293 cells.

Replication-defective human adenovirus type 5 (devoid of E1) harboring full length wild type or mutant Bcl-xL cDNA (Ad-Bcl-xL) or H-Ras cDNA (Ad-H-Ras) was generated by homologous recombination in 293 cells. Adenovirus harboring beta-galactosidase (Ad-LacZ) was used as a control.

Recombinant Proteins

The bacterial expression vectors for GST-fused Bcl-xL-full length-wild type (WT) and -mutant (S14A) and IP3R fragment 3 were generated by insertion of mouse Bcl-xL and IP3R fragment 3 cDNA amplified by PCR into the pCold-GST-vector. The BL21 E. coli strain was transformed with the expression vectors, grown in 3 ml LB medium containing ampicillin overnight at 37°C, and then transferred to 250 ml LB medium containing ampicillin. Protein expression was induced by addition of 1

mM isopropylthio- β -galactoside. After overnight culture at 15°C, the *E. coli* were lysed in lysis buffer (1% Triton X-100 and 1 mM DTT in PBS) with sonication. The lysate was incubated with 0.5 ml Glutathione-sepharose 4B (GE Healthcare) for 1 hour at 4°C. The sepharose was washed 3 times with 5 ml lysis buffer, and then suspended with 1 ml cleavage buffer (20 mM Tris pH 7, 150 mM NaCl, 1 mM DTT). A membrane-permeable synthetic peptide corresponding to the IP3R fragment 3 was generated by GeneScript as previously described with modification²³ using the following amino acid sequence (RKKRRQRRRGKKNVYTEIKCNLLPLDDIVRV).

In Vitro Kinase Assay

Recombinant active ERK1 was purchased from Millipore Sigma. Recombinant GST-tagged full-length Bcl-xL protein was generated using the pCold-GST-vector. Recombinant active ERK1 (10 ng, Millipore Sigma #14-439) was incubated with recombinant GST-Bcl-xL-WT (1 mg) in a kinase buffer (50 mM HEPES (pH 7.4), 15 mM MgCl₂ and 200 mM sodium vanadate) in the presence or absence of 100 mM ATP at 30°C for 15 min. Recombinant phosphorylated GST-Bcl-xL protein was separated by SDS-PAGE, followed by immunoblots with anti-Bcl-xL Ser14 phospho-specific antibody or staining with Coomassie Brilliant Blue and LC-MS/MS analysis.

Mass Spectrometry

A kinase reaction was performed using recombinant GST-tagged human Bcl-xL and recombinant active ERK1 protein. Phosphorylated protein was separated by SDS-PAGE and stained with Coomassie Brilliant Blue. The gel band of interest was excised for in-gel trypsin digestion. The resulting peptides were C18 desalted and analyzed directly by LC-MS/MS analysis on an Orbitrap Fusion Lumos MS instrument. The MS/MS spectra were searched against a Uniprot human database using the Sequest search engine on the Proteome Discoverer platform (V2.4). STY phosphorylation was set as the variable modification. The false discovery rate of protein identification is less than 1%.

In Vitro Binding Assays

Flag-Bcl-xL-S14D or -S14A protein was overexpressed using an adenovirus overexpression system in rat neonatal cardiomyocytes. Cardiomyocyte lysates were collected and Flag-tagged proteins were immunoprecipitated using Flag-agarose beads (Sigma Aldrich). Immunoprecipitated Flag-Bcl-xL-S14D or -S14A proteins were incubated with recombinant GST-fused IP3R fragment 3 in lysis buffer containing 50 mM Tris-HCl pH 7.4, 150 mM NaCl, 1% Triton-X 100, 1% Sodium Deoxycholate, Protease Inhibitor Cocktail (Sigma Aldrich), and Phosphatase Inhibitor Cocktail (Sigma Aldrich) with rotation for 1 hour at 4°C, followed by pull-down with Flag-agarose beads. After washing five times with lysis buffer, proteins were eluted with 5xSDS sample buffer, followed by SDS-PAGE and Coomassie Brilliant Blue staining.

RNA-Seq Library Preparation, Sequencing, and Data Analysis

Total RNA was isolated from mouse hearts using TRIzol (Invitrogen). Isolated RNA was checked for integrity on an Agilent Bioanalyzer 2100; samples with RNA integrity number >7.0 were used for subsequent processing. Total RNA was subjected to two rounds of poly(A) selection using oligo-d(T)25 magnetic beads (New England Biolabs). A paired-end (strand specific) cDNA library was prepared using the NEB Next Ultra-directional RNA-seq protocol. Briefly, poly(A)+ RNA was fragmented by heating at 94°C for 10 minutes, followed by reverse transcription and second strand cDNA synthesis using the reagents provided in the NEB Next kit. End-repaired cDNA was then ligated with double stranded DNA adapters, followed by purification of ligated DNA with AmpureXP beads. cDNA was then amplified by PCR for 15 cycles with a universal forward primer and a reverse primer with bar code. The sequencing of the cDNA libraries was performed on the Illumina NextSeq 500 platform (Illumina, San Diego, CA) using the single-read 1x75 cycles configuration. The raw reads files have been deposited in the NCBI Gene Expression Omnibus.

Raw reads were quality trimmed using Trimmomatic-0.39 with leading and trailing Q score 20, minimum length 35 bp. The cleaned reads were mapped to *Mus musculus* genome GRCm38 using HISAT2 (Version 2.2.1). The reference genome sequence and annotation files were downloaded from ENSEMBL release 97 (Mus_musculus.GRCm38.101fa and Mus_musculus.GRCm38.101.gtf). The aligned read counts were obtained using htseq-count as part of the package HTSeq-0.6.1. The Bioconductor package edgeR (Version 3.18.1 with limma 3.32.10) was used to perform the differential gene expression analysis under R environment, R version 4.1.1. Expression patterns of regulated genes were graphically represented in a heat map. Hierarchical clustering was performed to group genes with similar features in the expression profile. Normalized expression data were also analyzed with GSEA version 4.1.0 software using the JAVA program (Broad Institute, Cambridge, MA). All gene sets were obtained from the Molecular Signatures Database version 7.3 distributed on the GSEA Web site.

Quantitative RT-PCR

Total RNA was prepared from mouse hearts using TRIzol (Invitrogen) as previously described⁴³. cDNA was generated using 300 ng total RNA and SuperScript III Reverse Transcriptase (ThermoFisher). Using Maxima SYBR Green qPCR master mix (Fermentas), real-time RT-PCR was performed under the following conditions: 94°C for 10 minutes; 40 cycles of 94°C for 15 seconds, 58°C for 30 seconds, 72°C for 30 seconds; and a final elongation at 72°C for 15 minutes. Relative mRNA expression was determined by the $\Delta\Delta$ -Ct method normalized to the ribosomal RNA (18S) level. The following oligonucleotide primers were used: NPPA, sense 5'-ATGGGCTCCTTCTCCATCAC-3' and antisense 5'-ATCTTCGGTACCGGAAGCTG-3'; NPPB, sense 5'-AAGTCCTAGCCAGTCTCCAGA-3' and antisense 5'-GAGCTGTCTCTGGGCCATTTC-3'; MYH7, sense 5'-GCCAACACCAACCTGTCCAAGTTC-3' and antisense 5'-TGCAAAGGCTCCAGGTCTGAGGGC-3'; Rcan1.4, sense 5'-TCCAGCTTGGGCTTGACTGAG-3' and antisense 5'-

ACTGGAAGGTGGTGTCTTGT-3'; 18S rRNA, sense 5'-CGCGGTTCTATTTTGTTGGT-3' and antisense 5'-AGTCGGCATCGTTTATGGTC-3'.

Immunohistochemistry

The heart tissue was washed with PBS, fixed in 4% paraformaldehyde overnight, embedded in paraffin, and sectioned at 10- μ m thickness onto a glass slide. After de-paraffinization, sections were stained with wheat germ agglutinin (WGA) for evaluation of the cross-sectional area of cardiomyocytes, or TUNEL for evaluation of apoptosis. The outline of 100-200 myocytes was traced in each section, using ImageJ software (NIH). For co-staining with TUNEL and troponin T, the heart sections were first stained with TUNEL and washed with PBS, followed by incubation with cardiac troponin T (Invitrogen #MA5-12960) at 4°C overnight. After washing with PBS, the heart sections were incubated with Alexa 568 anti-mouse antibody at room temperature for 1 hour to visualize cardiac troponin T. The heart sections were mounted with VECTASHIELD Mounting Media with DAPI.

Reporter Gene Assay

NFAT reporter gene activity in rat neonatal cardiomyocytes was measured with a luciferase assay system (Promega). Cardiomyocytes were transfected with NFAT luciferase reporter plasmids (a gift from Dr. Toren Finkel, Addgene #10959) and Bcl-xL-WT, -S14A, or S14D in the pDC316 vector overnight (24 well plate) using LipofectAmine 2000 (Invitrogen). The NFAT reporter gene assay was performed after 4 hours of phenylephrine treatment in the presence or absence of 2-APB (25 μ M) or synthetic peptide (20 μ M). Cardiomyocytes were lysed with 50 μ l Reporter lysis buffer (24 wells each). The luminescence reaction was started by adding 5 μ l lysate to 50 μ l Reaction buffer, and luminescence was measured for 10 seconds using an OPTOCOMP I luminometer (MGM Instruments). The luminescence was normalized by protein content measured by protein assay kit (BioRad).

Cardiomyocyte cell size measurement

Rat neonatal cardiomyocytes were cultured on coverslips, washed with PBS twice, fixed with 3.7% paraformaldehyde for 15 minutes, and washed with PBS three times. Samples were permeabilized with PBST (0.5% Triton-X in PBS) for 15 min, and blocked in 5% BSA, 5% goat serum in PBST for 30 minutes at 37°C. Cardiomyocytes were stained with Alexa Fluor 555 phalloidin (Thermo Fisher Scientific, A34055). Samples were washed with PBS and mounted on glass slides with mounting medium (VECTASHIELD, Vector Laboratories). Cells were observed under a fluorescence microscope. Cell size was measured for 25 - 30 cells for each condition in each experiment and the mean value was taken as representative of the experiment. This experiment was performed independently five times (n = 5).

Isolation of ventricular cardiomyocytes for cell shortening and Ca²⁺ transient experiments

Left ventricular myocytes were enzymatically isolated from hearts of Bcl-xL S14A knock-in and control WT male mice (2-3 months). Mice were deeply anesthetized with isoflurane in a covered beaker before hearts were removed and perfused retrogradely in Langendorff fashion at 37°C with Ca²⁺-free Tyrode's solution containing 1.0 mg/ml collagenase type II (Worthington, Biochemical Corp., Lakewood, NJ, USA) and 0.1 mg/ml thermolysin (or protease type XIV, Sigma) for 10-12 min. After the enzyme solution was washed out, the hearts were transferred from the Langendorff apparatus to petri dishes. Left ventricles were gently teased apart with forceps. Ca²⁺ concentration was gradually increased to 1.0 mM. Finally, the cell suspension was filtered through a 200 µm nylon mesh. Cardiomyocytes were stored at room temperature and used within 8 hours after isolation. All electrophysiological experiments were performed at 35-37°C.

Mouse neonatal cardiac fibroblast isolation

Hearts were dissected from mice on postnatal day 1 to obtain neonatal fibroblasts. At this time, tails were also cut for genotyping. Neonatal cardiac fibroblasts were isolated by enzymatic digestion using

enzyme digestion medium, containing 1.0 mg/ml collagenase type II (Worthington, Biochemical Corp., Lakewood, NJ, USA) in 25 ml of PBS and 0.25% (wt/vol) trypsin, at 37° on a rocker for 15 min and incubation with DMEM containing 20% FBS and penicillin-streptomycin in 6-well plates.

Intracellular Ca²⁺ measurement

Ventricular myocytes were loaded with the Ca²⁺ indicator Fluo-4 AM (4 μm, Invitrogen, Grand Island, NY, USA) for 30 min. After washing and de-esterification (30 min), the myocytes were transferred to a heated chamber (37°C) on a Nikon Eclipse TE200 inverted microscope (Nikon, Tokyo, Japan) with a Fluor x40 oil objective lens (numerical aperture 1.3). The intracellular Ca²⁺ fluorescence (excitation/emission wavelengths: 485/530 nm) was recorded with a spatial resolution of 500 × 400 pixels at 50 frames per second by an iXon Charge-Coupled Device (CCD) camera (Andor Technology, Concord, MA, USA) operated with Imaging Workbench software (INDEC BioSystems, Los Altos, CA, USA). The Ca²⁺ fluorescence intensity was expressed as the ratio F/F₀ (fluorescence (F) over the baseline diastolic fluorescence (F₀)).

Measurement of single-cell shortening

Myocytes were placed in a heated chamber (37°C) on a Nikon Eclipse TE200 inverted microscope and were subjected to 1-Hz field-pacing using a stimulator (Grass Instruments, West Warwick, Rhode Island, USA). Changes in cell length were monitored by a video-based edge detection system (model VED-105, Crescent Electronics, Sandy, UT, USA) with a 30-ms time resolution. Single-cell shortening was recorded and analyzed using commercially available pCLAMP 10 software (Molecular Devices, Sunnyvale, CA). The cell shortening was calculated as a percentage of shortening from the baseline cell length in the relaxed state.

STATISTICS and REPRODUCIBILITY

All values are expressed as mean \pm SEM. Statistical analyses were carried out by 2-tailed unpaired Student *t* test for 2 groups or one- or two-way ANOVA followed by the Tukey post-hoc analysis for 3 groups or more unless otherwise stated. If the data distribution failed normality by the Shapiro-Wilk test or Kolmogorov-Smirnov test, the Mann-Whitney *U* test for 2 groups was performed. The statistical analyses used for each figure are indicated in the corresponding figure legends. Survival curves were plotted by the Kaplan-Meier method, with statistical significance analyzed by log-rank test. Statistical analyses were conducted using Prism 9 (GraphPad Software). All experiments are represented by multiple biological replicates or independent experiments. The number of replicates per experiment are indicated in the legends. All experiments were conducted using at least two independent experimental materials or cohorts to reproduce similar results. No sample was excluded from analysis. A *p* value of less than 0.05 was considered significant.

DATA AVAILABILITY

All data generated or analyzed during this study are included in this published article and its Supplementary Information. RNA-sequencing data have been deposited at GEO and are publicly available as of the date of publication (Accession numbers: GSE199705). No original code was generated in this study. Any additional information required to reanalyze the data reported in this paper is available from the lead contact upon request.

REFERENCES

1. Chong SJF, Marchi S, Petroni G, Kroemer G, Galluzzi L, Pervaiz S. Noncanonical Cell Fate Regulation by Bcl-2 Proteins. *Trends Cell Biol* **30**, 537-555 (2020).
2. Singh R, Letai A, Sarosiek K. Regulation of apoptosis in health and disease: the balancing act of BCL-2 family proteins. *Nat Rev Mol Cell Biol* **20**, 175-193 (2019).
3. Del Re DP, et al. Mst1 promotes cardiac myocyte apoptosis through phosphorylation and inhibition of Bcl-xL. *Mol Cell* **54**, 639-650 (2014).
4. Nakamura M, Zhai P, Del Re DP, Maejima Y, Sadoshima J. Mst1-mediated phosphorylation of Bcl-xL is required for myocardial reperfusion injury. *JCI Insight* **1**, e86217 (2016).
5. Ikeda S, et al. Hippo Deficiency Leads to Cardiac Dysfunction Accompanied by Cardiomyocyte Dedifferentiation During Pressure Overload. *Circ Res* **124**, 292-305 (2019).
6. Nakamura M, Sadoshima J. Mechanisms of physiological and pathological cardiac hypertrophy. *Nat Rev Cardiol* **15**, 387-407 (2018).
7. Luo Y, et al. Cooperative Binding of ETS2 and NFAT Links Erk1/2 and Calcineurin Signaling in the Pathogenesis of Cardiac Hypertrophy. *Circulation* **144**, 34-51 (2021).
8. Bueno OF, et al. The MEK1-ERK1/2 signaling pathway promotes compensated cardiac hypertrophy in transgenic mice. *EMBO J* **19**, 6341-6350 (2000).
9. Molkenkin JD, et al. A calcineurin-dependent transcriptional pathway for cardiac hypertrophy. *Cell* **93**, 215-228 (1998).
10. Njoroge JN, Teerlink JR. Pathophysiology and Therapeutic Approaches to Acute Decompensated Heart Failure. *Circ Res* **128**, 1468-1486 (2021).
11. Yue TL, et al. Extracellular signal-regulated kinase plays an essential role in hypertrophic agonists, endothelin-1 and phenylephrine-induced cardiomyocyte hypertrophy. *J Biol Chem* **275**, 37895-37901 (2000).

12. Nakamura M, et al. Glycogen Synthase Kinase-3alpha Promotes Fatty Acid Uptake and Lipotoxic Cardiomyopathy. *Cell Metab* **29**, 1119-1134 e1112 (2019).
13. Nakamura M, et al. Dietary carbohydrates restriction inhibits the development of cardiac hypertrophy and heart failure. *Cardiovasc Res* **117**, 2365-2376 (2021).
14. Ichida M, Finkel T. Ras regulates NFAT3 activity in cardiac myocytes. *J Biol Chem* **276**, 35243530 (2001).
15. Wang X, et al. RelB NF-kappaB represses estrogen receptor alpha expression via induction of the zinc finger protein Blimp1. *Mol Cell Biol* **29**, 3832-3844 (2009).
16. Denis GV, Yu Q, Ma P, Deeds L, Faller DV, Chen CY. Bcl-2, via its BH4 domain, blocks apoptotic signaling mediated by mitochondrial Ras. *J Biol Chem* **278**, 5775-5785 (2003).
17. Carne Trecesson S, et al. BCL-XL directly modulates RAS signalling to favour cancer cell stemness. *Nat Commun* **8**, 1123 (2017).
18. Pihán P, Carreras-Sureda A, Hetz C. BCL-2 family: integrating stress responses at the ER to control cell demise. *Cell Death & Differentiation* **24**, 1478-1487 (2017).
19. Kale J, Osterlund EJ, Andrews DW. BCL-2 family proteins: changing partners in the dance towards death. *Cell Death & Differentiation* **25**, 65-80 (2018).
20. Garcia MI, et al. Functionally redundant control of cardiac hypertrophic signaling by inositol 1,4,5-trisphosphate receptors. *J Mol Cell Cardiol* **112**, 95-103 (2017).
21. Nakayama H, et al. The IP3 receptor regulates cardiac hypertrophy in response to select stimuli. *Circ Res* **107**, 659-666 (2010).
22. Rosa N, et al. Bcl-xL acts as an inhibitor of IP3R channels, thereby antagonizing Ca(2+)-driven apoptosis. *Cell Death Differ*, **29**, 788-805 (2022).
23. Rong YP, et al. Targeting Bcl-2-IP3 receptor interaction to reverse Bcl-2's inhibition of apoptotic calcium signals. *Mol Cell* **31**, 255-265 (2008).

24. Pattingre S, et al. Bcl-2 antiapoptotic proteins inhibit Beclin 1-dependent autophagy. *Cell* **122**, 927-939 (2005).
25. Maejima Y, et al. Mst1 inhibits autophagy by promoting the interaction between Beclin1 and Bcl-2. *Nat Med* **19**, 1478-1488 (2013).
26. Chen R, et al. Bcl-2 functionally interacts with inositol 1,4,5-trisphosphate receptors to regulate calcium release from the ER in response to inositol 1,4,5-trisphosphate. *J Cell Biol* **166**, 193-203 (2004).
27. Rong YP, et al. The BH4 domain of Bcl-2 inhibits ER calcium release and apoptosis by binding the regulatory and coupling domain of the IP3 receptor. *Proc Natl Acad Sci U S A* **106**, 14397-14402 (2009).
28. Monaco G, et al. Selective regulation of IP3-receptor-mediated Ca²⁺ signaling and apoptosis by the BH4 domain of Bcl-2 versus Bcl-Xl. *Cell Death Differ* **19**, 295-309 (2012).
29. White C, et al. The endoplasmic reticulum gateway to apoptosis by Bcl-X(L) modulation of the InsP3R. *Nat Cell Biol* **7**, 1021-1028 (2005).
30. Yang J, Vais H, Gu W, Foskett JK. Biphasic regulation of InsP3 receptor gating by dual Ca²⁺ release channel BH3-like domains mediates Bcl-xL control of cell viability. *Proc Natl Acad Sci U S A* **113**, E1953-1962 (2016).
31. Ikeda S, et al. Yes-Associated Protein (YAP) Facilitates Pressure Overload-Induced Dysfunction in the Diabetic Heart. *JACC Basic Transl Sci* **4**, 611-622 (2019).
32. Zhang J, Simpson PC, Jensen BC. Cardiac alpha1A-adrenergic receptors: emerging protective roles in cardiovascular diseases. *Am J Physiol Heart Circ Physiol* **320**, H725-H733 (2021).
33. Meguro T, et al. Cyclosporine attenuates pressure-overload hypertrophy in mice while enhancing susceptibility to decompensation and heart failure. *Circ Res* **84**, 735-740 (1999).
34. Kashihara T, et al. YAP mediates compensatory cardiac hypertrophy through aerobic glycolysis in response to pressure overload. *J Clin Invest* **132**, e150595 (2022).

35. Purcell NH, et al. Genetic inhibition of cardiac ERK1/2 promotes stress-induced apoptosis and heart failure but has no effect on hypertrophy in vivo. *Proc Natl Acad Sci U S A* **104**, 1407414079 (2007).
36. Maillet M, van Berlo JH, Molkentin JD. Molecular basis of physiological heart growth: fundamental concepts and new players. *Nat Rev Mol Cell Biol* **14**, 38-48 (2013).
37. Carlson SM, et al. Large-scale discovery of ERK2 substrates identifies ERK-mediated transcriptional regulation by ETV3. *Sci Signal* **4**, rs11 (2011).
38. Hornbeck PV, Zhang B, Murray B, Kornhauser JM, Latham V, Skrzypek E. PhosphoSitePlus, 2014: mutations, PTMs and recalibrations. *Nucleic Acids Res* **43**, D512-520 (2015).
39. Matsuda T, et al. H-Ras Isoform Mediates Protection Against Pressure Overload-Induced Cardiac Dysfunction in Part Through Activation of AKT. *Circ Heart Fail* **10**, e003658 (2017).
40. Corcoran RB, et al. Synthetic lethal interaction of combined BCL-XL and MEK inhibition promotes tumor regressions in KRAS mutant cancer models. *Cancer Cell* **23**, 121-128 (2013).
41. Ackers-Johnson M, Li PY, Holmes AP, O'Brien SM, Pavlovic D, Foo RS. A Simplified, Langendorff-Free Method for Concomitant Isolation of Viable Cardiac Myocytes and Nonmyocytes From the Adult Mouse Heart. *Circ Res* **119**, 909-920 (2016).
42. Grossman W, Jones D, McLaurin LP. Wall stress and patterns of hypertrophy in the human left ventricle. *J Clin Invest* **56**, 56-64 (1975).
43. Keller MA, Huang CY, Ivessa A, Singh S, Romanienko PJ, Nakamura M. Bcl-x short-isoform is essential for maintaining homeostasis of multiple tissues. *iScience* **26**, 106409 (2023).

ACKNOWLEDGEMENTS

We thank Daniela Zablocki for critical reading of the manuscript. This study was supported in part by U.S. Public Health Service grants HL155766 (M.N.) and HL67724, HL91469, HL102738, HL112330, HL138720, HL144626, HL150881, and AG23039 (J.S.). This work was also supported by an American Heart Association Scientist Development Grant (17SDG33660358) (M.N.) and Merit Award 20 Merit 35120374 (J.S.), and by the Fondation Leducq Transatlantic Network of Excellence 15CVD04 (J.S.). The mass spectrometry data were obtained using an Orbitrap mass spectrometer funded in part by NIH grants NS046593 and 1S10OD025047-01, for the support of proteomics research at the Rutgers Newark campus.

AUTHOR CONTRIBUTIONS

M.N. and J.S. designed the experiments and wrote the paper; M.N. and M.A.K. conducted the *in vitro* and *in vivo* experiments; M.N. and P.Z. conducted the animal experiments and analyses; N.F. and L.H.X. conducted the Ca²⁺ transient experiments; T.L. and H.L. conducted the mass spectrometry analyses; M.N. conducted the gene expression analyses; Y.T. conducted immunohistochemistry analyses; D.D.R. provided the adenovirus harboring Bcl-xL mutants and H-Ras; S.I. provided technical support and suggestions regarding cell growth and death in the Hippo pathway; M.N. and J.S. generated project resources. All authors reviewed and commented on the manuscript.

DECLARATION OF INTERESTS

The authors declare no competing interests.

Fig. 1

FIGURE LEGENDS

Fig. 1 Knock-in (KI) mice in which Serine (Ser) 14 of Bcl-xL is replaced with Alanine show worse phenotypes in response to pressure overload. (a) Representative immunoblots showing phosphorylation of Bcl-xL at Ser14 in the heart with time course after transverse aortic constriction (TAC). Sham is a heart sample collected one hour after sham surgery. h; hours, and d; days after TAC. Lower panel shows densitometric analysis of relative expression of pBcl-xL (Ser14)/Bcl-xL in the heart. Kruskal-Wallis test with sham as the control. p values are shown in the figure (n=5). (b) Kaplan-Meier survival curves. Log-rank (Mantel-Cox) test $p = 0.0006$. n=25-36. Lower panel shows the survival curves 1 week after TAC. (c) Representative pictures of M-mode echocardiography. Yellow lines indicate left ventricular (LV) end-systolic and -diastolic diameters. Vertical scale bar indicates 5 mm and horizontal scale bar indicates 100 ms. (d) Ejection fraction (EF) with time course after TAC or sham surgery. Two-way ANOVA with Tukey's multiple comparison test. **** $p < 0.0001$ compared to +/+ (wild type) control. # $p < 0.05$ and ##### $p < 0.0001$ compared to knock-in/+. &&& $p < 0.0001$, &&& $p < 0.001$, && $p < 0.01$ and & $p < 0.05$ compared to Sham (unpaired t test or Mann-Whitney test). n=616 (TAC) and 5 (Sham). (e) LV end-diastolic pressure at the indicated time points after TAC or sham surgery, evaluated by hemodynamic study. Two-way ANOVA with Tukey's multiple comparison test. **** $p < 0.0001$. n=5. (f) Lung weight normalized by tibia length at the indicated time points after TAC or sham surgery. Two-way ANOVA with Tukey's multiple comparison test. **** $p < 0.0001$, *** $p < 0.001$, and ** $p < 0.01$. n=5-8. N represents biologically independent replicates. Data are mean \pm SEM.

Fig. 2

Fig. 2 Phosphorylation of Bcl-xL at Serine 14 is essential for compensatory hypertrophy in response to pressure overload. Both heterozygous and homozygous Serine (S14A) knock-in (KI) mice were used in a and e whereas homozygous mice were used in b-d and f-h. **(a)** Heart weight normalized by tibia length at the indicated time points after TAC or sham. n=5-8. **(b)** Wheat Germ Agglutinin (WGA) staining of the indicated heart tissues. Scale bar; 100 μm . **(c)** Quantitative analysis of relative cardiomyocyte size. n=5. **(d)** Relative *NPPA*, *NPPB*, *MYH7*, and *Rcan1.4* gene expressions. *NPPA*, *NPPB*, and *MYH7*: n=6 (sham) and 9 (TAC). *Rcan1.4*: n=4. **(e)** Calculated end-diastolic wall stress of the indicated mice one-week post-TAC (n=5). **(f)** Terminal deoxynucleotidyl transferase dUTP nick end labeling (TUNEL) staining of the indicated heart tissues. Arrows indicate TUNEL-positive nuclei. Scale bar; 100 μm . **(g)** Quantitative analysis of TUNEL-positive nuclei. n=5. **(h)** Immunoblots showing cleaved caspase 3 and 9 expression levels in the heart after TAC or sham. In all graphs, WT is indicated by +/+ and knock-in (KI) is indicated by KI/+ (heterozygous) or KI/KI (homozygous). *n* represents biologically independent replicates. Two-way ANOVA with Tukey's multiple comparison test. **** $p < 0.0001$, *** $p < 0.001$, ** $p < 0.01$, * $p < 0.05$, ns not significant. Data are mean \pm SEM.

Fig. 3

Fig. 3 The Ras-MEK-ERK pathway is activated immediately after hypertrophic stimulation, phosphorylating Bcl-xL at Serine 14 and promoting nuclear translocation of NFAT3. (a) Heat map of differentially expressed genes in the hearts of WT and Bcl-xL-S14A knock-in (KI) mice 9 hours after TAC or sham surgery. n=3. (b) Gene set enrichment analysis plots of the FcεR1, ERK and Integrin pathways enriched after TAC compared to sham surgery in WT mice (Supplementary Table 1). (c) Immunoblots showing the phosphorylation status of ERK1/2 in the hearts of WT and KI mice after TAC or sham surgery. (d) Immunoblots showing the phosphorylation status of Bcl-xL (Ser14), Mst1 (Thr183), Akt (Ser473), ERK1/2 (Thr202/Tyr204), and GSK-3α/β (Ser21/9) in cardiomyocytes in response to phenylephrine (PE). (e) Relative expression of pBcl-xL (Ser14)/Bcl-xL. Kruskal-Wallis test with vehicle as the control. *p* values are shown in the figure (n=5). (f) Immunoblots showing the effect of a MEK inhibitor (PD0325901) on PE-induced Bcl-xL phosphorylation at Ser14 in cardiomyocytes. (g) Relative expression of pBcl-xL (Ser14)/Bcl-xL. Kruskal-Wallis test with vehicle as the control. *p* values are shown in the figure (n=5). (h) MS/MS spectrum of a doubly charged ion (*m/z* 646.81) corresponding to the peptide sequence ⁷ELVVDFLpSYK¹⁶ with a phosphorylation modification at S¹⁴ in Bcl-2-like protein 1. The observed *y*- and *b*-ion series confirmed the peptide sequence and phosphorylation modification site. (i) Immunoblots showing the nuclear and cytosolic localization of NFAT3 in WT and S14A KI adult mouse cardiomyocytes transduced with adenovirus harboring H-Ras or LacZ. (j) Quantification analysis of nuclear and cytosolic expression ratios of NFAT3 (n=5). Two-way ANOVA with Tukey's multiple comparison test. **** *p* < 0.0001, ns not significant. (k) Immunoprecipitation assay using α-H-Ras antibody with heart lysates from mice subjected to 9 hours of pressure overload, followed by immunoblots with α-Bcl-xL antibody. The numbers indicate the ratio of Bcl-xL (upper arrow) to H-Ras (lower arrow) by densitometric analysis. (l) FLAG-pull down assay using rat neonatal ventricular cardiomyocytes transduced with adenovirus harboring FLAG-Bcl-xL for

two days in the presence of PE or vehicle for 20 mins. Immunoblots in *in vitro* experiments using rat neonatal ventricular cardiomyocytes were repeated at least three times using independently prepared cardiomyocytes unless otherwise indicated. *n* represents biologically independent replicates. Data are mean \pm SEM.

Fig. 4

Fig. 4 Wild type (WT), but not Bcl-xL-S14A knock-in (KI), cardiomyocytes exhibit increases in Ca²⁺ transient amplitude and sarcoplasmic reticulum (SR) Ca²⁺ content after TAC-induced pressure overload, which are suppressed by IP3R inhibition. (a) Hemodynamic stress differentially alters intracellular Ca²⁺ dynamics in WT and KI cardiomyocytes, as shown in the Ca²⁺ transient amplitude, SR Ca²⁺ content, fractional Ca²⁺ release, and T₅₀. Cardiomyocytes were isolated from the indicated mouse hearts 1 day after TAC (18-24 cells from 3 hearts/group). Two-way ANOVA with Tukey's multiple comparison test. (b) Representative traces of Ca²⁺ transient (*F/F0*) with 2-APB treatment in WT and KI cardiomyocytes 1 day after TAC. Orange dotted lines indicate the level of *F/F0* at the time of 2-APB administration and arrows indicate the direction of change in *F/F0*. (c) Ca²⁺ transient amplitude before and after 2-APB treatment in cardiomyocytes 1 day after TAC (13-14 cells from 3 hearts/group). Paired *t* test (WT) or Wilcoxon matched-pairs signed rank test (KI). (d) Quantification of intracellular Ca²⁺ dynamics after 2-APB treatment in WT and KI cardiomyocytes 1 day after TAC. n=13-14 cells (Ca²⁺ transient amplitude and T₅₀, Mann-Whitney *U* test) and 9-10 cells (SR Ca²⁺ content and fractional Ca²⁺ release, unpaired *t* test) from 3 hearts/group. **** *p* < 0.0001, *** *p* < 0.001, ** *p* < 0.01, * *p* < 0.05, ns not significant. Data are mean ± SEM.

Fig. 5

a

Biocarta_Calcineurin pathway

NES: 1.46; Nominal p value: 0.0

GO_Calcineurin-mediated signaling

NES: 1.32; Nominal p value: 0.0

WP_Calcium regulation in the cardiac cell

NES: 1.55; Nominal p value: 0.0

b

c

d

e

Fig. 5 Bcl-xL Serine (Ser) 14 phosphorylation disrupts its inhibitory interaction with IP3R, enhancing hypertrophic stimuli-induced calcineurin-NFAT signaling. (a) Gene set enrichment analysis plots of calcineurin and calcium regulation pathways enriched in WT compared to knock-in (KI) mouse hearts after 9 hours of TAC. (b) The relative NFAT transcriptional activity in response to phenylephrine (PE) with or without 2-APB in cardiomyocytes transfected with adenovirus harboring Bcl-xL-WT or -S14A mutant (n=6 independently prepared cardiomyocyte preparations/cultures). (c) Immunoprecipitation assay with anti-IP3R type 2 antibody using WT and KI mouse hearts 1 day after TAC. Repeated three times. (d) Flag pull-down assay using Flag-Bcl-xL-S14D or -S14A and recombinant GST-IP3R-fragment3, followed by Coomassie Brilliant Blue staining. Right panel shows quantification analysis of immunoprecipitated proteins (GST-IP3R-fragment 3 versus Flag-Bcl-xL-S14D or -S14A) (n=8 independently prepared cardiomyocyte preparations/cultures, followed by *in vitro* binding assay). (e) The relative NFAT transcriptional activity in response to PE with or without synthetic peptide corresponding to the amino acid sequence in IP3R Fragment 3 (n=6, independently prepared cardiomyocyte preparations/cultures). Two-way ANOVA with Tukey's multiple comparison test. **** $p < 0.0001$, *** $p < 0.001$, ** $p < 0.01$, * $p < 0.05$, ns not significant. Data are mean \pm SEM.

Supplementary Table 1

Biocarta pathways	SIZE	NES	NOM p-val	FDR q-val
1 BIOCARTA FCER1_PATHWAY	37	1.75	0	0.322
2 BIOCARTA ERK_PATHWAY	25	1.75	0	0.184
3 BIOCARTA IL1R_PATHWAY	26	1.74	0.087	0.138
4 BIOCARTA INTEGRIN_PATHWAY	31	1.7	0	0.126
5 BIOCARTA VIP_PATHWAY	23	1.66	0	0.138
6 BIOCARTA BCR_PATHWAY	31	1.64	0	0.144
7 BIOCARTA TFF_PATHWAY	19	1.63	0	0.13
8 BIOCARTA TCR_PATHWAY	37	1.62	0	0.126
9 BIOCARTA BARRESTIN_SRC_PATHWAY	16	1.6	0	0.149
10 BIOCARTA FMLP_PATHWAY	30	1.6	0	0.139
11 BIOCARTA HDAC_PATHWAY	22	1.59	0	0.136
12 BIOCARTA MET_PATHWAY	31	1.59	0	0.132
13 BIOCARTA IL2RB_PATHWAY	32	1.59	0	0.125
14 BIOCARTA BAD_PATHWAY	22	1.58	0	0.12
15 BIOCARTA HIVNEF_PATHWAY	52	1.57	0	0.115
16 BIOCARTA NFKB_PATHWAY	19	1.56	0.094	0.114
17 BIOCARTA GPCR_PATHWAY	28	1.53	0	0.125
18 BIOCARTA RAC1_PATHWAY	20	1.5	0	0.132
19 BIOCARTA PDGF_PATHWAY	28	1.49	0	0.131
20 BIOCARTA EIF_PATHWAY	16	1.49	0	0.13

SUPPLEMENTARY INFORMATION

Supplementary Table 1. Gene sets enriched in the heart after TAC compared to after sham

operation in WT mice. ERK1/2 were identified as significantly activated kinases in the heart during acute pressure overload. Among the top 4 gene sets enriched in TAC vs. Sham, ERK1/2 are involved in 3 Biocarta pathways, including the FCER1 pathway, ERK pathway, and Integrin pathway (as shown in bold).

Supplementary Figure 1

Supplementary Figure 1 Bcl-xL knock-in (KI) mice in which Serine (Ser) 14 of Bcl-xL is replaced with Alanine, develop more severe cardiac dysfunction in response to pressure overload than control wild type (WT) mice. (a) Representative immunoblots showing phosphorylation of Bcl-xL at Ser14 in the heart with time course after Sham. h; hours, and d; days after Sham surgery. Lower panel shows densitometric analysis of relative expression of pBcl-xL (Ser14)/Bcl-xL in the heart. Kruskal-Wallis test with sham 0 h as the control. p values are shown in the figure (n=4). (b) Pressure gradient between femoral artery and ascending aorta. One-way ANOVA (n=5). (c) Immunoblots showing phosphorylation of Bcl-xL at Ser14 and tubulin as a loading control in the WT and Bcl-xL (S14A) homozygous KI mouse heart one hour after TAC. Lower panel shows densitometric analysis (n=6). (d) Echocardiographic parameters of interventricular septum at end-diastole (IVSd), left ventricular (LV) posterior wall at end-diastole (LVPWd), and LV diameter at end-diastole (LVDd) and end-systole (LVDs) (n=6-16 for TAC and 5 for Sham). Two-way ANOVA with Tukey's multiple comparison test. n represents biologically independent replicates. **** $p < 0.0001$ and ** $p < 0.01$ compared to +/+ mice. ## $p < 0.01$ compared to KI/+ mice. &&& $p < 0.0001$, &&& $p < 0.001$, && $p < 0.01$ and & $p < 0.05$ compared to Sham (unpaired t test or Mann-Whitney test). (e) The percentage of TUNEL positive nuclei in the hearts of S14A KI and WT mice four weeks post-TAC. n=5. Unpaired t test. ns not significant. (f) Co-staining of WT and S14A KI mouse hearts 2 weeks after TAC with terminal deoxynucleotidyl transferase dUTP nick end labeling (TUNEL) and cardiac troponin T. Arrows indicate TUNEL-positive cardiomyocytes. Scale bar; 20 μm . Right panel shows quantitative analysis of TUNEL-positive cardiomyocytes. n=5. ns not significant. (g) Relative *NPPA*, *MYH7*, and *Rcan1.4* gene expressions in response to 100 μM phenylephrine (PE) or vehicle for 24 h in adult cardiomyocytes isolated from WT and homozygous S14A KI mice (n=4). Two-way ANOVA with Tukey's multiple comparison test. n represents biologically independent replicates. Data are mean \pm SEM.

Supplementary Figure 2

Supplementary Figure 2 Inhibition of the MEK-ERK pathway suppresses hypertrophy and decreases systolic function in wild type (WT) but not Bcl-xL-S14A homozygous knock-in (KI) mice. (a) Heatmap of *NPPA* and *Myh7* gene expressions in WT and KI mouse hearts using the RNA-sequencing data (related to Fig. 3a). (b) Representative immunoblots showing pBcl-xL (Ser14) in cardiomyocytes treated with an Akt inhibitor (Triciribine) or vehicle following treatment with phenylephrine (PE) for the indicated times (minutes). Repeated three times. (c) *In vitro* kinase assay using active ERK1 and recombinant Bcl-xL-WT, S14A mutant, or GST alone, followed by blots with Bcl-xL and Bcl-xL Ser14 phospho-specific antibody. Repeated twice. (d) Immunoblots showing the expression of H-Ras and GAPDH as a loading control in adult cardiomyocytes transduced with adenovirus harboring H-Ras or LacZ. Repeated twice. (e) Representative immunoblots showing pERK1/2 and its quantification in WT mouse hearts after 9 hours of TAC. PD0325901 was used to inhibit MEK. One-way ANOVA followed by Tukey's multiple comparison test (n=5). (f) Echocardiographic parameters of the interventricular septum at end-diastole (IVSd), left ventricular (LV) posterior wall at end-diastole (LVPWd), LV diameter at end-diastole (LVDd) and end-systole (LVDs), and ejection fraction (EF) 1 week after TAC. n=7 (Vehicle) and 8 (MEK Inhibitor). (g) Heart weight (HW) normalized by tibia length (TL) and lung weight normalized by TL 1 week after TAC. n=7 (Vehicle) and 8 (MEK Inhibitor). (h) Wheat Germ Agglutinin (WGA) staining of the indicated heart tissues and quantification analysis of relative cardiomyocyte size. Scale bar; 100 μ m. n=7 (Vehicle) and 8 (MEK Inhibitor). Two-way ANOVA with Tukey's multiple comparison test unless otherwise stated. In all graphs, **** $p < 0.0001$, ** $p < 0.01$, * $p < 0.05$, ns not significant. *n* represents biologically independent replicates unless otherwise indicated. Data are mean \pm SEM.

Supplementary Figure 3

Supplementary Figure 3 Wild type (WT) but not Bcl-xL-S14A homozygous knock-in (KI) cardiomyocytes exhibit increased Ca^{2+} transient amplitude and sarcoplasmic reticulum (SR) Ca^{2+} content after TAC-induced pressure overload. (a) Representative traces of Ca^{2+} transient (F/F_0) in WT and KI cardiomyocytes after 1 day of TAC or at baseline. **(b)** Representative traces of sarcomere shortening in cardiomyocytes isolated from WT and KI mouse hearts after two days of TAC. **(c)** Quantification analysis of sarcomere shortening (5-7 cells/heart from 2 hearts/group). Unpaired t test. **(d)** Immunoblots showing the expression of FLAG-Bcl-xL-WT and -S14A and -S14D mutants in cardiomyocytes. Repeated three times. **(e)** Relative NFAT activity in cardiomyocytes expressing Bcl-xL-S14A, -S14D, or LacZ in response to phenylephrine (PE). (n=5 independently prepared cardiomyocyte preparations/cultures). **(f)** Phalloidin staining for determination of size of rat neonatal cardiomyocytes expressing Bcl-xL-S14A, -S14D, or LacZ in the presence or absence of PE. **(g)** Quantification analysis of cardiomyocyte size (n=5 independently prepared cardiomyocyte preparations/cultures). **(h)** Our proposed signaling pathway, involving Bcl-xL phosphorylation in hypertrophic stimuli-induced calcium-calcineurin-NFAT signaling. Phosphorylation of Bcl-xL at Ser14 by ERK disrupts its inhibitory interaction with IP3R, thereby augmenting Ca^{2+} signaling to promote compensatory hypertrophy in response to pressure overload. **(i)** Immunoblots showing the levels of autophagy-related proteins in the hearts of S14A homozygous KI and WT mice at baseline. **(j)** Representative immunoblots showing phosphorylation of Bcl-xL at Ser14 in WT mouse neonatal cardiac fibroblasts with time course after 50 nM angiotensin II (AngII) stimulation. m; minutes and h; hours. Lower panel shows densitometric analysis of relative expression of pBcl-xL (Ser14)/Bcl-xL. Kruskal-Wallis test with vehicle as the control. p values are shown in the figure (n=4, independently prepared cells). **(k)** Representative traces of cytosolic Ca^{2+} levels (F/F_0) in WT and S14A homozygous KI mouse cardiac fibroblasts with AngII stimulation. Right panel shows quantitative analysis of AngII-induced changes in cytosolic Ca^{2+} levels

(n=3-4, independently prepared cells). Two-way ANOVA with Tukey's multiple comparison test unless otherwise stated. **** $p < 0.0001$, *** $p < 0.001$, ** $p < 0.01$, * $p < 0.05$, ns not significant.

Data are mean \pm SEM.

Supplementary Figure 4

Supplementary Figure 4 Schema of our proposed mechanism. Activation of the MEK1-ERK1/2 pathway by hypertrophic stimuli promotes Bcl-xL-Ser14 phosphorylation, which disrupts its inhibitory interaction with IP3R, thereby augmenting Ca^{2+} release from the sarcoplasmic reticulum (SR) and calcineurin-NFAT signaling. This mechanism is crucial for the development of adaptive hypertrophy to suppress wall stress and prevent acute decompensated heart failure.

Supplementary Figure 5 Uncropped Western blot images

Supplementary Figure 5 Uncropped scan images of immunoblots membrane

Fig. 1a

Fig. 2h

Fig. 3f

Fig. 3c

Fig. 3d

Fig. 3i**Fig. 3k****Fig. 3l****Fig. 5c****Fig. 5d****Supplementary Fig. 1a****Supplementary Fig. 1c**
Supplementary Fig. 2b

Supplementary Fig. 2e

Supplementary Fig. 3d

Supplementary Fig. 2c

Supplementary Fig. 2d

Supplementary Fig. 3i

Supplementary Fig. 3j

REVIEWERS' COMMENTS

Reviewer #2 (Remarks to the Author):

The authors addressed my concerns sufficiently

Point by point responses

We appreciate all of the valuable comments from the reviewer.

Response to Reviewer #2:

The authors addressed my concerns sufficiently.

Thank you very much for your time and invaluable suggestions.